# IMPROVING DIFFUSION MODELS FOR INVERSE PROBLEMS USING OPTIMAL POSTERIOR COVARIANCE

## ABSTRACT

Recent diffusion models provide a promising alternative zero-shot solution to noisy linear inverse problems without retraining for specific inverse problems. In this paper, we propose the first unified framework for diffusion-based zero-shot methods from the view of approximating conditional posterior mean for the reverse process. We reveal that recent diffusion-based zero-shot methods are equivalent to making isotropic Gaussian approximation to intractable posterior distributions over clean images given diffused noisy images, with only difference in handcrafted design of isotropic posterior covariances. Inspired by this finding, we develop the optimal posterior covariance of the posterior distribution via maximum likelihood estimation. We provide a general solution based on three approaches specifically designed for posterior covariance optimization, by training from scratch and using pre-trained models with and without reverse covariances. Remarkably, the proposed framework can be achieved in a plug-and-play fashion based on pre-trained unconditional diffusion models by converting reverse covariances or via Monte Carlo estimation without reverse covariances. Experimental results demonstrate that the proposed framework significantly outperforms existing zero-shot methods and enhances the robustness to hyper-parameters.

## 1 INTRODUCTION

Noisy linear inverse problems are widely studied for a variety of tasks in the field of image processing, including denoising, inpainting, deblurring, super-resolution, and compressive sensing. The noisy linear inverse problems are formulated to accommodate to widely adopted degradation model where images are measured with linear projection under the noise corruption.

Recently, diffusion models have been emerging as promising methods for solving inverse problems. According to the training strategies, these methods can be categorized into two groups: 1) *supervised methods* that aim to learn a conditional diffusion model using datasets consisting of pairs of degraded and clean images (Saharia et al. (2022); Whang et al. (2022); Luo et al. (2023); Chan et al. (2023)), and 2) *zero-shot methods* that leverage pre-trained unconditional diffusion models for conditional sampling in various scenarios of inverse problems without the requirement of retraining. In this paper, we focus on the zero-shot methods that can accommodate to various tasks without retraining. To ensure the consistency of optimizing fidelity under conditional sampling, existing zero-shot methods either adopt projection onto the measurement subspace (Choi et al. (2021); Lugmayr et al. (2022); Song et al. (2022); Wang et al. (2023); Zhu et al. (2023)) or leverage the similar idea as classifier guidance (Song et al. (2021b); Dhariwal & Nichol (2021)) to modify the sampling process with the likelihood score (Song et al. (2023); Chung et al. (2023)).

Diffusion models initially establish a forward process that introduces noise to the original data $\boldsymbol{x}_0$, generating noisy data $\boldsymbol{x}_t$ at time $t$, and then implement the reverse process to generate $\boldsymbol{x}_0$ obeying the original data distribution. The key to realize the reverse process is the posterior mean $\mathbb{E}[\boldsymbol{x}_0|\boldsymbol{x}_t] = \int \boldsymbol{x}_0 p(\boldsymbol{x}_0|\boldsymbol{x}_t)\mathrm{d}\boldsymbol{x}_0$, which would be the optimal estimation of $\boldsymbol{x}_0$ given $\boldsymbol{x}_t$ in the sense of minimizing the mean square error. In the reverse process for conditional sampling in inverse problems, estimating $\boldsymbol{x}_0$ is based on both $\boldsymbol{x}_t$ and measurement $\boldsymbol{y}$; hence, the optimal estimation is determined by the *conditional posterior mean* $\mathbb{E}[\boldsymbol{x}_0|\boldsymbol{x}_t, \boldsymbol{y}]$.

In this paper, we provide the first unified framework for zero-shot methods from the view of approximating $\mathbb{E}[\boldsymbol{x}_0|\boldsymbol{x}_t, \boldsymbol{y}]$. We reveal that recent zero-shot methods (Song et al. (2023); Chung et al.

(2023); Wang et al. (2023); Zhu et al. (2023) are equivalent to isotropic Gaussian approximations to intractable posterior distributions $p_t(\boldsymbol{x}_0|\boldsymbol{x}_t)$ over clean images $\boldsymbol{x}_0$ given diffused noisy images $\boldsymbol{x}_t$ but differ in various handcrafted isotropic posterior covariances. This finding inspires us to develop optimal posterior covariances via maximum likelihood estimation (MLE) to further optimize recent zero-shot methods. Consequently, we achieve a general solution to zero-shot methods by developing three approaches for optimizing posterior covariances by training from scratch and using pre-trained models with and without reverse covariances. Remarkably, leveraging the pre-trained unconditional diffusion models (Nichol & Dhariwal (2021)), the proposed framework can be applied to various zero-shot methods in a plug-and-play fashion. Experimental results demonstrate that the proposed framework significantly outperforms existing zero-shot methods and enhances the robustness to hyperparameters.

## 2 PRELIMINARIES

### 2.1 SOLVING INVERSE PROBLEMS UNDER BAYESIAN FRAMEWORK

In inverse problems, we wish to recover the unknown signal $\boldsymbol{x}_0 \in \mathbb{R}^d$ given noisy measurements $\boldsymbol{y} \in \mathbb{R}^m$, where

$$\boldsymbol{y} = \boldsymbol{A}\boldsymbol{x}_0 + \boldsymbol{n} \tag{1}$$

It is usually assumed that by the expert knowledge of the sensing device, $\boldsymbol{A} \in \mathbb{R}^{m \times d}$ is known and $\boldsymbol{n} \sim \mathcal{N}(\boldsymbol{0}, \sigma^2 \boldsymbol{I})$ is an i.i.d. additive Gaussian noise with a known standard deviation of $\sigma$. However, recovering $\boldsymbol{x}_0$ from the degraded measurements $\boldsymbol{y}$ becomes ill-posed due to the underdetermined $\boldsymbol{A}$ and the existence of noise. To solve the ill-posed inverse problem, a prior is required to constrain the solution according to desired image statistics (Feng et al. (2023)). From a Bayesian perspective, the prior is described by assuming $\boldsymbol{x}_0$ obeys an unknown prior distribution $p(\boldsymbol{x}_0)$, and we have a likelihood function $p(\boldsymbol{y}|\boldsymbol{x}_0) = \mathcal{N}(\boldsymbol{y}|\boldsymbol{A}\boldsymbol{x}_0, \sigma^2 \boldsymbol{I})$ determined by equation 1. We solve inverse problems by formulating a posterior distribution $p(\boldsymbol{x}_0|\boldsymbol{y})$ over clean images given noisy measurements, which in principle is determined given $p(\boldsymbol{x}_0)$ and $p(\boldsymbol{y}|\boldsymbol{x}_0)$ by Bayes' theorem: $p(\boldsymbol{x}_0|\boldsymbol{y}) = p(\boldsymbol{x}_0)p(\boldsymbol{y}|\boldsymbol{x}_0) / \int p(\boldsymbol{x}_0)p(\boldsymbol{y}|\boldsymbol{x}_0)\mathrm{d}\boldsymbol{x}_0$.

### 2.2 DIFFUSION MODELS AND CONDITIONING

The goal is using diffusion models to model the complex posterior distribution $p(\boldsymbol{x}_0|\boldsymbol{y})$ for solving inverse problems. To begin with, let us define a family of Gaussian perturbation kernels $p_t(\boldsymbol{x}_t|\boldsymbol{x}_0)$ of $\boldsymbol{x}_0 \sim p(\boldsymbol{x}_0)$ by injecting i.i.d. Gaussian noise of standard deviation $\sigma_t$ to $\boldsymbol{x}_0$ and then scaling by the factor of $s_t$, i.e., $p_t(\boldsymbol{x}_t|\boldsymbol{x}_0) = \mathcal{N}(\boldsymbol{x}_t|s_t\boldsymbol{x}_0, s_t^2\sigma_t^2\boldsymbol{I})$. The standard deviation $\sigma_t$ is monotonically increased with respect to time $t \in [0, T]$, starting from $\sigma_0 = 0$ and reaching a value of $\sigma_T$ being much larger than the standard deviation of $p(\boldsymbol{x}_0)$, ensuring that samples from $\boldsymbol{x}_T \sim p(\boldsymbol{x}_T)$ are indistinguishable to samples from $\mathcal{N}(\boldsymbol{0}, s_T^2\sigma_T^2\boldsymbol{I})$. Since $\boldsymbol{x}_t$ is independent to $\boldsymbol{y}$ once $\boldsymbol{x}_0$ is known, we characterize a joint distribution between $\boldsymbol{x}_0, \boldsymbol{y}$ and $\boldsymbol{x}_t$ as $p_t(\boldsymbol{x}_0, \boldsymbol{y}, \boldsymbol{x}_t) = p(\boldsymbol{x}_0)p(\boldsymbol{y}|\boldsymbol{x}_0)p_t(\boldsymbol{x}_t|\boldsymbol{x}_0)$, which can also be represented by an probabilistic graphical model $\boldsymbol{y} \leftarrow \boldsymbol{x}_0 \rightarrow \boldsymbol{x}_t$. There exist multiple formulations of diffusion models in the literature (Song & Ermon (2019); Ho et al. (2020); Song et al. (2021a); Kingma et al. (2021); Song et al. (2021b); Karras et al. (2022)). Here we use ordinary differential equation (ODE) formulation and select $s_t = 1, \sigma_t = t$ suggested in Karras et al. (2022) for simplicity. Let us consider the following ODE:

$$\mathrm{d}\boldsymbol{x}_t = \frac{\boldsymbol{x}_t - \mathbb{E}[\boldsymbol{x}_0|\boldsymbol{x}_t]}{t}\mathrm{d}t, \ \ \boldsymbol{x}_T \sim p_T(\boldsymbol{x}_T) \tag{2}$$

The only source of randomness of such ODE is induced by the initial sample $\boldsymbol{x}_T \sim p_T(\boldsymbol{x}_T)$, and it possesses an important property that $\boldsymbol{x}_t$ generated by the ODE maintains the exact same marginals to $\boldsymbol{x}_t$ obtained by injecting Gaussian noise to $\boldsymbol{x}_0$, i.e., $p_t(\boldsymbol{x}_t)$. Generally, diffusion models approximate $\mathbb{E}[\boldsymbol{x}_0|\boldsymbol{x}_t]$ with a time-dependent denoiser $D_t(\boldsymbol{x}_t)$ (which is referred to as the unconditional diffusion model throughout the paper), trained via minimizing a simple $L_2$ loss for all $t \in [0, T]$:

$$\min_{D_t} \mathbb{E}_{p_t(\boldsymbol{x}_0, \boldsymbol{x}_t)}[\|\boldsymbol{x}_0 - D_t(\boldsymbol{x}_t)\|_2^2] \tag{3}$$

With sufficient data and model capacity, the optimal $D_t(\boldsymbol{x}_t)$ is the minimum mean square error (MMSE) estimator of $\boldsymbol{x}_0$ given $\boldsymbol{x}_t$ and equals to the posterior mean $\mathbb{E}[\boldsymbol{x}_0|\boldsymbol{x}_t]$. Thus, samples

| Methods | Guidance | $r_t$ |
|---|---|---|
| DPS (Chung et al. (2023)) | I | approach 0 |
| $\Pi$GDM (Song et al. (2023)) | I | $\sqrt{\sigma_t^2/(\sigma_t^2 + 1)}$ |
| DDNM (Wang et al. (2023)) | II | any |
| DiffPIR (Zhu et al. (2023)) | II | $\sigma_t/\sqrt{\lambda}$ |

Table 1: **Summary of zero-shot methods.** Recent methods can be regarded as making isotropic Gaussian approximations to the posterior distributions over clean images given noisy images.

from $p(\boldsymbol{x}_0)$ can be obtained by sampling $\boldsymbol{x}_T$ from $\mathcal{N}(\boldsymbol{0}, s_T^2\sigma_T^2\boldsymbol{I})$ and then simulating equation 2 from $t = T$ to $t = 0$ using black box ODE solver that replaces $\mathbb{E}[\boldsymbol{x}_0|\boldsymbol{x}_t]$ with a well-trained $D_t(\boldsymbol{x}_t)$.

To solve inverse problems, we are interested in $p(\boldsymbol{x}_0|\boldsymbol{y})$ and formulate an ODE whose marginals are $p_t(\boldsymbol{x}_t|\boldsymbol{y})$ such that we can simulate the ODE to sample from $p(\boldsymbol{x}_0|\boldsymbol{y})$. The desired ODE is given by

$$\mathrm{d}\boldsymbol{x}_t = \frac{\boldsymbol{x}_t - \mathbb{E}[\boldsymbol{x}_0|\boldsymbol{x}_t, \boldsymbol{y}]}{t}\mathrm{d}t, \ \ \boldsymbol{x}_T \sim p_T(\boldsymbol{x}_T|\boldsymbol{y}) \tag{4}$$

For sufficiently large $\sigma_T$, samples $\boldsymbol{x}_T \sim p(\boldsymbol{x}_T|\boldsymbol{y})$ are also indistinguishable to samples from $\mathcal{N}(\boldsymbol{0}, s_T^2\sigma_T^2\boldsymbol{I})$ (Appendix D.2, Dhariwal & Nichol (2021)). Therefore, the unconditioned sampling procedure of equation 2 can be utilized for the conditioned sampling as equation 4, except for introducing the conditional posterior mean $\mathbb{E}[\boldsymbol{x}_0|\boldsymbol{x}_t, \boldsymbol{y}]$ to substitute $\mathbb{E}[\boldsymbol{x}_0|\boldsymbol{x}_t]$[1].

# 3 A UNIFIED FRAMEWORK FOR SOLVING INVERSE PROBLEMS USING UNCONDITIONAL DIFFUSION MODELS

To solve inverse problems, a common approach is to approximate $\mathbb{E}[\boldsymbol{x}_0|\boldsymbol{x}_t, \boldsymbol{y}]$ by training a conditional diffusion model using supervised learning. However, this approach requires training separate models for different inverse problems, which can be computationally demanding. In this work, we aim to leverage pre-trained unconditional diffusion models for conditional sampling in inverse problems. This approach allows us to utilize the models without the need for additional training. We summarize three steps for solving the target inverse problem:

1. Obtain an estimation of $\mathbb{E}[\boldsymbol{x}_0|\boldsymbol{x}_t]$ through the unconditional diffusion model;

2. Estimate the conditional posterior mean $\mathbb{E}[\boldsymbol{x}_0|\boldsymbol{x}_t, \boldsymbol{y}]$ based on $\mathbb{E}[\boldsymbol{x}_0|\boldsymbol{x}_t]$;

3. Substitute the estimation for $\mathbb{E}[\boldsymbol{x}_0|\boldsymbol{x}_t]$ in sampling process (see equation 2 and equation 4).

Recent diffusion-based zero-shot methods, including DPS (Chung et al. (2023)), $\Pi$GDM (Song et al. (2023)), DDNM (Wang et al. (2023)), and DiffPIR (Zhu et al. (2023)), follow the three steps, but they vary in the approach to estimating $\mathbb{E}[\boldsymbol{x}_0|\boldsymbol{x}_t, \boldsymbol{y}]$ (step 2). Here, we establish a unified framework embracing these methods, showing that they in fact estimate $\mathbb{E}[\boldsymbol{x}_0|\boldsymbol{x}_t, \boldsymbol{y}]$ via making isotropic Gaussian approximations $\mathcal{N}(\boldsymbol{x}_0|\mathbb{E}[\boldsymbol{x}_0|\boldsymbol{x}_t], r_t^2\boldsymbol{I})$ to the intractable posterior $p_t(\boldsymbol{x}_0|\boldsymbol{x}_t)$ with different $r_t$, as shown in Table 1. We classify these methods into two categories, Guidance I and II, with detailed descriptions as follows.

## 3.1 TYPE I GUIDANCE: APPROXIMATING THE LIKELIHOOD SCORE FUNCTION

We classify DPS and $\Pi$GDM into one category, referred to as Type I guidance. Type I guidance is closely related to classifier guidance (Dhariwal & Nichol (2021)), where the conditional posterior mean $\mathbb{E}[\boldsymbol{x}_0|\boldsymbol{x}_t, \boldsymbol{y}]$ is approximated based on the following proposition:

**Proposition 1.** *The conditional posterior mean equals to the posterior mean drifted by scaled likelihood score function, formally*

$$\mathbb{E}[\boldsymbol{x}_0|\boldsymbol{x}_t, \boldsymbol{y}] = \mathbb{E}[\boldsymbol{x}_0|\boldsymbol{x}_t] + s_t\sigma_t^2\nabla_{\boldsymbol{x}_t}\log p_t(\boldsymbol{y}|\boldsymbol{x}_t). \tag{5}$$

We can leverage the pre-trained unconditional diffusion model to approximate $\mathbb{E}[\boldsymbol{x}_0|\boldsymbol{x}_t]$. However, the likelihood score $\nabla_{\boldsymbol{x}_t}\log p_t(\boldsymbol{y}|\boldsymbol{x}_t)$ is computationally intractable. In inverse problems, only the

---

[1]Of note, this conclusion can be extended to DDPM, DDIM, or SDE-formalized diffusion models.

likelihood $p(\boldsymbol{y}|\boldsymbol{x}_0)$ at $t = 0$ is known, while the likelihood for any $t > 0$ is given by an intractable integral over all possible $\boldsymbol{x}_0$:

$$p_t(\boldsymbol{y}|\boldsymbol{x}_t) = \int p(\boldsymbol{y}|\boldsymbol{x}_0)p_t(\boldsymbol{x}_0|\boldsymbol{x}_t)\mathrm{d}\boldsymbol{x}_0 \tag{6}$$

**DPS (Chung et al. (2023))** DPS can be viewed as approximating $p_t(\boldsymbol{x}_0|\boldsymbol{x}_t)$ using a delta distribution centered at the posterior mean $\mathbb{E}[\boldsymbol{x}_0|\boldsymbol{x}_t]$, which can be regarded as the limit of the Gaussian $\mathcal{N}(\boldsymbol{x}_0|\mathbb{E}[\boldsymbol{x}_0|\boldsymbol{x}_t], r_t^2\boldsymbol{I})$ when the variance $r_t^2$ approaches zero. In such case, the likelihood $p_t(\boldsymbol{y}|\boldsymbol{x}_t)$ is approximated by

$$p_t(\boldsymbol{y}|\boldsymbol{x}_t) \approx \int p(\boldsymbol{y}|\boldsymbol{x}_0)\delta(\boldsymbol{x}_0 - \mathbb{E}[\boldsymbol{x}_0|\boldsymbol{x}_t])\mathrm{d}\boldsymbol{x}_0 = p(\boldsymbol{y}|\boldsymbol{x}_0 = \mathbb{E}[\boldsymbol{x}_0|\boldsymbol{x}_t]). \tag{7}$$

However, simply replacing $p_t(\boldsymbol{y}|\boldsymbol{x}_t)$ with $p(\boldsymbol{y}|\boldsymbol{x}_0 = \mathbb{E}[\boldsymbol{x}_0|\boldsymbol{x}_t])$ to compute the likelihood score does not perform well in practice. For example, $\log p(\boldsymbol{y}|\boldsymbol{x}_0 = \mathbb{E}[\boldsymbol{x}_0|\boldsymbol{x}_t])$ becomes unbounded in the noiseless case (i.e., $\sigma \to 0$). To address this issue, DPS empirically adjusted the strength of the guidance by replacing the likelihood score $\nabla_{\boldsymbol{x}_t} \log p_t(\boldsymbol{y}|\boldsymbol{x}_t)$ with $-\zeta_t \nabla_{\boldsymbol{x}_t}\|\boldsymbol{y} - \boldsymbol{A}\mathbb{E}[\boldsymbol{x}_0|\boldsymbol{x}_t]\|_2^2$, where $\zeta_t = \zeta/\|\boldsymbol{y} - \boldsymbol{A}\mathbb{E}[\boldsymbol{x}_0|\boldsymbol{x}_t]\|$ with a hyper-parameter $\zeta$.

**ΠGDM (Song et al. (2023))** The simple delta distribution used in DPS is a very rough approximation to $p_t(\boldsymbol{x}_0|\boldsymbol{x}_t)$ as it completely ignores the uncertainty of $\boldsymbol{x}_0$ given $\boldsymbol{x}_t$. As $t$ increases, $p_t(\boldsymbol{x}_0|\boldsymbol{x}_t)$ has larger uncertainty that close to the original data distribution, in which case choosing $r_t > 0$ could be more reasonable. In ΠGDM, $r_t$ is heuristically selected as $\sqrt{\sigma_t^2/(1 + \sigma_t^2)}$ under the assumption that the original data distribution $p(\boldsymbol{x}_0)$ is the standard normal distribution $\mathcal{N}(\boldsymbol{0}, \boldsymbol{I})$. In such case, the likelihood $p_t(\boldsymbol{y}|\boldsymbol{x}_t)$ is approximated by

$$p_t(\boldsymbol{y}|\boldsymbol{x}_t) \approx \int \mathcal{N}(\boldsymbol{y}|\boldsymbol{A}\boldsymbol{x}_0, \sigma^2\boldsymbol{I})\mathcal{N}(\boldsymbol{x}_0|\mathbb{E}[\boldsymbol{x}_0|\boldsymbol{x}_t], r_t^2\boldsymbol{I})\mathrm{d}\boldsymbol{x}_0 = \mathcal{N}(\boldsymbol{y}|\boldsymbol{A}\mathbb{E}[\boldsymbol{x}_0|\boldsymbol{x}_t], \sigma^2\boldsymbol{I} + r_t^2\boldsymbol{A}\boldsymbol{A}^T) \tag{8}$$

In equation 8, the score is a vector-Jacobian product that can be computed using back-propagation (Song et al. (2023)). Proposition 1 can be used for achieving different purposes, including refining the estimation of $\boldsymbol{x}_0$ for learning sampling patterns in MRI (Ravula et al., 2023) and realizing guidance for flow-based generative models to solve inverse problems (Pokle et al., 2023).

### 3.2 TYPE II GUIDANCE: APPROXIMATING THE CONDITIONAL POSTERIOR MEAN USING PROXIMAL SOLUTION

We classify DiffPIR and DDNM into one category, referred to as Type II guidance. Type II guidance approximates the conditional posterior mean $\mathbb{E}[\boldsymbol{x}_0|\boldsymbol{x}_t, \boldsymbol{y}]$ by finding a solution $\hat{\boldsymbol{x}}_0^{(t)}$ which is both close to $\mathbb{E}[\boldsymbol{x}_0|\boldsymbol{x}_t]$ and consistent with the measurement $\boldsymbol{y}$. Compared to Type I guidance, Type II guidance is more efficient by circumventing the expensive back-propagation through the denoiser. We show that, though Type II guidance is developed from completely different perspective, it can also be regarded as making isotropic Gaussian approximation to the true posterior $p_t(\boldsymbol{x}_0|\boldsymbol{x}_t)$.

**DiffPIR (Zhu et al. (2023))** The core step in DiffPIR is replacing the denosing results $\mathbb{E}[\boldsymbol{x}_0|\boldsymbol{x}_t]$ in the unconditional sampling process with the solution of the following proximal problem:

$$\hat{\boldsymbol{x}}_0^{(t)} = \arg\min_{\boldsymbol{x}_0}\|\boldsymbol{y} - \boldsymbol{A}\boldsymbol{x}_0\|^2 + \rho_t\|\boldsymbol{x}_0 - \mathbb{E}[\boldsymbol{x}_0|\boldsymbol{x}_t]\|^2, \quad \rho_t = \frac{\lambda\sigma^2}{\sigma_t^2}, \tag{9}$$

where $\lambda$ is a hyper-parameter. Our key insight is that such a step can still be interpreted as approximating $\mathbb{E}[\boldsymbol{x}_0|\boldsymbol{x}_t, \boldsymbol{y}]$ via making isotropic Gaussian approximations $\mathcal{N}(\boldsymbol{x}_0|\mathbb{E}[\boldsymbol{x}_0|\boldsymbol{x}_t], r_t^2\boldsymbol{I})$ to the intractable posterior distributions $p_t(\boldsymbol{x}_0|\boldsymbol{x}_t)$. In fact, the mean of the distribution $p_t(\boldsymbol{x}_0|\boldsymbol{x}_t, \boldsymbol{y}) \propto p(\boldsymbol{y}|\boldsymbol{x}_0)p_t(\boldsymbol{x}_0|\boldsymbol{x}_t)$ [2] is $\mathbb{E}[\boldsymbol{x}_0|\boldsymbol{x}_t, \boldsymbol{y}]$. However, it is intractable to find the mean of $p_t(\boldsymbol{x}_0|\boldsymbol{x}_t, \boldsymbol{y})$. To address this, we introduce a Gaussian approximation $q_t(\boldsymbol{x}_0|\boldsymbol{x}_t) = \mathcal{N}(\boldsymbol{x}_0|\mathbb{E}[\boldsymbol{x}_0|\boldsymbol{x}_t], r_t^2\boldsymbol{I})$ of $p_t(\boldsymbol{x}_0|\boldsymbol{x}_t)$ and approximate $p_t(\boldsymbol{x}_0|\boldsymbol{x}_t, \boldsymbol{y})$ using $q_t(\boldsymbol{x}_0|\boldsymbol{x}_t, \boldsymbol{y}) \propto p(\boldsymbol{y}|\boldsymbol{x}_0)q_t(\boldsymbol{x}_0|\boldsymbol{x}_t)$. Therefore,

---

[2] Here we leverage the conditional independent between $\boldsymbol{y}$ and $\boldsymbol{x}_t$ given $\boldsymbol{x}_0$, so $p_t(\boldsymbol{y}|\boldsymbol{x}_0, \boldsymbol{x}_t) = p(\boldsymbol{y}|\boldsymbol{x}_0)$.

$q_t(\boldsymbol{x}_0|\boldsymbol{x}_t, \boldsymbol{y})$ is a Gaussian with its mean obtained by solving the optimization problem:

$$\mathbb{E}_q[\boldsymbol{x}_0|\boldsymbol{x}_t, \boldsymbol{y}] = \arg\max_{\boldsymbol{x}_0} \log q_t(\boldsymbol{x}_0|\boldsymbol{x}_t, \boldsymbol{y}) = \arg\max_{\boldsymbol{x}_0} \log p(\boldsymbol{y}|\boldsymbol{x}_0) + \log q_t(\boldsymbol{x}_0|\boldsymbol{x}_t)$$

$$= \arg\min_{\boldsymbol{x}_0}\|\boldsymbol{y} - \boldsymbol{A}\boldsymbol{x}_0\|^2 + \frac{\sigma^2}{r_t^2}\|\boldsymbol{x}_0 - \mathbb{E}[\boldsymbol{x}_0|\boldsymbol{x}_t]\|^2, \tag{10}$$

where $\mathbb{E}_q[\boldsymbol{x}_0|\boldsymbol{x}_t, \boldsymbol{y}]$ denotes the mean of $q_t(\boldsymbol{x}_0|\boldsymbol{x}_t, \boldsymbol{y})$. Note that equation 9 used in DiffPIR can be obtained from equation 10 by setting $r_t = \sigma_t/\sqrt{\lambda}$. Therefore, DiffPIR can be viewed as using the Gaussian approximation $q_t(\boldsymbol{x}_0|\boldsymbol{x}_t) = \mathcal{N}(\boldsymbol{x}_0|\mathbb{E}[\boldsymbol{x}_0|\boldsymbol{x}_t], (\sigma_t^2/\lambda)\boldsymbol{I})$ for $p_t(\boldsymbol{x}_0|\boldsymbol{x}_t)$.

**DDNM (Wang et al. (2023))** The core step of DDNM resorts to range-null space decomposition. Specifically, to enforce the solution $\hat{\boldsymbol{x}}_0^{(t)}$ satisfying the measurement consistency $\boldsymbol{y} = \boldsymbol{A}\hat{\boldsymbol{x}}_0^{(t)}$, DDNM replaces the range space component of $\mathbb{E}[\boldsymbol{x}_0|\boldsymbol{x}_t]$ with $\boldsymbol{A}^\dagger\boldsymbol{y}$ but keeps the null space component unchanged, i.e., $\hat{\boldsymbol{x}}_0^{(t)} = \boldsymbol{A}^\dagger\boldsymbol{y} + (\boldsymbol{I} - \boldsymbol{A}^\dagger\boldsymbol{A})\mathbb{E}[\boldsymbol{x}_0|\boldsymbol{x}_t]$. We find that the replacement of range space component with $\boldsymbol{A}^\dagger\boldsymbol{y}$ is equivalent to replacing $\mathbb{E}[\boldsymbol{x}_0|\boldsymbol{x}_t]$ with $\mathbb{E}_q[\boldsymbol{x}_0|\boldsymbol{x}_t, \boldsymbol{y}]$ when the measurement noise $\sigma$ vanishes. Thus, DDNM is equivalent to approximating $p_t(\boldsymbol{x}_0|\boldsymbol{x}_t)$ using an isotropic Gaussian. In Proposition 2, we formalize the equivalency for DDNM.

**Proposition 2.** *For any $r_t > 0$, $\mathbb{E}_q[\boldsymbol{x}_0|\boldsymbol{x}_t, \boldsymbol{y}]$ approaches $\hat{\boldsymbol{x}}_0^{(t)}$ used in DDNM as the variance of measurement noise approaches zero. Formally, we have*

$$\lim_{\sigma\to 0}\mathbb{E}_q[\boldsymbol{x}_0|\boldsymbol{x}_t, \boldsymbol{y}] = \boldsymbol{A}^\dagger\boldsymbol{y} + (\boldsymbol{I} - \boldsymbol{A}^\dagger\boldsymbol{A})\mathbb{E}[\boldsymbol{x}_0|\boldsymbol{x}_t]. \tag{11}$$

# 4 IMPROVING DIFFUSION MODELS FOR INVERSE PROBLEMS USING OPTIMAL POSTERIOR COVARIANCE

With the above discussion, prior works can be regarded as making isotropic Gaussian approximations to the intractable posterior distributions $p_t(\boldsymbol{x}_0|\boldsymbol{x}_t)$ over clean images $\boldsymbol{x}_0$ given diffused noisy images $\boldsymbol{x}_t$. This motivates us to investigate whether the performance of existing approaches can be further improved through the use of "better" posterior covariances in theses Gaussian approximations. Specifically, we consider to approximate $p_t(\boldsymbol{x}_0|\boldsymbol{x}_t)$ using Gaussian with posterior covariance $\boldsymbol{\Sigma}_t(\boldsymbol{x}_t)$, such that $q_t(\boldsymbol{x}_0|\boldsymbol{x}_t) = \mathcal{N}(\boldsymbol{x}_0|\mathbb{E}[\boldsymbol{x}_0|\boldsymbol{x}_t], \boldsymbol{\Sigma}_t(\boldsymbol{x}_t))$.

**Type I guidance:** The likelihood is approximated as follows in a similar way of equation 8:

$$p_t(\boldsymbol{y}|\boldsymbol{x}_t) \approx \mathcal{N}(\boldsymbol{y}|\boldsymbol{A}\mathbb{E}[\boldsymbol{x}_0|\boldsymbol{x}_t], \sigma^2\boldsymbol{I} + \boldsymbol{A}\boldsymbol{\Sigma}_t(\boldsymbol{x}_t)\boldsymbol{A}^T) \tag{12}$$

**Type II guidance:** We consider to solve the following *auto-weighted* proximal problem:

$$\mathbb{E}_q[\boldsymbol{x}_0|\boldsymbol{x}_t, \boldsymbol{y}] = \arg\min_{\boldsymbol{x}_0}\|\boldsymbol{y} - \boldsymbol{A}\boldsymbol{x}_0\|^2 + \sigma^2\|\boldsymbol{x}_0 - \mathbb{E}[\boldsymbol{x}_0|\boldsymbol{x}_t]\|_{\boldsymbol{\Sigma}_t^{-1}(\boldsymbol{x}_t)}^2 \tag{13}$$

where $\|\boldsymbol{x}\|_{\boldsymbol{\Lambda}}^2$ denotes $\boldsymbol{x}^T\boldsymbol{\Lambda}\boldsymbol{x}$. For commonly seen $\boldsymbol{A}$, the score of equation 12 and the minimizer of equation 13 can be obtained by deriving closed-form solution under isotropic posterior covariance case $\boldsymbol{\Sigma}_t(\boldsymbol{x}_t) = r_t^2(\boldsymbol{x}_t)\boldsymbol{I}$, or using conjugate gradient method (CG) for more general case (details in Appendix B and Appendix C).

Below, we develop three alternative ways to achieve optimal posterior covariance via MLE, respectively designed for three common cases: 1) pre-trained unconditional diffusion model is unavailable (Section 4.1), 2) reverse covariance prediction is available from the given unconditional diffusion model (Section 4.2), and 3) reverse covariance prediction is not available (Section 4.3).

## 4.1 OPTIMIZING POSTERIOR COVARIANCE BY LEARNING FROM SCRATCH

We can additionally learn an optimal covariance of $q_t(\boldsymbol{x}_0|\boldsymbol{x}_t)$ in the pre-training stage of unconditional diffusion model, in contrast to the prior works that only learn the mean of $q_t(\boldsymbol{x}_0|\boldsymbol{x}_t)$, i.e., the denoiser. Specifically, the mean and covariance of $q_t(\boldsymbol{x}_0|\boldsymbol{x}_t)$ is parameterized using time-dependent neural networks $\boldsymbol{\mu}_t$ and $\boldsymbol{\Sigma}_t$, such that $q_t(\boldsymbol{x}_0|\boldsymbol{x}_t) = \mathcal{N}(\boldsymbol{\mu}_t(\boldsymbol{x}_t), \boldsymbol{\Sigma}_t(\boldsymbol{x}_t))$. In the pre-training stage, we learn $q_t(\boldsymbol{x}_0|\boldsymbol{x}_t)$ by minimizing the weighting integral of expected forward KL divergence between $p_t(\boldsymbol{x}_0|\boldsymbol{x}_t)$ and $q_t(\boldsymbol{x}_0|\boldsymbol{x}_t)$.

$$\min_q \int \omega_t \mathbb{E}_{p_t(\boldsymbol{x}_t)}[D_{KL}(p_t(\boldsymbol{x}_0|\boldsymbol{x}_t)\|q_t(\boldsymbol{x}_0|\boldsymbol{x}_t))]\mathrm{d}t \tag{14}$$

Since minimizing the KL divergence $D_{KL}(p_t(\boldsymbol{x}_0|\boldsymbol{x}_t)\|q_t(\boldsymbol{x}_0|\boldsymbol{x}_t))$ equals to maximizing the log-likelihood of $q_t(\boldsymbol{x}_0|\boldsymbol{x}_t)$, equation 14 can be optimized in a tractable way as

$$\max_q \int \mathbb{E}_{p_t(\boldsymbol{x}_0,\boldsymbol{x}_t)}[\omega_t \log q_t(\boldsymbol{x}_0|\boldsymbol{x}_t)]\mathrm{d}t. \tag{15}$$

However, letting all the elements of $\boldsymbol{\Sigma}_t(\boldsymbol{x}_t)$ be learnable is computationally demanding, especially for high-dimensional signals. Therefore, we restrict to diagonal posterior covariance: $\boldsymbol{\Sigma}_t(\boldsymbol{x}_t) = \mathrm{diag}[\boldsymbol{r}_t^2(\boldsymbol{x}_t)]$, where $\boldsymbol{r}_t^2(\boldsymbol{x}_t)$ is posterior variances[3]. Besides, as shown in Section 4.2, such restriction also allow us to directly leverage off-the-shelf reverse covariance prediction to obtain posterior covariance prediction. Also, in Proposition 3, we develop the solution to equation 15 under diagonal posterior covariance.

**Proposition 3.** *The optimal mean $\boldsymbol{\mu}_t^*(\boldsymbol{x}_t)$ and the optimal posterior variances $\boldsymbol{r}_t^{*2}(\boldsymbol{x}_t)$ to equation 15 are obtained by*

$$\boldsymbol{\mu}_t^*(\boldsymbol{x}_t) = \mathbb{E}[\boldsymbol{x}_0|\boldsymbol{x}_t] \tag{16}$$

$$\boldsymbol{r}_t^{*2}(\boldsymbol{x}_t) = \mathbb{E}_{p_t(\boldsymbol{x}_0|\boldsymbol{x}_t)}[(\boldsymbol{x}_0 - \mathbb{E}[\boldsymbol{x}_0|\boldsymbol{x}_t])^2] \tag{17}$$

Proposition 3 shows that the optimal mean $\boldsymbol{\mu}_t^*(\boldsymbol{x}_t)$ of $q_t(\boldsymbol{x}_0|\boldsymbol{x}_t)$ is the MMSE estimator $\mathbb{E}[\boldsymbol{x}_0|\boldsymbol{x}_t]$. This result suggests that unconditional sampling can be achieved by plugging a well-trained $\boldsymbol{\mu}_t(\boldsymbol{x}_t)$ in equation 2, while $\boldsymbol{\mu}_t$ and $\boldsymbol{\Sigma}_t$ are used together for solving inverse problems.

## 4.2 Optimizing Posterior Variances by Converting Optimal Reverse Variances

In addition to obtain optimal posterior covariance by training diffusion models from scratch using equation 14, we provide an alternative solution to leverage the pre-trained diffusion models. Recent pre-trained diffusion models often predict the optimal *reverse variances* (Nichol & Dhariwal (2021)) for improving performance when using ancestral sampling of *discrete* diffusion models (or denoising diffusion probabilistic models (DDPM)) (Ho et al. (2020); Sohl-Dickstein et al. (2015)). To avoid pre-training, we directly convert the optimization of posterior variances to the optimization reverse variances. To this end, we establish a connection between the optimal posterior variances $\boldsymbol{r}_t^{*2}(\boldsymbol{x}_t)$ and the optimal reverse variances $\boldsymbol{v}_t^{*2}(\boldsymbol{x}_t)$ proposed in (Bao et al. (2022b)).

Unlike continuous ODE formulation introduced in Section 2.2, diffusion models pre-trained under the DDPM framework are latent variable models defined by $p(\boldsymbol{x}_0) = \int p(\boldsymbol{x}_{0:T})\mathrm{d}\boldsymbol{x}_{1:T}$. The joint distribution $p(\boldsymbol{x}_{0:T})$ is referred to as the *reverse process* defined as a Markov chain of learnable Gaussian transitions starting at $p(\boldsymbol{x}_T) = \mathcal{N}(\boldsymbol{0}, \boldsymbol{I})$ (Ho et al. (2020)):

$$p(\boldsymbol{x}_{0:T}) = p(\boldsymbol{x}_T) \prod_{t=1}^T p(\boldsymbol{x}_{t-1}|\boldsymbol{x}_t), \quad p(\boldsymbol{x}_{t-1}|\boldsymbol{x}_t) = \mathcal{N}(\boldsymbol{x}_{t-1}|\boldsymbol{m}_t(\boldsymbol{x}_t), \boldsymbol{C}_t(\boldsymbol{x}_t)). \tag{18}$$

DDPM defines a forward process $q(\boldsymbol{x}_{1:T}|\boldsymbol{x}_0)$ by gradually injecting noise to the data. Please refer to Appendix A.4 for the detailed definition of $q(\boldsymbol{x}_{1:T}|\boldsymbol{x}_0)$. We fit $p(\boldsymbol{x}_0)$ to the original data distribution $q(\boldsymbol{x}_0)$ by minimizing the KL divergence between the forward and the reverse processes:

$$\min_p D_{KL}(q(\boldsymbol{x}_{0:T})\|p(\boldsymbol{x}_{0:T})), \quad q(\boldsymbol{x}_{0:T}) = q(\boldsymbol{x}_0)q(\boldsymbol{x}_{1:T}|\boldsymbol{x}_0) \tag{19}$$

Bao et al. (2022b) propose a theoretical result that determines the optimal solution to equation 19 under the signal-independent isotropic covariance case: $\boldsymbol{C}_t(\boldsymbol{x}_t) = c_t^2\boldsymbol{I}$, and generalize the result in Bao et al. (2022a). In Theorem 1, we present the result for signal-dependent diagonal covariance $\boldsymbol{C}_t(\boldsymbol{x}_t) = \mathrm{diag}[\boldsymbol{v}_t^2(\boldsymbol{x}_t)]$ as used in (Nichol & Dhariwal (2021)).

**Theorem 1.** *Let $\boldsymbol{C}_t(\boldsymbol{x}_t) = \mathrm{diag}[\boldsymbol{v}_t^2(\boldsymbol{x}_t)]$ be a signal-dependent diagonal covariance for the reverse covariance. When $\tilde{\boldsymbol{\mu}}_t, \beta_t, \bar{\alpha}_t, \tilde{\beta}_t$ are determined by the forward process $q(\boldsymbol{x}_{1:T}|\boldsymbol{x}_0)$, the optimal solutions $\boldsymbol{m}_t^*(\boldsymbol{x}_t)$ and $\boldsymbol{v}_t^{*2}(\boldsymbol{x}_t)$ to equation 19 are*

$$\boldsymbol{m}_t^*(\boldsymbol{x}_t) = \tilde{\boldsymbol{\mu}}_t(\boldsymbol{x}_t, \mathbb{E}[\boldsymbol{x}_0|\boldsymbol{x}_t]), \tag{20}$$

$$\boldsymbol{v}_t^{*2}(\boldsymbol{x}_t) = \tilde{\beta}_t + \left(\frac{\sqrt{\bar{\alpha}_{t-1}}\beta_t}{1 - \bar{\alpha}_t}\right)^2 \cdot \boldsymbol{r}_t^{*2}(\boldsymbol{x}_t) \tag{21}$$

---

[3]Variances refer to the diagonal elements of the covariance matrix.

| | Inpaint (Random) | Deblur (Gaussian) | Deblur (Motion) | Super resolution ($4\times$) |
|---|---|---|---|---|
| Avg. MAD | $1.6975 \times 10^{-6}$ | $6.4629 \times 10^{-4}$ | $1.5461 \times 10^{-3}$ | $1.1937 \times 10^{-2}$ |

Table 2: **DDNM v.s. DiffPIR in noiseless inverse problems.** We report the averaged MAD between their conditional posterior means averaged over all sampling steps and test images.

where $\boldsymbol{r}_t^{*2}(\boldsymbol{x}_t)$ is the optimal posterior variances determined by equation 17 under $s_t = \sqrt{\bar{\alpha}_t}$ and $\sigma_t = \sqrt{\bar{\beta}_t / \bar{\alpha}_t}$, and $\tilde{\beta}_t = \frac{\bar{\beta}_{t-1}}{\bar{\beta}_t} \beta_t$.

Theorem 1 implies that, given the reverse variances $\hat{\boldsymbol{v}}_t^2(\boldsymbol{x}_t)$ predicted by a pre-trained DDPM model at time step $t$, the posterior variances are obtained according to equation 21 by

$$\hat{\boldsymbol{r}}_t^2(\boldsymbol{x}_t) = (\hat{\boldsymbol{v}}_t^2(\boldsymbol{x}_t) - \tilde{\beta}_t) \cdot \left( \frac{\sqrt{\bar{\alpha}_{t-1}} \beta_t}{1 - \bar{\alpha}_t} \right)^{-2}. \tag{22}$$

### 4.3 Optimizing Posterior Variances Without Reverse Variances

We further develop the optimal posterior variances using a pre-trained unconditional diffusion model without providing the reverse variances prediction $\hat{\boldsymbol{v}}_t^2(\boldsymbol{x}_t)$ (e.g., Ho et al. (2020)). In this case, we consider $\boldsymbol{\Sigma}_t(\boldsymbol{x}_t)$ is signal-independent. Suppose $\boldsymbol{\Sigma}_t(\boldsymbol{x}_t) = r_t^2 \boldsymbol{I}$ is an isotropic covariance with a time-dependent standard deviation $r_t$. We directly optimize $r_t$ by forcing the derivative of equation 15 with an optimal $\boldsymbol{\mu}_t$ with regard to $r_t$ to zero. Thus, we obtain that

$$r_t^{*2} = \frac{1}{d} \mathbb{E}_{p_t(\boldsymbol{x}_0, \boldsymbol{x}_t)} [\|\boldsymbol{x}_0 - \mathbb{E}[\boldsymbol{x}_0 | \boldsymbol{x}_t]\|^2]. \tag{23}$$

Note that we only estimate the expected reconstruction error of the MMSE estimator $\mathbb{E}[\boldsymbol{x}_0 | \boldsymbol{x}_t]$. Similar to Bao et al. (2022b), given a pre-trained unconditional diffusion model $D_t$, $r_t^{*2}$ is estimated using Monte Carlo samples as

$$\hat{r}_t^2 = \frac{1}{dM} \sum_{m=1}^M \|\boldsymbol{x}_0^{(m)} - D_t(\boldsymbol{x}_t^{(m)})\|_2^2, \qquad \boldsymbol{x}_0^{(m)}, \boldsymbol{x}_t^{(m)} \sim p_t(\boldsymbol{x}_0, \boldsymbol{x}_t) \tag{24}$$

where $M$ is the number of Monte Carlo samples. In the discrete-time case, $\hat{r}_t^2$ is pre-computed for any $t$ and reused in subsequent computations. In the continuous-time case, we pre-compute $\hat{r}_t^2$ for 1000 discrete time steps, and for any $t$ used in sampling, we use the pre-computed result with the nearest time step to $t$.

## 5 Experiments

We implement our techniques and re-implement prior works on a newly written codebase based on an open source diffusion codebase k-diffusion[4]. This allows us to minimize the effect of different implementation for fair comparisons, and to allow the prior works using more advanced sampler which is not supported by the original codebase. Following (Chung et al. (2023); Wang et al. (2023)), we performed experiments on the FFHQ 256×256 and ImageNet 256×256 datasets to compare different methods. For each dataset, we report the results averaged over 100 validation images. Following (Chung et al. (2023); Zhu et al. (2023)), the pre-trained unconditional diffusion models are from Dhariwal & Nichol (2021) and Chung et al. (2023) for ImageNet and FFHQ, respectively.

The degradation models are specified mostly following (Zhu et al. (2023)): (i) For inpainting, 50 percent of the total pixels are masked out at random. (ii) For Gaussian deblurring and motion deblurring, we use the same setup of the bluring kernels to (Chung et al. (2023)). (iii) For super resolution (SR), we consider bicubic downsampling. All measurements are corrupted by Gaussian noise with $\sigma = 0.05$. For more experimental details and results, see Appendix E.

---

[4] https://github.com/crowsonkb/k-diffusion

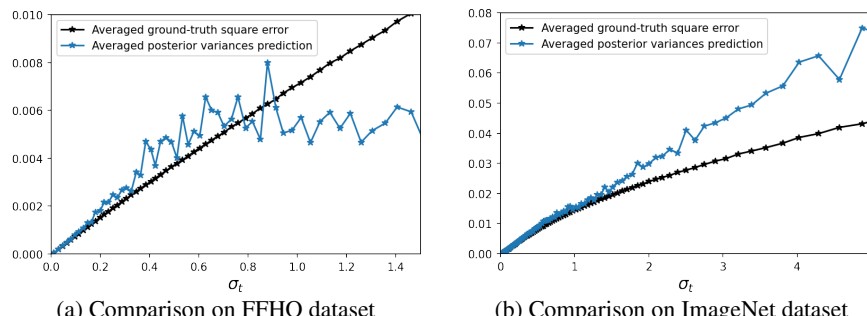

(a) Comparison on FFHQ dataset    (b) Comparison on ImageNet dataset

Figure 1: **Comparing the averaged values of $e$ and $\hat{r}_t^2(x_t)$.** $\hat{r}_t^2(x_t)$ obtained using equation 22 is promising only in low noise level region.

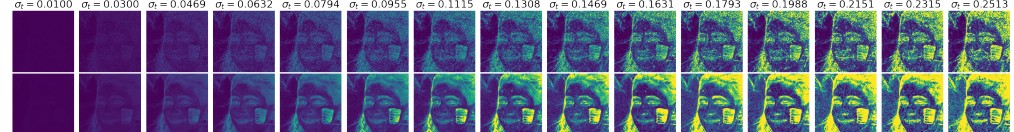

Figure 2: **Visualization of $e$ and $\hat{r}_t^2(x_t)$ of an example image $x_0$ at different $t$ (averaged over RGB channels for better visualization).** Top row: $e$; Bottom row: $\hat{r}_t^2(x_t)$.

| Dataset | Method | Inpaint (Random) | | | Deblur (Gaussian) | | | Deblur (Motion) | | | Super resolution ($4\times$) | | |
|---|---|---|---|---|---|---|---|---|---|---|---|---|---|
| | | PSNR ↑ | LPIPS ↓ | FID ↓ | PSNR ↑ | LPIPS ↓ | FID ↓ | PSNR ↑ | LPIPS ↓ | FID ↓ | PSNR ↑ | LPIPS ↓ | FID ↓ |
| FFHQ | DPS | 32.62 | 0.1323 | 49.46 | 24.93 | 0.3652 | 136.12 | 19.83 | 0.4924 | 212.48 | 27.26 | 0.2054 | 61.36 |
| | ΠGDM | 30.98 | 0.1422 | 49.89 | 27.62 | 0.1910 | 59.93 | 26.69 | 0.2209 | 66.14 | 27.49 | 0.2005 | 61.46 |
| | Analytic (*Ours*) | 33.76 | 0.0845 | 28.83 | 27.71 | 0.1850 | 53.09 | 26.73 | 0.2183 | 64.77 | 27.58 | 0.1968 | 59.83 |
| | Convert (*Ours*) | 33.91 | 0.0794 | 25.90 | 27.74 | 0.1836 | 52.42 | 26.77 | 0.2156 | 62.88 | 27.57 | 0.1962 | 58.37 |
| ImageNet | DPS | 29.55 | 0.1490 | 36.58 | 22.23 | 0.4630 | 173.77 | 19.00 | 0.5554 | 282.21 | 23.82 | 0.3231 | 92.89 |
| | ΠGDM | 27.16 | 0.2328 | 64.96 | 24.28 | 0.3429 | 102.89 | 23.56 | 0.3781 | 113.89 | 24.16 | 0.3552 | 100.36 |
| | Analytic (*Ours*) | 29.16 | 0.1446 | 35.51 | 24.32 | 0.3334 | 93.21 | 23.60 | 0.3668 | 113.39 | 24.25 | 0.3495 | 95.33 |
| | Convert (*Ours*) | 29.48 | 0.1329 | 29.14 | 24.34 | 0.3327 | 95.23 | 23.58 | 0.3656 | 109.61 | 24.22 | 0.3477 | 96.76 |

Table 3: **Quantitative results (PSNR, FID, LPIPS) on FFHQ and ImageNet dataset for Type I guidance.** We use bold and underline for the best and second best, respectively.

## 5.1 Validation of theoretical results

**DDNM as noiseless DiffPIR.** To validate Proposition 2, we re-implement DDNM under DiffPIR codebase. DiffPIR deal with noiseless inverse problems by setting $\sigma$ to relative low value for equation 9 (0.001 in DiffPIR codebase), and we aim to demonstrate that directly use the DDNM solution (equation 11) can produce the similar results under noiseless case. In Table 2, we report the Mean Absolute Difference (MAD, MAD between $x$ and $y$ is defined by $\|x - y\|_1/d$) between their conditional posterior means (equation 9 and equation 11) averaged over all sampling steps and test images to validate Proposition 2. As can be seen, the MAD is negligible in comparison to the data range ($[-1, 1]$). MAD for debluring and super resolution is relatively larger than inpainting, since several approximations are made for computing $A^\dagger$, while for inpainting $A^\dagger$ is exact (see Appendix E.4 for details).

**Posterior variances prediction.** To validate the effectiveness of equation 22 in practice, we compare the ground-truth square errors made by the denoiser $e = (x_0 - D_t(x_t))^2$ with the posterior variances prediction $\hat{r}_t^2(x_t)$. We compare $e$ with $\hat{r}_t^2(x_t)$ because posterior variances $r_t^{*2}(x_t)$ is the MMSE estimator of $e$ given $x_t$ (assuming $D_t(x_t) = \mathbb{E}[x_0|x_t]$), as suggested by equation 17, implying that a good $\hat{r}_t^2(x_t)$ should be a reliable predictor of $e$. Therefore, we may use the comparison between $e$ and $\hat{r}_t^2(x_t)$ as a sanity check to gauge the effectiveness of equation 22 in practice. We plot their averaged values over all pixels and test images in Figure 1. We visualize $e$ and $\hat{r}_t^2(x_t)$ of an example image $x_0$ at different $t$ in Figure 2. As shown in Figure 1, the posterior variances prediction obtained via equation 22 is accurate only in low noise level region. This is reasonable, since the optimal reverse variance $v^{*2}(x_t)$ is bounded by the upper bound $\beta_t$ and the lower bound $\tilde{\beta}_t$, and they are almost equal at high noise level region (Nichol & Dhariwal (2021)). Thus, equation 22 becomes a 0/0 limit that possesses high numerical instability, when $v^{*2}(x_t) - \tilde{\beta}_t \approx 0$.

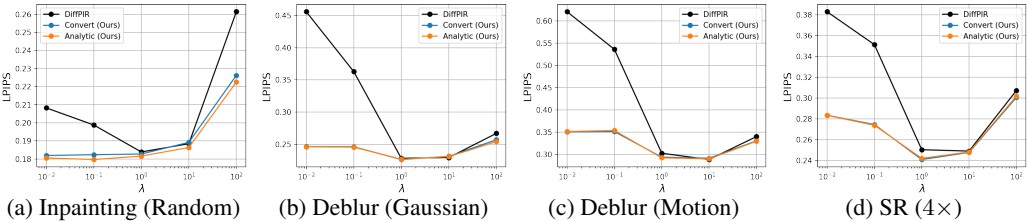

Figure 3: **Quantitative results (LPIPS) on FFHQ dataset for Type II guidance.** We report the LPIPS performance under different $\lambda$.

## 5.2 QUANTITATIVE RESULTS

To evaluate different methods, we use three common metrics: Peak Signal to Noise Ratio (PSNR), Learned Perceptual Image Patch Similarity (LPIPS, Zhang et al. (2018)) and Frechet Inception Distance (FID, Heusel et al. (2017)). For the sampler setup[5], all Type I guidance methods use the same Heun's $2^{nd}$ deterministic sampler suggested in (Karras et al. (2022)) with 50 sampling steps, and all Type II guidance methods use the same Heun's $2^{nd}$ stochastic sampler ($S_{\text{churn}} = 80, S_{\text{tmin}} = 0.05, S_{\text{tmax}} = 1, S_{\text{noise}} = 1.007$, definition see Karras et al. (2022)) with 50 sampling steps since we found that Type II guidance does not perform well using deterministic samplers.

In initial experiments, we found that using optimal posterior covariance for all sampling steps results in poor performance. This may due to using Gaussian approximation $q_t(\boldsymbol{x}_0|\boldsymbol{x}_t)$ to posterior $p_t(\boldsymbol{x}_0|\boldsymbol{x}_t)$ is reasonable only for small noise level (Xiao et al. (2022))[6]. To address this, we only use the optimal posterior covariance at the last few sampling steps for our methods[7], while for high noise level we use $\Pi$GDM and DiffPIR covariances for Type I and Type II guidance, respectively.

Table 3 summarizes the results for Type I guidance. `DPS` and $\Pi$GDM and refer to the posterior covariance type discussed in Section 3.1, and `Analytic` and `Convert` refer to the posterior covariance type obtained using approaches presented in Section 4.3 and Section 4.2, respectively. Note that for Type I guidance, our methods achieve the best results on almost all tasks. Although DPS outperforms us in several cases, we observe that its performance is very unstable (see DPS performance in debluring tasks). For Type II guidance, since we use DiffPIR covariance in high noise level region, the performance of our method and DiffPIR baseline are both influenced by the hyper-parameter $\lambda$. Therefore, we report the performance under different $\lambda$ in Figure 10. We observe that our method is more robust than DiffPIR to the hyper-parameter $\lambda$, which demonstrate the optimality of the posterior covariance obtained via MLE in inverse problems.

## 6 CONCLUSIONS

We show that recent diffusion-based zero-shot methods for inverse problems can be regarded as making isotropic Gaussian approximations to the intractable posterior distributions. Inspired by this fact, we propose to improve existing approaches through the use of optimal posterior covariances obtained via MLE in these Gaussian approximations. We introduce three strategies for obtaining optimal posterior covariances: maximum likelihood pre-training, conversion of off-the-shelf reverse covariance predictions, and analytical estimation using Monte Carlo methods in the absence of reverse covariance predictions. Empirically, our techniques significantly improve the performance of the prior works, or their robustness to hyper-parameters.

---

[5]To understand how to leverage pre-trained DDPM model to perform sampling under perturbation kernels given in Section 2.2, please refer to Appendix D.

[6]To the best of our knowledge, why using Gaussian approximation with heuristic posterior covariance proposed in prior works produce good empirical results still remains largely open.

[7]Empirically, we found that sampling with optimal covariance when $\sigma_t < 0.2$ (12 out of 50 steps), yields highly stable results.

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

## A  PROOFS TO THEORETICAL RESULTS

**Lemma 1** (Tweedie's formula). *If the joint distribution between $\boldsymbol{x}_0, \boldsymbol{x}_t$ is given by $p_t(\boldsymbol{x}_0, \boldsymbol{x}_t) = p(\boldsymbol{x}_0)p_t(\boldsymbol{x}_t|\boldsymbol{x}_0)$ with $p_t(\boldsymbol{x}_t|\boldsymbol{x}_0) = \mathcal{N}(\boldsymbol{x}_t|s_t\boldsymbol{x}_0, s_t^2\sigma_t^2\boldsymbol{I})$, then $\nabla_{\boldsymbol{x}_t}\log p_t(\boldsymbol{x}_t) = \frac{1}{s_t^2\sigma_t^2}(s_t\mathbb{E}[\boldsymbol{x}_0|\boldsymbol{x}_t] - \boldsymbol{x}_t)$.*

*Proof.*

$$\nabla_{\boldsymbol{x}_t}\log p_t(\boldsymbol{x}_t) = \frac{\nabla_{\boldsymbol{x}_t}p_t(\boldsymbol{x}_t)}{p_t(\boldsymbol{x}_t)} \tag{25}$$

$$= \frac{1}{p_t(\boldsymbol{x}_t)}\nabla_{\boldsymbol{x}_t}\int p(\boldsymbol{x}_0)p_t(\boldsymbol{x}_t|\boldsymbol{x}_0)\mathrm{d}\boldsymbol{x}_0 \tag{26}$$

$$= \frac{1}{p_t(\boldsymbol{x}_t)}\int p(\boldsymbol{x}_0)\nabla_{\boldsymbol{x}_t}p_t(\boldsymbol{x}_t|\boldsymbol{x}_0)\mathrm{d}\boldsymbol{x}_0 \tag{27}$$

$$= \frac{1}{p_t(\boldsymbol{x}_t)}\int p(\boldsymbol{x}_0)p_t(\boldsymbol{x}_t|\boldsymbol{x}_0)\nabla_{\boldsymbol{x}_t}\log p_t(\boldsymbol{x}_t|\boldsymbol{x}_0)\mathrm{d}\boldsymbol{x}_0 \tag{28}$$

$$= \int p_t(\boldsymbol{x}_0|\boldsymbol{x}_t)\nabla_{\boldsymbol{x}_t}\log p_t(\boldsymbol{x}_t|\boldsymbol{x}_0)\mathrm{d}\boldsymbol{x}_0 \tag{29}$$

$$= \mathbb{E}_{p_t(\boldsymbol{x}_0|\boldsymbol{x}_t)}[\nabla_{\boldsymbol{x}_t}\log p_t(\boldsymbol{x}_t|\boldsymbol{x}_0)] \tag{30}$$

For Gaussian perturbation kernel $p_t(\boldsymbol{x}_t|\boldsymbol{x}_0) = \mathcal{N}(\boldsymbol{x}_t|s_t\boldsymbol{x}_0, s_t^2\sigma_t^2\boldsymbol{I})$, we have $\nabla_{\boldsymbol{x}_t}\log p_t(\boldsymbol{x}_t|\boldsymbol{x}_0) = \frac{1}{s_t^2\sigma_t^2}(s_t\boldsymbol{x}_0 - \boldsymbol{x}_t)$. Plug it into equation 30, we conclude the proof. □

**Lemma 2** (Conditional Tweedie's formula). *If the joint distribution between $\boldsymbol{x}_0, \boldsymbol{y}, \boldsymbol{x}_t$ is given by $p_t(\boldsymbol{x}_0, \boldsymbol{y}, \boldsymbol{x}_t) = p(\boldsymbol{x}_0)p(\boldsymbol{y}|\boldsymbol{x}_0)p_t(\boldsymbol{x}_t|\boldsymbol{x}_0)$ with $p_t(\boldsymbol{x}_t|\boldsymbol{x}_0) = \mathcal{N}(\boldsymbol{x}_t|s_t\boldsymbol{x}_0, s_t^2\sigma_t^2\boldsymbol{I})$, then $\nabla_{\boldsymbol{x}_t}\log p_t(\boldsymbol{x}_t|\boldsymbol{y}) = \frac{1}{s_t^2\sigma_t^2}(s_t\mathbb{E}[\boldsymbol{x}_0|\boldsymbol{x}_t, \boldsymbol{y}] - \boldsymbol{x}_t)$*

*Proof.*

$$\nabla_{\boldsymbol{x}_t}\log p_t(\boldsymbol{x}_t|\boldsymbol{y}) = \frac{\nabla_{\boldsymbol{x}_t}p_t(\boldsymbol{x}_t|\boldsymbol{y})}{p_t(\boldsymbol{x}_t|\boldsymbol{y})} \tag{31}$$

$$= \frac{1}{p_t(\boldsymbol{x}_t|\boldsymbol{y})}\nabla_{\boldsymbol{x}_t}\int p_t(\boldsymbol{x}_t|\boldsymbol{x}_0, \boldsymbol{y})p(\boldsymbol{x}_0|\boldsymbol{y})\mathrm{d}\boldsymbol{x}_0 \tag{32}$$

$$= \frac{1}{p_t(\boldsymbol{x}_t|\boldsymbol{y})}\nabla_{\boldsymbol{x}_t}\int p_t(\boldsymbol{x}_t|\boldsymbol{x}_0)p(\boldsymbol{x}_0|\boldsymbol{y})\mathrm{d}\boldsymbol{x}_0 \tag{33}$$

$$= \frac{1}{p_t(\boldsymbol{x}_t|\boldsymbol{y})}\int p(\boldsymbol{x}_0|\boldsymbol{y})\nabla_{\boldsymbol{x}_t}p_t(\boldsymbol{x}_t|\boldsymbol{x}_0)\mathrm{d}\boldsymbol{x}_0 \tag{34}$$

$$= \frac{1}{p_t(\boldsymbol{x}_t|\boldsymbol{y})}\int p(\boldsymbol{x}_0|\boldsymbol{y})p_t(\boldsymbol{x}_t|\boldsymbol{x}_0, \boldsymbol{y})\nabla_{\boldsymbol{x}_t}\log p_t(\boldsymbol{x}_t|\boldsymbol{x}_0)\mathrm{d}\boldsymbol{x}_0 \tag{35}$$

$$= \int p_t(\boldsymbol{x}_0|\boldsymbol{x}_t, \boldsymbol{y})\nabla_{\boldsymbol{x}_t}\log p_t(\boldsymbol{x}_t|\boldsymbol{x}_0)\mathrm{d}\boldsymbol{x}_0 \tag{36}$$

$$= \mathbb{E}_{p_t(\boldsymbol{x}_0|\boldsymbol{x}_t, \boldsymbol{y})}[\nabla_{\boldsymbol{x}_t}\log p_t(\boldsymbol{x}_t|\boldsymbol{x}_0)] \tag{37}$$

where equation 33 and equation 35 are due to the conditional independent between $\boldsymbol{x}_t$ and $\boldsymbol{y}$ given $\boldsymbol{x}_0$, such that $p_t(\boldsymbol{x}_t|\boldsymbol{x}_0, \boldsymbol{y}) = p_t(\boldsymbol{x}_t|\boldsymbol{x}_0)$. For Gaussian perturbation kernel $p_t(\boldsymbol{x}_t|\boldsymbol{x}_0) = \mathcal{N}(\boldsymbol{x}_t|s_t\boldsymbol{x}_0, s_t^2\sigma_t^2\boldsymbol{I})$, we have $\nabla_{\boldsymbol{x}_t}\log p_t(\boldsymbol{x}_t|\boldsymbol{x}_0) = \frac{1}{s_t^2\sigma_t^2}(s_t\boldsymbol{x}_0 - \boldsymbol{x}_t)$. Plug it into equation 37, we conclude the proof. □

### A.1  DERIVATION OF THE MARGINAL PRESERVING PROPERTY OF DIFFUSION ODEs

For the sake of completeness, here we prove that the ODEs given in equation 2 and equation 4 respectively maintain the exact same marginals to $p_t(\boldsymbol{x}_t)$ and $p_t(\boldsymbol{x}_t|\boldsymbol{y})$.

*Proof.* By borrowing the results from (Equation 4, Karras et al. (2022)) and setting $s_t = 1, \sigma_t = t$, $\boldsymbol{x}_t$ determined by the following ODE preserves the marginal $p_t(\boldsymbol{x}_t)$ for all $t \in [0, T]$:

$$\mathrm{d}\boldsymbol{x}_t = -t\nabla_{\boldsymbol{x}_t} \log p_t(\boldsymbol{x}_t)\mathrm{d}t, \quad \boldsymbol{x}_T \sim p_T(\boldsymbol{x}_T) \tag{38}$$

Using the posterior mean $\mathbb{E}[\boldsymbol{x}_0|\boldsymbol{x}_t]$ to represent the score $\nabla_{\boldsymbol{x}_t} \log p_t(\boldsymbol{x}_t)$ using Lemma 1, we recover equation 2:

$$\mathrm{d}\boldsymbol{x}_t = \frac{\boldsymbol{x}_t - \mathbb{E}[\boldsymbol{x}_0|\boldsymbol{x}_t]}{t}\mathrm{d}t, \quad \boldsymbol{x}_T \sim p_T(\boldsymbol{x}_T) \tag{39}$$

Likewise, the following ODE preserves the marginal $p_t(\boldsymbol{x}_t|\boldsymbol{y})$ for all $t \in [0, T]$:

$$\mathrm{d}\boldsymbol{x}_t = -t\nabla_{\boldsymbol{x}_t} \log p_t(\boldsymbol{x}_t|\boldsymbol{y})\mathrm{d}t, \quad \boldsymbol{x}_T \sim p_T(\boldsymbol{x}_T|\boldsymbol{y}) \tag{40}$$

By Lemma 2, we recover equation 4:

$$\mathrm{d}\boldsymbol{x}_t = \frac{\boldsymbol{x}_t - \mathbb{E}[\boldsymbol{x}_0|\boldsymbol{x}_t, \boldsymbol{y}]}{t}\mathrm{d}t, \quad \boldsymbol{x}_T \sim p_T(\boldsymbol{x}_T|\boldsymbol{y}) \tag{41}$$

$\square$

## A.2 DERIVATION OF PROPOSITION 1

*Proof.* To relate $\mathbb{E}[\boldsymbol{x}_0|\boldsymbol{x}_t, \boldsymbol{y}]$ to $\mathbb{E}[\boldsymbol{x}_0|\boldsymbol{x}_t]$, we note that

$$\nabla_{\boldsymbol{x}_t} \log p_t(\boldsymbol{x}_t|\boldsymbol{y}) = \nabla_{\boldsymbol{x}_t} \log p_t(\boldsymbol{x}_t) + \nabla_{\boldsymbol{x}_t} \log p_t(\boldsymbol{y}|\boldsymbol{x}_t) \tag{42}$$

Using Lemma 1 and Lemma 2, we have

$$\frac{1}{s_t^2\sigma_t^2}(s_t\mathbb{E}[\boldsymbol{x}_0|\boldsymbol{x}_t, \boldsymbol{y}] - \boldsymbol{x}_t) = \frac{1}{s_t^2\sigma_t^2}(s_t\mathbb{E}[\boldsymbol{x}_0|\boldsymbol{x}_t] - \boldsymbol{x}_t) + \nabla_{\boldsymbol{x}_t} \log p_t(\boldsymbol{y}|\boldsymbol{x}_t) \tag{43}$$

and consequently,

$$\mathbb{E}[\boldsymbol{x}_0|\boldsymbol{x}_t, \boldsymbol{y}] = \mathbb{E}[\boldsymbol{x}_0|\boldsymbol{x}_t] + s_t\sigma_t^2\nabla_{\boldsymbol{x}_t} \log p_t(\boldsymbol{y}|\boldsymbol{x}_t) \tag{44}$$

$\square$

## A.3 DERIVATION OF PROPOSITION 2

*Proof.* When $\sigma \to 0$, finding the minimizer of equation 10 is equivalent to solving the following hard-constraint optimization problem :

$$\min_{\hat{\boldsymbol{x}}_0}\|\hat{\boldsymbol{x}}_0 - \mathbb{E}[\boldsymbol{x}_0|\boldsymbol{x}_t]\|_2^2 \quad \text{s.t.} \quad \boldsymbol{y} = \boldsymbol{A}\hat{\boldsymbol{x}}_0 \tag{45}$$

We define the Lagrangian $\mathcal{L}(\hat{\boldsymbol{x}}_0, \lambda) = \frac{1}{2}\|\hat{\boldsymbol{x}}_0 - \mathbb{E}[\boldsymbol{x}_0|\boldsymbol{x}_t]\|_2^2 + \lambda^T(\boldsymbol{y} - \boldsymbol{A}\hat{\boldsymbol{x}}_0)$, where $\lambda$ is the Lagrangian multiplier. By the optimality condition, we have

$$\nabla_{\hat{\boldsymbol{x}}_0}\mathcal{L} = \hat{\boldsymbol{x}}_0 - \mathbb{E}[\boldsymbol{x}_0|\boldsymbol{x}_t] - \boldsymbol{A}^T\lambda = \boldsymbol{0} \tag{46}$$

$$\nabla_{\lambda}\mathcal{L} = \boldsymbol{y} - \boldsymbol{A}\hat{\boldsymbol{x}}_0 = \boldsymbol{0} \tag{47}$$

Multiplying $\boldsymbol{A}$ to equation 46 and combining the condition of equation 47 gives:

$$\boldsymbol{A}(\hat{\boldsymbol{x}}_0 - \mathbb{E}[\boldsymbol{x}_0|\boldsymbol{x}_t] - \boldsymbol{A}^T\lambda) = \boldsymbol{0} \tag{48}$$

$$\Rightarrow \boldsymbol{y} - \boldsymbol{A}\mathbb{E}[\boldsymbol{x}_0|\boldsymbol{x}_t] - \boldsymbol{A}\boldsymbol{A}^T\lambda = \boldsymbol{0} \tag{49}$$

$$\Rightarrow \boldsymbol{A}\boldsymbol{A}^T\lambda = \boldsymbol{y} - \boldsymbol{A}\mathbb{E}[\boldsymbol{x}_0|\boldsymbol{x}_t] \tag{50}$$

Multiplying $\boldsymbol{A}^\dagger$ to equation 50 and leveraging the property $\boldsymbol{A}^\dagger\boldsymbol{A}\boldsymbol{A}^T = \boldsymbol{A}^T$, we have

$$\boldsymbol{A}^T\lambda = \boldsymbol{A}^\dagger\boldsymbol{y} - \boldsymbol{A}^\dagger\boldsymbol{A}\mathbb{E}[\boldsymbol{x}_0|\boldsymbol{x}_t] \tag{51}$$

and consequently,

$$\hat{\boldsymbol{x}}_0 = \mathbb{E}[\boldsymbol{x}_0|\boldsymbol{x}_t] + \boldsymbol{A}^\dagger\boldsymbol{y} - \boldsymbol{A}^\dagger\boldsymbol{A}\mathbb{E}[\boldsymbol{x}_0|\boldsymbol{x}_t] \tag{52}$$

$$= \boldsymbol{A}^\dagger\boldsymbol{y} + (\boldsymbol{I} - \boldsymbol{A}^\dagger\boldsymbol{A})\mathbb{E}[\boldsymbol{x}_0|\boldsymbol{x}_t] \tag{53}$$

$\square$

A.4 DERIVATION OF THEOREM 1

*Proof.* We follow Bao et al. (2022b) to derive the relationship between the optimal reverse variances $\boldsymbol{v}_t^2(\boldsymbol{x}_t)$ and optimal posterior variances $\boldsymbol{r}_t^2(\boldsymbol{x}_t)$ based on more general non-Markov forward process introduced by Song et al. (2021a). To find the optimal solution to equation 19, we present a much simpler proof than Bao et al. (2022b) using functional derivatives motivated by Rezende & Viola (2018). Given a noise schedule $\{\beta_t\}_{t=1}^T$ and $\alpha_t = 1 - \beta_t$, the forward process $q(\boldsymbol{x}_{1:T}|\boldsymbol{x}_0)$ is defined as

$$q(\boldsymbol{x}_{1:T}|\boldsymbol{x}_0) = q(\boldsymbol{x}_T|\boldsymbol{x}_0)\prod_{t=2}^T q(\boldsymbol{x}_{t-1}|\boldsymbol{x}_t, \boldsymbol{x}_0) \tag{54}$$

$$q(\boldsymbol{x}_{t-1}|\boldsymbol{x}_t, \boldsymbol{x}_0) = \mathcal{N}(\boldsymbol{x}_{t-1}|\tilde{\boldsymbol{\mu}}_t(\boldsymbol{x}_t, \boldsymbol{x}_0), \lambda_t^2\boldsymbol{I}) \tag{55}$$

$$\tilde{\boldsymbol{\mu}}_t(\boldsymbol{x}_t, \boldsymbol{x}_0) = \sqrt{\bar{\alpha}_{t-1}}\boldsymbol{x}_0 + \sqrt{\bar{\beta}_{t-1} - \lambda_t^2} \cdot \frac{\boldsymbol{x}_t - \sqrt{\bar{\alpha}_t}\boldsymbol{x}_0}{\sqrt{\bar{\beta}_t}} \tag{56}$$

where $\bar{\alpha}_t = \prod_{i=1}^t \alpha_i$ and $\bar{\beta}_t = 1 - \bar{\alpha}_t$. Song et al. (2021a) show that for arbitrary choice of $\lambda_t$, the marginal distributions $q(\boldsymbol{x}_t|\boldsymbol{x}_0)$ maintain $q(\boldsymbol{x}_t|\boldsymbol{x}_0) = \mathcal{N}(\boldsymbol{x}_t|\sqrt{\bar{\alpha}_t}\boldsymbol{x}_0, \bar{\beta}_t\boldsymbol{I})$. DDPM forward process is a special case when $\lambda_t^2 = \tilde{\beta}_t$ with $\tilde{\beta}_t = \frac{\bar{\beta}_{t-1}}{\bar{\beta}_t}\beta_t$, which is used in Nichol & Dhariwal (2021) for pre-training DDPM model.

To fit the data distribution $q(\boldsymbol{x}_0)$, we define the reverse process, given by

$$p(\boldsymbol{x}_{0:T}) = p(\boldsymbol{x}_T)\prod_{t=1}^T p(\boldsymbol{x}_{t-1}|\boldsymbol{x}_t), \quad p(\boldsymbol{x}_{t-1}|\boldsymbol{x}_t) = \mathcal{N}(\boldsymbol{x}_{t-1}|\boldsymbol{m}_t(\boldsymbol{x}_t), \text{diag}[\boldsymbol{v}_t^2(\boldsymbol{x}_t)]) \tag{57}$$

To train $p(\boldsymbol{x}_0)$ we minimize the KL divergence between the forward and the reverse process:

$$\min_p D_{KL}(q(\boldsymbol{x}_{0:T})||p(\boldsymbol{x}_{0:T})), \quad q(\boldsymbol{x}_{0:T}) = q(\boldsymbol{x}_0)q(\boldsymbol{x}_{1:T}|\boldsymbol{x}_0) \tag{58}$$

which is equivalent to minimizing the variational bound $\mathbb{E}_q[L_{\text{vb}}]$ on negative log-likelihood of data distribution $q(\boldsymbol{x}_0)$ with $L_{\text{vb}}$ given as follows:

$$L_{\text{vb}} = L_0 + L_1 + ... + L_T \tag{59}$$

$$L_0 = -\log p(\boldsymbol{x}_0|\boldsymbol{x}_1) \tag{60}$$

$$L_{t-1} = D_{KL}(q(\boldsymbol{x}_{t-1}|\boldsymbol{x}_0, \boldsymbol{x}_t)||p(\boldsymbol{x}_{t-1}|\boldsymbol{x}_t)) \tag{61}$$

$$L_T = D_{KL}(q(\boldsymbol{x}_T|\boldsymbol{x}_0)||p(\boldsymbol{x}_T)) \tag{62}$$

For $t \in [2, T]$, $L_{t-1}$ are KL divergences between two Gaussians, which possess analytical forms:

$$L_{t-1} \equiv \log\frac{|\text{diag}[\boldsymbol{v}_t^2(\boldsymbol{x}_t)]|}{|\lambda_t^2\boldsymbol{I}|} + \|\tilde{\boldsymbol{\mu}}_t(\boldsymbol{x}_t, \boldsymbol{x}_0) - \boldsymbol{m}(\boldsymbol{x}_t)\|_{\text{diag}[\boldsymbol{v}_t^2(\boldsymbol{x}_t)]^{-1}}^2 + \text{tr}[\lambda_t^2\text{diag}[\boldsymbol{v}_t^2(\boldsymbol{x}_t)]^{-1}] \tag{63}$$

$$\equiv \sum_{i=1}^d \log \boldsymbol{v}_t^2(\boldsymbol{x}_t)_i + \frac{(\tilde{\boldsymbol{\mu}}_t(\boldsymbol{x}_t, \boldsymbol{x}_0)_i - \boldsymbol{m}_t(\boldsymbol{x}_t)_i)^2}{\boldsymbol{v}_t^2(\boldsymbol{x}_t)_i} + \frac{\lambda_t^2}{\boldsymbol{v}_t^2(\boldsymbol{x}_t)_i} \tag{64}$$

where "$\equiv$" denotes "equals up to a constant and a scaling factor" and $i$ indexes the elements of an vector.

Note that minimizing $\mathbb{E}_q[L_{\text{vb}}]$ can be decomposed into $T$ independent optimization sub-problems:

$$\min_{\boldsymbol{m}_t, \boldsymbol{v}_t} \mathbb{E}_{q(\boldsymbol{x}_0, \boldsymbol{x}_t)}[L_{t-1}], \quad t \in [1, T] \tag{65}$$

The optimal $\boldsymbol{m}_t$ and $\boldsymbol{v}_t$ can be found by taking the functional derivatives of $\mathbb{E}_{q(\boldsymbol{x}_0, \boldsymbol{x}_t)}[L_{t-1}]$ w.r.t $\boldsymbol{m}_t$ and $\boldsymbol{v}_t^2$ then set to zero:

$$\frac{\delta\mathbb{E}_{q(\boldsymbol{x}_0, \boldsymbol{x}_t)}[L_{t-1}]}{\delta\boldsymbol{m}_t(\boldsymbol{x}_t)_i} \equiv \mathbb{E}_{q(\boldsymbol{x}_0)}[q(\boldsymbol{x}_t|\boldsymbol{x}_0)\frac{\boldsymbol{m}_t(\boldsymbol{x}_t)_i - \tilde{\boldsymbol{\mu}}_t(\boldsymbol{x}_t, \boldsymbol{x}_0)_i}{\boldsymbol{v}_t^2(\boldsymbol{x}_t)_i}] = 0 \tag{66}$$

$$\frac{\delta\mathbb{E}_{q(\boldsymbol{x}_0, \boldsymbol{x}_t)}[L_{t-1}]}{\delta\boldsymbol{v}_t^2(\boldsymbol{x}_t)_i} \equiv \mathbb{E}_{q(\boldsymbol{x}_0)}[q(\boldsymbol{x}_t|\boldsymbol{x}_0)(\frac{1}{\boldsymbol{v}_t^2(\boldsymbol{x}_t)_i} - \frac{(\tilde{\boldsymbol{\mu}}_t(\boldsymbol{x}_t, \boldsymbol{x}_0)_i - \boldsymbol{m}(\boldsymbol{x}_t)_i)^2}{(\boldsymbol{v}_t^2(\boldsymbol{x}_t)_i)^2} - \frac{\lambda_t^2}{(\boldsymbol{v}_t^2(\boldsymbol{x}_t)_i)^2})] = 0 \tag{67}$$

We can solve for optimal $\boldsymbol{m}_t(\boldsymbol{x}_t)_i$ by rearranging equation 66:

$$\mathbb{E}_{q(\boldsymbol{x}_0)}[q(\boldsymbol{x}_t|\boldsymbol{x}_0)]\boldsymbol{m}_t(\boldsymbol{x}_t)_i = \mathbb{E}_{q(\boldsymbol{x}_0)}[q(\boldsymbol{x}_t|\boldsymbol{x}_0)\tilde{\boldsymbol{\mu}}_t(\boldsymbol{x}_t, \boldsymbol{x}_0)_i] \tag{68}$$

$$\Rightarrow q(\boldsymbol{x}_t)\boldsymbol{m}_t(\boldsymbol{x}_t)_i = \int q(\boldsymbol{x}_0)q(\boldsymbol{x}_t|\boldsymbol{x}_0)\tilde{\boldsymbol{\mu}}_t(\boldsymbol{x}_t, \boldsymbol{x}_0)_i \mathrm{d}\boldsymbol{x}_0 \tag{69}$$

$$\Rightarrow \boldsymbol{m}_t(\boldsymbol{x}_t)_i = \int q(\boldsymbol{x}_0|\boldsymbol{x}_t)\tilde{\boldsymbol{\mu}}_t(\boldsymbol{x}_t, \boldsymbol{x}_0)_i \mathrm{d}\boldsymbol{x}_0 \tag{70}$$

$$\Rightarrow \boldsymbol{m}_t(\boldsymbol{x}_t)_i = \tilde{\boldsymbol{\mu}}_t(\boldsymbol{x}_t, \mathbb{E}[\boldsymbol{x}_0|\boldsymbol{x}_t])_i \tag{71}$$

where equation 71 is due to the linearity of $\tilde{\boldsymbol{\mu}}_t$ w.r.t $\boldsymbol{x}_0$.

Likewise, rearranging equation 67 gives

$$\boldsymbol{v}_t^2(\boldsymbol{x}_t)_i = \lambda_t^2 + \mathbb{E}_{q(\boldsymbol{x}_0|\boldsymbol{x}_t)}[(\tilde{\boldsymbol{\mu}}_t(\boldsymbol{x}_t, \boldsymbol{x}_0)_i - \boldsymbol{m}(\boldsymbol{x}_t)_i)^2] \tag{72}$$

By plugging the optimal $\boldsymbol{m}_t(\boldsymbol{x}_t)_i$ determined by equation 71 into equation 72 and dropping the element index $i$, we conclude the proof:

$$\boldsymbol{v}_t^2(\boldsymbol{x}_t) = \lambda_t^2 + \mathbb{E}_{q(\boldsymbol{x}_0|\boldsymbol{x}_t)}[(\tilde{\boldsymbol{\mu}}_t(\boldsymbol{x}_t, \boldsymbol{x}_0) - \tilde{\boldsymbol{\mu}}_t(\boldsymbol{x}_t, \mathbb{E}[\boldsymbol{x}_0|\boldsymbol{x}_t]))^2] \tag{73}$$

$$= \lambda_t^2 + \mathbb{E}_{q(\boldsymbol{x}_0|\boldsymbol{x}_t)}[(\tilde{\boldsymbol{\mu}}_t(\boldsymbol{0}, \boldsymbol{x}_0 - \mathbb{E}[\boldsymbol{x}_0|\boldsymbol{x}_t]))^2] \tag{74}$$

$$= \lambda_t^2 + (\sqrt{\bar{\alpha}_{t-1}} - \sqrt{\bar{\beta}_{t-1} - \lambda_t^2}\sqrt{\frac{\bar{\alpha}_t}{\bar{\beta}_t}})^2 \cdot \mathbb{E}_{q(\boldsymbol{x}_0|\boldsymbol{x}_t)}[(\boldsymbol{x}_0 - \mathbb{E}[\boldsymbol{x}_0|\boldsymbol{x}_t])^2] \tag{75}$$

$$= \lambda_t^2 + (\sqrt{\bar{\alpha}_{t-1}} - \sqrt{\bar{\beta}_{t-1} - \lambda_t^2}\sqrt{\frac{\bar{\alpha}_t}{\bar{\beta}_t}})^2 \cdot \boldsymbol{r}_t^2(\boldsymbol{x}_t) \tag{76}$$

$$\square$$

We are often given a pre-trained DDPM model with learned reverse variances. The following Corollary of Theorem 1 gives a simplified relationship between optimal posterior variances and optimal reverse variances under DDPM case:

**Corollary 1.** *For the DDPM forward process $\lambda_t^2 = \tilde{\beta}_t$, the optimal posterior variances $\boldsymbol{r}_t^{*2}(\boldsymbol{x}_t)$ and optimal reverse variances $\boldsymbol{v}_t^{*2}(\boldsymbol{x}_t)$ are related by*

$$\boldsymbol{v}_t^{*2}(\boldsymbol{x}_t) = \tilde{\beta}_t + (\frac{\sqrt{\bar{\alpha}_{t-1}}\beta_t}{1 - \bar{\alpha}_t})^2 \cdot \boldsymbol{r}_t^{*2}(\boldsymbol{x}_t) \tag{77}$$

**Remark 1.** *From Theorem 1, we also know that to perform optimal ancestral sampling, we only need to provide the MMSE estimator $\mathbb{E}[\boldsymbol{x}_0|\boldsymbol{x}_t]$ to compute the reverse mean $\boldsymbol{m}_t^*(\boldsymbol{x}_t)$ and the posterior variances $\boldsymbol{r}_t^{*2}(\boldsymbol{x}_t)$ to compute the reverse variances $\boldsymbol{v}_t^{*2}(\boldsymbol{x}_t)$, which can be both obtained from the proposed maximum likelihood pre-training (equation 15). This may provide an alternative way for pre-trainning DDPM model that differs from (Nichol & Dhariwal (2021)).*

## A.5 DERIVATION OF PROPOSITION 3

*Proof.* Deriving the optimal solution to equation 15 under the diagonal posterior covariance case, i.e., $\boldsymbol{\Sigma}_t(\boldsymbol{x}_t) = \mathrm{diag}(\boldsymbol{r}_t^2(\boldsymbol{x}_t))$, is similar to Appendix A.4. Note that we seek for point-wise maximizer of equation 15, i.e., find the optimum of

$$\mathbb{E}_{p_t(\boldsymbol{x}_0, \boldsymbol{x}_t)}[\log q_t(\boldsymbol{x}_0|\boldsymbol{x}_t)] \equiv \mathbb{E}_{p_t(\boldsymbol{x}_0, \boldsymbol{x}_t)}[\sum_{i=1}^{d} \frac{1}{\boldsymbol{r}_t^2(\boldsymbol{x}_t)_i}(\boldsymbol{x}_{0i} - \boldsymbol{\mu}_t(\boldsymbol{x}_t)_i)^2 + \log \boldsymbol{r}_t^2(\boldsymbol{x}_t)_i] \tag{78}$$

For any $t$, taking the functional derivatives of $\mathbb{E}_{p_t(\boldsymbol{x}_0, \boldsymbol{x}_t)}[\log q_t(\boldsymbol{x}_0|\boldsymbol{x}_t)]$ w.r.t $\boldsymbol{\mu}_t(\boldsymbol{x}_t)_i$ and $\boldsymbol{r}_t^2(\boldsymbol{x}_t)_i$ and then set to zero, we obtain the optimality conditions:

$$\mathbb{E}_{p(\boldsymbol{x}_0)}[p(\boldsymbol{x}_t|\boldsymbol{x}_0)(\boldsymbol{\mu}_t(\boldsymbol{x}_t)_i - \boldsymbol{x}_{0i})] = 0 \tag{79}$$

$$\mathbb{E}_{p(\boldsymbol{x}_0)}[p(\boldsymbol{x}_t|\boldsymbol{x}_0)(-\frac{1}{(\boldsymbol{r}_t^2(\boldsymbol{x}_t)_i)^2}(\boldsymbol{x}_{0i} - \boldsymbol{\mu}_t(\boldsymbol{x}_t)_i)^2 + \frac{1}{\boldsymbol{r}_t^2(\boldsymbol{x}_t)_i}] = 0 \tag{80}$$

Combining the optimality conditions given by equation 79 and equation 80, we conclude the proof.

$$\square$$

# B    CLOSED-FORM SOLUTIONS FOR IMPLEMENTING GUIDANCES

In this section, we provide important closed-form results for implementing efficient Type I and Type II guidance under isotropic posterior covariance case: $\Sigma_t(\boldsymbol{x}_t) = r_t^2(\boldsymbol{x}_t)\boldsymbol{I}$. Before we delve into the closed-form results, we first give some important notations. We define the downsampling operator given sampling position $\boldsymbol{m} \in \{0,1\}^{d \times 1}$ as $\boldsymbol{D_m} \in \{0,1\}^{\|m\|_0 \times d}$, which selects rows of a given matrix that corresponds to one in $\boldsymbol{m}$ and when performing left multiplication. We use $\boldsymbol{D}_{\downarrow s}$ to denote the standard $s$-folds downsampling operator, which is equivalent to $\boldsymbol{D_m}$ when ones in $\boldsymbol{m}$ are spaced evenly. For image signal, it selects the upper-left pixel for each distinct $s \times s$ patch (Zhang et al. (2020)) when performing left multiplication to the *vectorized* image. We use $\boldsymbol{D}_{\Downarrow s}$ to denote the distinct block downsampler, i.e., averaging $s$ length $d/s$ distinct blocks of a vector. For image signal, it averaging distinct $d/s \times d/s$ blocks (Zhang et al. (2020)) when performing left multiplication to the *vectorized* image. We denote the Fourier transform matrix for $d$-dimensional signal as $\boldsymbol{F}$, Fourier transform matrix for $d/s$-dimensional signal as $\boldsymbol{F}_{\downarrow s}$, the Fourier transform of a vector $\boldsymbol{v}$ as $\hat{\boldsymbol{v}}$, and the complex conjugate of a complex vector $\boldsymbol{v}$ as $\bar{\boldsymbol{v}}$. We use $\odot$ to denote element-wise multiplication, and the divisions used below are also element-wise.

**Lemma 3.** *Performing $s$-fold standard downsampling in spacial domain is equivalent to performing $s$-fold block downsampling in frequency domain: $\boldsymbol{D}_{\Downarrow s} = \boldsymbol{F}_{\downarrow s}\boldsymbol{D}_{\downarrow s}\boldsymbol{F}^{-1}$.*

*Proof.* Considering an arbitrary $d$-dimensional signal in frequency domain $\hat{\boldsymbol{x}}[k], \ k = 0, 2, .., d-1$. Multiplying $\boldsymbol{F}_{\downarrow s}\boldsymbol{D}_{\downarrow s}\boldsymbol{F}^{-1}$ to $\hat{\boldsymbol{x}}$ is equivalent to letting $\hat{\boldsymbol{x}}$ go through the following linear system and obtain the output $\hat{\boldsymbol{y}}$:

$$\boldsymbol{x}[n] = \frac{1}{d}\sum_{k=0}^{d-1}\hat{\boldsymbol{x}}[k]e^{j\frac{2\pi}{d}kn} \tag{81}$$

$$\boldsymbol{x}_{\downarrow s}[n] = \boldsymbol{x}[ns] \tag{82}$$

$$\hat{\boldsymbol{y}}[k] = \sum_{n=0}^{d/s-1}\boldsymbol{x}_{\downarrow s}[n]e^{-j\frac{2\pi}{d/s}kn} \tag{83}$$

We now use $\hat{\boldsymbol{x}}$ to represent $\hat{\boldsymbol{y}}$:

$$\hat{\boldsymbol{y}}[k] = \sum_{n=0}^{d/s-1}\frac{1}{d}\sum_{k'=0}^{d-1}\hat{\boldsymbol{x}}[k']e^{j\frac{2\pi}{d}k'ns}e^{-j\frac{2\pi}{d/s}kn} \tag{84}$$

$$= \sum_{n=0}^{d/s-1}\frac{1}{d}e^{-j\frac{2\pi}{d/s}kn}\sum_{k'=0}^{d-1}\hat{\boldsymbol{x}}[k']e^{j\frac{2\pi}{d/s}k'n} \tag{85}$$

$$= \sum_{n=0}^{d/s-1}\frac{1}{d}e^{-j\frac{2\pi}{d/s}kn}\left(\sum_{k'=0}^{d/s-1} + \sum_{k'=d/s}^{2d/s-1} + \sum_{k'=2d/s}^{3d/s-1} + ... + \sum_{k'=(s-1)d/s}^{sd/s-1}\right)\hat{\boldsymbol{x}}[k']e^{j\frac{2\pi}{d/s}k'n} \tag{86}$$

$$= \sum_{n=0}^{d/s-1}\frac{1}{d}e^{-j\frac{2\pi}{d/s}kn}\sum_{k'=0}^{d/s-1}(\hat{\boldsymbol{x}}[k'] + \hat{\boldsymbol{x}}[k'+d/s] + ... + \hat{\boldsymbol{x}}[k'+(s-1)d/s])e^{j\frac{2\pi}{d/s}k'n} \tag{87}$$

$$= \sum_{n=0}^{d/s-1}e^{-j\frac{2\pi}{d/s}kn}\frac{1}{d/s}\sum_{k'=0}^{d/s-1}\frac{\hat{\boldsymbol{x}}[k'] + \hat{\boldsymbol{x}}[k'+d/s] + ... + \hat{\boldsymbol{x}}[k'+(s-1)d/s]}{s}e^{j\frac{2\pi}{d/s}k'n} \tag{88}$$

$$= \frac{\hat{\boldsymbol{x}}[k] + \hat{\boldsymbol{x}}[k+d/s] + ... + \hat{\boldsymbol{x}}[k+(s-1)d/s]}{s} \tag{89}$$

$$\tag{90}$$

where equation 87 is because $e^{j\frac{2\pi}{d/s}k'n}$ has period of $d/s$ in $k'$ and equation 89 is because $d/s$-dimensional inverse Fourier transform and Fourier transform are canceled out. From equation 89 we have $\hat{\boldsymbol{y}} = \boldsymbol{D}_{\Downarrow s}\hat{\boldsymbol{x}}$. So $\boldsymbol{D}_{\Downarrow s}\hat{\boldsymbol{x}} = \boldsymbol{F}_{\downarrow s}\boldsymbol{D}_{\downarrow s}\boldsymbol{F}^{-1}\hat{\boldsymbol{x}}$ for any $\hat{\boldsymbol{x}}$, and consequently, $\boldsymbol{D}_{\Downarrow s} = \boldsymbol{F}_{\downarrow s}\boldsymbol{D}_{\downarrow s}\boldsymbol{F}^{-1}$. $\square$

## B.1 CLOSED-FORM SOLUTIONS TO VECTOR-JACOBIAN PRODUCTS IN TYPE I GUIDANCE

When using Type I guidance, we are required to approximate the likelihood score, which can be computed via vector-Jacobian product (Song et al. (2023)):

$$\nabla_{\boldsymbol{x}_t} \log p(\boldsymbol{y}|\boldsymbol{x}_t) \approx (\boldsymbol{v}^T \frac{\partial \mathbb{E}[\boldsymbol{x}_0|\boldsymbol{x}_t]}{\partial \boldsymbol{x}_t})^T \tag{91}$$

where $\boldsymbol{v} = \boldsymbol{A}^T(\sigma^2\boldsymbol{I} + \boldsymbol{A}\boldsymbol{\Sigma}_t(\boldsymbol{x}_t)\boldsymbol{A}^T)^{-1}(\boldsymbol{y} - \boldsymbol{A}\mathbb{E}[\boldsymbol{x}_0|\boldsymbol{x}_t])$. Below we provide closed-form solution to $\boldsymbol{v}$ under three common degradation operators: i) inpainting, ii) debluring, and iii) super resolution, then we can approximate the conditional posterior mean $\mathbb{E}[\boldsymbol{x}_0|\boldsymbol{x}_t, \boldsymbol{y}]$ as in illustrated in Algorithm 1.

---

**Algorithm 1:** PyTorch-style pseudocode for implementing Type I guidance

---

```
def type_I_guidance(x, y, A, sigma):
    # ------------------------------------------------------------
    # Input:
    #     x:   x_t
    #     y:   y
    #     A: include information of A and measurement noise std σ
    #     sigma:   σ_t
    # Output:
    #     cond_x0_mean:  Approximated E[x_0|x_t,y] based on Proposition 1
    # ------------------------------------------------------------
    x = x.requires_grad_()
    x0_mean, x0_var = uncond_x0_mean_var(x, sigma) # Obtain mean and covariance of q_t(x_0|x_t)
    v = v_solution(A, y, x0_mean, x0_var) # Compute v using closed-form solutions
    likelihood_score = torch.autograd.grad((v.detach() * x0_mean).sum(), x)[0]
    cond_x0_mean = x0_mean + sigma.pow(2) * likelihood_score # Proposition 1
    return cond_x0_mean.clip(-1, 1)# Clip to the data range [-1, 1]
```

---

**Inpainting.** The observation model for image inpainting can be expressed as:

$$\boldsymbol{y} = \underbrace{\boldsymbol{D_m}}_{\boldsymbol{A}} \boldsymbol{x}_0 + \boldsymbol{n} \tag{92}$$

and the closed-form solution to $\boldsymbol{v}$ in image inpainting is given by the following:

$$\boldsymbol{v} = \frac{\tilde{\boldsymbol{y}} - \boldsymbol{m} \odot \mathbb{E}[\boldsymbol{x}_0|\boldsymbol{x}_t]}{\sigma^2 + r_t^2(\boldsymbol{x}_t)} \tag{93}$$

where $\tilde{\boldsymbol{y}} = \boldsymbol{D_m}^T\boldsymbol{y} = \boldsymbol{m} \odot (\boldsymbol{x}_0 + \tilde{\boldsymbol{n}})$, $\tilde{\boldsymbol{n}} \sim \mathcal{N}(\boldsymbol{0}, \boldsymbol{I})$ is the zero-filling measurements that fills the masked region with zeros and processes the exact same size to $\boldsymbol{x}_0$. In practice, the measurements in inpainting are usually stored in the form of $\tilde{\boldsymbol{y}}$, while $\boldsymbol{y}$ used here is for mathematical convenient.

*Proof.*

$$\boldsymbol{v} = \boldsymbol{D_m}^T(\sigma^2\boldsymbol{I} + \boldsymbol{D_m}r_t^2(\boldsymbol{x}_t)\boldsymbol{I}\boldsymbol{D_m}^T)^{-1}(\boldsymbol{y} - \boldsymbol{D_m}\mathbb{E}[\boldsymbol{x}_0|\boldsymbol{x}_t]) \tag{94}$$

$$= \boldsymbol{D_m}^T(\sigma^2\boldsymbol{I} + r_t^2(\boldsymbol{x}_t)\boldsymbol{D_m}\boldsymbol{D_m}^T)^{-1}(\boldsymbol{y} - \boldsymbol{D_m}\mathbb{E}[\boldsymbol{x}_0|\boldsymbol{x}_t]) \tag{95}$$

$$= \boldsymbol{D_m}^T((\sigma^2 + r_t^2(\boldsymbol{x}_t))\boldsymbol{I})^{-1}(\boldsymbol{y} - \boldsymbol{D_m}\mathbb{E}[\boldsymbol{x}_0|\boldsymbol{x}_t]) \tag{96}$$

$$= \frac{\boldsymbol{D_m}^T(\boldsymbol{y} - \boldsymbol{D_m}\mathbb{E}[\boldsymbol{x}_0|\boldsymbol{x}_t])}{\sigma^2 + r_t^2(\boldsymbol{x}_t)} \tag{97}$$

$$= \frac{\tilde{\boldsymbol{y}} - \boldsymbol{m} \odot \mathbb{E}[\boldsymbol{x}_0|\boldsymbol{x}_t]}{\sigma^2 + r_t^2(\boldsymbol{x}_t)} \tag{98}$$

where equation 96 and equation 98 are because $\boldsymbol{D_m}\boldsymbol{D_m}^T = \boldsymbol{I}$ and $\boldsymbol{D_m}^T\boldsymbol{D_m} = \text{diag}(\boldsymbol{m})$. □

**Debluring.** The observation model for image debluring can be expressed as:

$$\boldsymbol{y} = \boldsymbol{x}_0 * \boldsymbol{k} + \boldsymbol{n} \tag{99}$$

where $\boldsymbol{k}$ is the blurring kernel and $*$ is convolution operator. By assuming $*$ is a circular convolution operator, we can convert equation 99 to the canonical form $\boldsymbol{y} = \boldsymbol{A}\boldsymbol{x}_0 + \boldsymbol{n}$ by leveraging the convolution property of Fourier transform:

$$\boldsymbol{y} = \underbrace{\boldsymbol{F}^{-1}\text{diag}(\hat{\boldsymbol{k}})\boldsymbol{F}}_{\boldsymbol{A}} \boldsymbol{x}_0 + \boldsymbol{n} \tag{100}$$

and the closed-form solution to $\boldsymbol{v}$ in image debluring is given by the following:

$$\boldsymbol{v} = \boldsymbol{F}^{-1}(\bar{\hat{\boldsymbol{k}}} \odot \frac{\boldsymbol{F}(\boldsymbol{y} - \boldsymbol{A}\mathbb{E}[\boldsymbol{x}_0|\boldsymbol{x}_t])}{\sigma^2 + r_t^2(\boldsymbol{x}_t)\bar{\hat{\boldsymbol{k}}} \odot \hat{\boldsymbol{k}}}) \tag{101}$$

*Proof.* Since $\boldsymbol{A}$ is a real matrix, we have $\boldsymbol{A}^T = \boldsymbol{A}^H = \boldsymbol{F}^{-1}\mathrm{diag}(\bar{\hat{\boldsymbol{k}}})\boldsymbol{F}$, then

$$\boldsymbol{v} = \boldsymbol{F}^{-1}\mathrm{diag}(\bar{\hat{\boldsymbol{k}}})\boldsymbol{F}(\sigma^2\boldsymbol{I} + \boldsymbol{F}^{-1}\mathrm{diag}(\hat{\boldsymbol{k}})\boldsymbol{F}r_t^2(\boldsymbol{x}_t)\boldsymbol{I}\boldsymbol{F}^{-1}\mathrm{diag}(\bar{\hat{\boldsymbol{k}}})\boldsymbol{F})^{-1}(\boldsymbol{y} - \boldsymbol{A}\mathbb{E}[\boldsymbol{x}_0|\boldsymbol{x}_t]) \tag{102}$$

$$= \boldsymbol{F}^{-1}\mathrm{diag}(\bar{\hat{\boldsymbol{k}}})\boldsymbol{F}(\sigma^2\boldsymbol{I} + r_t^2(\boldsymbol{x}_t)\boldsymbol{F}^{-1}\mathrm{diag}(\hat{\boldsymbol{k}})\mathrm{diag}(\bar{\hat{\boldsymbol{k}}})\boldsymbol{F})^{-1}(\boldsymbol{y} - \boldsymbol{A}\mathbb{E}[\boldsymbol{x}_0|\boldsymbol{x}_t]) \tag{103}$$

$$= \boldsymbol{F}^{-1}\mathrm{diag}(\bar{\hat{\boldsymbol{k}}})\boldsymbol{F}(\sigma^2\boldsymbol{I} + r_t^2(\boldsymbol{x}_t)\boldsymbol{F}^{-1}\mathrm{diag}(\hat{\boldsymbol{k}} \odot \bar{\hat{\boldsymbol{k}}})\boldsymbol{F})^{-1}(\boldsymbol{y} - \boldsymbol{A}\mathbb{E}[\boldsymbol{x}_0|\boldsymbol{x}_t]) \tag{104}$$

$$= \boldsymbol{F}^{-1}\mathrm{diag}(\bar{\hat{\boldsymbol{k}}})\boldsymbol{F}(\boldsymbol{F}^{-1}(\sigma^2\boldsymbol{I} + r_t^2(\boldsymbol{x}_t)\mathrm{diag}(\hat{\boldsymbol{k}} \odot \bar{\hat{\boldsymbol{k}}}))\boldsymbol{F})^{-1}(\boldsymbol{y} - \boldsymbol{A}\mathbb{E}[\boldsymbol{x}_0|\boldsymbol{x}_t]) \tag{105}$$

$$= \boldsymbol{F}^{-1}\mathrm{diag}(\bar{\hat{\boldsymbol{k}}})\boldsymbol{F}(\boldsymbol{F}^{-1}\mathrm{diag}(\sigma^2 + r_t^2(\boldsymbol{x}_t)\hat{\boldsymbol{k}} \odot \bar{\hat{\boldsymbol{k}}})\boldsymbol{F})^{-1}(\boldsymbol{y} - \boldsymbol{A}\mathbb{E}[\boldsymbol{x}_0|\boldsymbol{x}_t]) \tag{106}$$

$$= \boldsymbol{F}^{-1}\mathrm{diag}(\bar{\hat{\boldsymbol{k}}})\boldsymbol{F}\boldsymbol{F}^{-1}\mathrm{diag}(\sigma^2 + r_t^2(\boldsymbol{x}_t)\hat{\boldsymbol{k}} \odot \bar{\hat{\boldsymbol{k}}})^{-1}\boldsymbol{F}(\boldsymbol{y} - \boldsymbol{A}\mathbb{E}[\boldsymbol{x}_0|\boldsymbol{x}_t]) \tag{107}$$

$$= \boldsymbol{F}^{-1}(\bar{\hat{\boldsymbol{k}}} \odot \frac{\boldsymbol{F}(\boldsymbol{y} - \boldsymbol{A}\mathbb{E}[\boldsymbol{x}_0|\boldsymbol{x}_t])}{\sigma^2 + r_t^2(\boldsymbol{x}_t)\hat{\boldsymbol{k}} \odot \bar{\hat{\boldsymbol{k}}}}) \tag{108}$$

$\square$

**Super resolution.** According to (Zhang et al. (2020)), the observation model for image super resolution can be *approximately* expressed as:

$$\boldsymbol{y} = (\boldsymbol{x}_0 * \boldsymbol{k})_{\downarrow s} + \boldsymbol{n} \tag{109}$$

By leveraging the convolution property of Fourier transform, we can convert equation 109 to the canonical form $\boldsymbol{y} = \boldsymbol{A}\boldsymbol{x}_0 + \boldsymbol{n}$:

$$\boldsymbol{y} = \underbrace{\boldsymbol{D}_{\downarrow s}\boldsymbol{F}^{-1}\mathrm{diag}(\hat{\boldsymbol{k}})\boldsymbol{F}}_{\boldsymbol{A}}\boldsymbol{x}_0 + \boldsymbol{n} \tag{110}$$

and the closed-form solution to $\boldsymbol{v}$ in image super resolution is given by the following:

$$\boldsymbol{v} = \boldsymbol{F}^{-1}(\bar{\hat{\boldsymbol{k}}} \odot_s \frac{\boldsymbol{F}_{\downarrow s}(\boldsymbol{y} - \boldsymbol{A}\mathbb{E}[\boldsymbol{x}_0|\boldsymbol{x}_t])}{\sigma^2 + r_t^2(\boldsymbol{x}_t)(\bar{\hat{\boldsymbol{k}}} \odot \hat{\boldsymbol{k}})_{\Downarrow s}}) \tag{111}$$

where $\odot_s$ denotes block processing operator with element-wise multiplication (Zhang et al. (2020)).

*Proof.* Since $\boldsymbol{A}$ is a real matrix, we have $\boldsymbol{A}^T = \boldsymbol{A}^H = \boldsymbol{F}^{-1}\mathrm{diag}(\bar{\hat{\boldsymbol{k}}})\boldsymbol{F}\boldsymbol{D}_{\downarrow s}^T$, then

$$\boldsymbol{v} = \boldsymbol{F}^{-1}\mathrm{diag}(\bar{\hat{\boldsymbol{k}}})\boldsymbol{F}\boldsymbol{D}_{\downarrow s}^T(\sigma^2\boldsymbol{I} + \boldsymbol{D}_{\downarrow s}\boldsymbol{F}^{-1}\mathrm{diag}(\hat{\boldsymbol{k}})\boldsymbol{F}r_t^2(\boldsymbol{x}_t)\boldsymbol{I}\boldsymbol{F}^{-1}\mathrm{diag}(\bar{\hat{\boldsymbol{k}}})\boldsymbol{F}\boldsymbol{D}_{\downarrow s}^T)^{-1}(\boldsymbol{y} - \boldsymbol{A}\mathbb{E}[\boldsymbol{x}_0|\boldsymbol{x}_t]) \tag{112}$$

$$= \boldsymbol{F}^{-1}\mathrm{diag}(\bar{\hat{\boldsymbol{k}}})\boldsymbol{F}\boldsymbol{D}_{\downarrow s}^T(\sigma^2\boldsymbol{I} + r_t^2(\boldsymbol{x}_t)\boldsymbol{D}_{\downarrow s}\boldsymbol{F}^{-1}\mathrm{diag}(\hat{\boldsymbol{k}} \odot \bar{\hat{\boldsymbol{k}}})\boldsymbol{F}\boldsymbol{D}_{\downarrow s}^T)^{-1}(\boldsymbol{y} - \boldsymbol{A}\mathbb{E}[\boldsymbol{x}_0|\boldsymbol{x}_t]) \tag{113}$$

$$= \boldsymbol{F}^{-1}\mathrm{diag}(\bar{\hat{\boldsymbol{k}}})\boldsymbol{F}\boldsymbol{D}_{\downarrow s}^T(\boldsymbol{D}_{\downarrow s}\boldsymbol{F}^{-1}(\sigma^2\boldsymbol{I} + r_t^2(\boldsymbol{x}_t)\mathrm{diag}(\hat{\boldsymbol{k}} \odot \bar{\hat{\boldsymbol{k}}}))\boldsymbol{F}\boldsymbol{D}_{\downarrow s}^T)^{-1}(\boldsymbol{y} - \boldsymbol{A}\mathbb{E}[\boldsymbol{x}_0|\boldsymbol{x}_t]) \tag{114}$$

$$= \boldsymbol{F}^{-1}\mathrm{diag}(\bar{\hat{\boldsymbol{k}}})\boldsymbol{F}\boldsymbol{D}_{\downarrow s}^T(\boldsymbol{D}_{\downarrow s}\boldsymbol{F}^{-1}\mathrm{diag}(\sigma^2 + r_t^2(\boldsymbol{x}_t)\hat{\boldsymbol{k}} \odot \bar{\hat{\boldsymbol{k}}})\boldsymbol{F}\boldsymbol{D}_{\downarrow s}^T)^{-1}(\boldsymbol{y} - \boldsymbol{A}\mathbb{E}[\boldsymbol{x}_0|\boldsymbol{x}_t]) \tag{115}$$

By Lemma 3, we have $\boldsymbol{D}_{\downarrow s}\boldsymbol{F}^{-1} = \boldsymbol{F}_{\downarrow s}^{-1}\boldsymbol{D}_{\Downarrow s}$. Taking the hermitian transpose to both side and leveraging $\boldsymbol{F}^{-1} = \frac{1}{d}\boldsymbol{F}^H$ and $\boldsymbol{F}_{\downarrow s}^{-1} = \frac{1}{d/s}\boldsymbol{F}_{\downarrow s}^H$, we also have $\boldsymbol{F}\boldsymbol{D}_{\downarrow s}^T = s\boldsymbol{D}_{\Downarrow s}^T\boldsymbol{F}_{\downarrow s}$. So

$$\boldsymbol{v} = \boldsymbol{F}^{-1}\mathrm{diag}(\bar{\hat{\boldsymbol{k}}})s\boldsymbol{D}_{\Downarrow s}^T\boldsymbol{F}_{\downarrow s}(\boldsymbol{F}_{\downarrow s}^{-1}\boldsymbol{D}_{\Downarrow s}\mathrm{diag}(\sigma^2 + r_t^2(\boldsymbol{x}_t)\hat{\boldsymbol{k}}\odot\bar{\hat{\boldsymbol{k}}})s\boldsymbol{D}_{\Downarrow s}^T\boldsymbol{F}_{\downarrow s})^{-1}(\boldsymbol{y} - \boldsymbol{A}\mathbb{E}[\boldsymbol{x}_0|\boldsymbol{x}_t])$$
(116)

$$= \boldsymbol{F}^{-1}\mathrm{diag}(\bar{\hat{\boldsymbol{k}}})s\boldsymbol{D}_{\Downarrow s}^T\boldsymbol{F}_{\downarrow s}(\boldsymbol{F}_{\downarrow s}^{-1}\mathrm{diag}(\sigma^2 + r_t^2(\boldsymbol{x}_t)(\hat{\boldsymbol{k}}\odot\bar{\hat{\boldsymbol{k}}})_{\Downarrow s})\boldsymbol{F}_{\downarrow s})^{-1}(\boldsymbol{y} - \boldsymbol{A}\mathbb{E}[\boldsymbol{x}_0|\boldsymbol{x}_t])$$
(117)

$$= \boldsymbol{F}^{-1}\mathrm{diag}(\bar{\hat{\boldsymbol{k}}})s\boldsymbol{D}_{\Downarrow s}^T\boldsymbol{F}_{\downarrow s}\boldsymbol{F}_{\downarrow s}^{-1}\mathrm{diag}(\sigma^2 + r_t^2(\boldsymbol{x}_t)(\hat{\boldsymbol{k}}\odot\bar{\hat{\boldsymbol{k}}})_{\Downarrow s})^{-1}\boldsymbol{F}_{\downarrow s}(\boldsymbol{y} - \boldsymbol{A}\mathbb{E}[\boldsymbol{x}_0|\boldsymbol{x}_t])$$
(118)

$$= \boldsymbol{F}^{-1}(\bar{\hat{\boldsymbol{k}}}\odot s\boldsymbol{D}_{\Downarrow s}^T\frac{\boldsymbol{F}_{\downarrow s}(\boldsymbol{y} - \boldsymbol{A}\mathbb{E}[\boldsymbol{x}_0|\boldsymbol{x}_t])}{\sigma^2 + r_t^2(\boldsymbol{x}_t)(\hat{\boldsymbol{k}}\odot\bar{\hat{\boldsymbol{k}}})_{\Downarrow s}})$$
(119)

$$= \boldsymbol{F}^{-1}(\bar{\hat{\boldsymbol{k}}}\odot_s\frac{\boldsymbol{F}_{\downarrow s}(\boldsymbol{y} - \boldsymbol{A}\mathbb{E}[\boldsymbol{x}_0|\boldsymbol{x}_t])}{\sigma^2 + r_t^2(\boldsymbol{x}_t)(\bar{\hat{\boldsymbol{k}}}\odot\hat{\boldsymbol{k}})_{\Downarrow s}})$$
(120)

$\square$

## B.2 CLOSED-FORM SOLUTIONS TO PROXIMAL PROBLEMS IN TYPE II GUIDANCE

When using Type II guidance, we are required to solve the following *auto-weighted* proximal problem:

$$\mathbb{E}_q[\boldsymbol{x}_0|\boldsymbol{x}_t, \boldsymbol{y}] = \arg\min_{\boldsymbol{x}_0}\|\boldsymbol{y} - \boldsymbol{A}\boldsymbol{x}_0\|^2 + \sigma^2\|\boldsymbol{x}_0 - \mathbb{E}[\boldsymbol{x}_0|\boldsymbol{x}_t]\|_{\boldsymbol{\Sigma}_t^{-1}(\boldsymbol{x}_t)}^2$$
(121)

which has general closed-form solution given by

$$\mathbb{E}_q[\boldsymbol{x}_0|\boldsymbol{x}_t, \boldsymbol{y}] = (\boldsymbol{\Sigma}_t(\boldsymbol{x}_t)^{-1} + \frac{1}{\sigma^2}\boldsymbol{A}^T\boldsymbol{A})^{-1}(\frac{1}{\sigma^2}\boldsymbol{A}^T\boldsymbol{y} + \boldsymbol{\Sigma}_t(\boldsymbol{x}_t)^{-1}\mathbb{E}[\boldsymbol{x}_0|\boldsymbol{x}_t])$$
(122)

The derivation of equation 122 can be directly obtained via computing the mean of $q_t(\boldsymbol{x}_0|\boldsymbol{x}_t, \boldsymbol{y})$ by using the Bayes' theorem for Gaussian variables (Bishop (2006)).

Below we provide closed-form solution to $\mathbb{E}_q[\boldsymbol{x}_0|\boldsymbol{x}_t, \boldsymbol{y}]$ under three common degradation operators same as for Type I guidance, then we can directly replace the denoising result in unconditional diffusion step with the closed-form solution of $\mathbb{E}_q[\boldsymbol{x}_0|\boldsymbol{x}_t, \boldsymbol{y}]$ to solve target inverse problem. The closed-form results are obtained by borrowing the results from (Zhu et al. (2023)) and setting $\rho_t$ to $\frac{\sigma^2}{r_t^2(\boldsymbol{x}_t)}$.

**Inpainting.** The observation model for image inpainting see equation 92, and the closed-form solution to $\mathbb{E}_q[\boldsymbol{x}_0|\boldsymbol{x}_t, \boldsymbol{y}]$ is given as follows:

$$\mathbb{E}_q[\boldsymbol{x}_0|\boldsymbol{x}_t, \boldsymbol{y}] = \frac{\tilde{\boldsymbol{y}} + \rho_t\mathbb{E}[\boldsymbol{x}_0|\boldsymbol{x}_t]}{\rho_t + \boldsymbol{m}}, \quad \rho_t = \frac{\sigma^2}{r_t^2(\boldsymbol{x}_t)}$$
(123)

**Debluring.** The observation model for image debluring see equation 99 and equation 100, and the closed-form solution to $\mathbb{E}_q[\boldsymbol{x}_0|\boldsymbol{x}_t, \boldsymbol{y}]$ is given as follows:

$$\mathbb{E}_q[\boldsymbol{x}_0|\boldsymbol{x}_t, \boldsymbol{y}] = \boldsymbol{F}^{-1}\frac{\bar{\hat{\boldsymbol{k}}}\odot\boldsymbol{F}\boldsymbol{y} + \rho_t\boldsymbol{F}\mathbb{E}[\boldsymbol{x}_0|\boldsymbol{x}_t]}{\bar{\hat{\boldsymbol{k}}}\odot\hat{\boldsymbol{k}} + \rho_t}, \quad \rho_t = \frac{\sigma^2}{r_t^2(\boldsymbol{x}_t)}$$
(124)

**Super resolution.** The observation model for image super resolution see equation 109 and equation 110, and the closed-form solution to $\mathbb{E}_q[\boldsymbol{x}_0|\boldsymbol{x}_t, \boldsymbol{y}]$ is given as follows:

$$\mathbb{E}_q[\boldsymbol{x}_0|\boldsymbol{x}_t, \boldsymbol{y}] = \boldsymbol{F}^{-1}\frac{1}{\rho_t}(\boldsymbol{d} - \bar{\hat{\boldsymbol{k}}}\odot_s\frac{(\hat{\boldsymbol{k}}\odot\boldsymbol{d})_{\Downarrow s}}{(\bar{\hat{\boldsymbol{k}}}\odot\hat{\boldsymbol{k}})_{\Downarrow s} + \rho_t}), \quad \rho_t = \frac{\sigma^2}{r_t^2(\boldsymbol{x}_t)}$$
(125)

where $\boldsymbol{d} = \hat{\boldsymbol{k}}\odot(\boldsymbol{F}\boldsymbol{y}_{\uparrow s}) + \rho_t\boldsymbol{F}\mathbb{E}[\boldsymbol{x}_0|\boldsymbol{x}_t]$ and $\uparrow_s$ denotes the standard $s$-fold upsampling, i.e., upsampling the spatial size by filling the new entries with zeros (Zhang et al., 2020).

## C USING CONJUGETE GRADIENT METHOD FOR GENERAL POSTERIOR COVARIANCE

For general posterior covariance, the closed-form solution for $\boldsymbol{v}$ or $\mathbb{E}_q[\boldsymbol{x}_0|\boldsymbol{x}_t, \boldsymbol{y}]$ is usually unavailable. However, we can still approximate their solutions by using numerical solutions of linear equations. Specifically, we can see that for Type II guidance, $\mathbb{E}_q[\boldsymbol{x}_0|\boldsymbol{x}_t, \boldsymbol{y}]$ satisfies the following linear equation:

$$(\boldsymbol{\Sigma}_t(\boldsymbol{x}_t)^{-1} + \frac{1}{\sigma^2}\boldsymbol{A}^T\boldsymbol{A})\mathbb{E}_q[\boldsymbol{x}_0|\boldsymbol{x}_t, \boldsymbol{y}] = \frac{1}{\sigma^2}\boldsymbol{A}^T\boldsymbol{y} + \boldsymbol{\Sigma}_t(\boldsymbol{x}_t)^{-1}\mathbb{E}[\boldsymbol{x}_0|\boldsymbol{x}_t] \tag{126}$$

Since $\boldsymbol{\Sigma}_t(\boldsymbol{x}_t)^{-1} + \frac{1}{\sigma^2}\boldsymbol{A}^T\boldsymbol{A}$ is in fact the precision matrix of $q_t(\boldsymbol{x}_0|\boldsymbol{x}_t, \boldsymbol{y})$, we know that it is symmetric and positive-definite. Therefore, we can solve equation 126 using conjugate gradient (CG) method (Hestenes et al. (1952)).

Likewise, for Type I guidance, it easy to see that $\sigma^2\boldsymbol{I} + \boldsymbol{A}\boldsymbol{\Sigma}_t(\boldsymbol{x}_t)\boldsymbol{A}^T$ is symmetric and positive-definite. To approximate $\boldsymbol{v}$, we define a temporal variable $\boldsymbol{u}$, and approximate it using CG to solve the following linear equation:

$$(\sigma^2\boldsymbol{I} + \boldsymbol{A}\boldsymbol{\Sigma}_t(\boldsymbol{x}_t)\boldsymbol{A}^T)\boldsymbol{u} = \boldsymbol{y} - \boldsymbol{A}\mathbb{E}[\boldsymbol{x}_0|\boldsymbol{x}_t] \tag{127}$$

Then $\boldsymbol{v}$ can be obtained as follows:

$$\boldsymbol{v} = \boldsymbol{A}^T\boldsymbol{u} \tag{128}$$

In our experiments, only the Convert-type posterior covariance (Section 4.2) requires to use the CG method, as the posterior covariances used in prior works and Analytic-type posterior covariance (Section 4.3) are isotropic that possess closed-form solutions. We use the black box CG method implemented in `scipy.sparse.linalg.cg` with default hyper-parameter setting.

## D CONVERTING OPTIMAL SOLUTIONS BETWEEN DIFFERENT PERTURBATION KERNELS

Suppose we are given a family of optimal solutions $q_t$ defined by the following MLE objectives for all $t \in [0, T]$:

$$\max_{q_t} \mathbb{E}_{\boldsymbol{x}_0\sim p(\boldsymbol{x}_0), \epsilon\sim\mathcal{N}(\mathbf{0}, \boldsymbol{I})} \log q_t(\boldsymbol{x}_0|\boldsymbol{x}_t = s_t(\boldsymbol{x}_0 + \sigma_t\epsilon)) \tag{129}$$

As can be seen, $q_t$ equal to the optimal solutions to equation 15 when the perturbation kernels $p_t(\boldsymbol{x}_t|\boldsymbol{x}_0)$ are set to $\mathcal{N}(\boldsymbol{x}_t|s_t\boldsymbol{x}_0, s_t^2\sigma_t^2\boldsymbol{I})$. Now, suppose we want to perform sampling based on the diffusion ODE (SDE) under the perturbation kernels $\mathcal{N}(\boldsymbol{x}_t|\tilde{s}_t\boldsymbol{x}_0, \tilde{s}_t^2\tilde{\sigma}_t^2\boldsymbol{I})$. This means that we are required to provide optimal solutions $\tilde{q}_t$ defined by the following objectives for all $t \in [0, T]$:

$$\max_{\tilde{q}_t} \mathbb{E}_{\boldsymbol{x}_0\sim p(\boldsymbol{x}_0), \epsilon\sim\mathcal{N}(\mathbf{0}, \boldsymbol{I})} \log \tilde{q}_t(\boldsymbol{x}_0|\boldsymbol{x}_t = \tilde{s}_t(\boldsymbol{x}_0 + \tilde{\sigma}_t\epsilon)) \tag{130}$$

The idea is that, $\tilde{q}_t$ can be directly represented by $q_t$, so we do not require to perform re-training:

$$\tilde{q}_t(\boldsymbol{x}_0|\boldsymbol{x}_t = \boldsymbol{x}) = q_{t'}(\boldsymbol{x}_0|\boldsymbol{x}_{t'} = \frac{s_{t'}}{\tilde{s}_t}\boldsymbol{x}), \quad \sigma_{t'} = \tilde{\sigma}_t \tag{131}$$

*Proof.* We only need to show that $\tilde{q}_t$ defined by equation 129 and equation 131 are equivalent to $\tilde{q}_t$ defined by equation 130. From equation 131, we actually know that $q_{t'}(\boldsymbol{x}_0|\boldsymbol{x}_{t'} = \boldsymbol{x}) = \tilde{q}_t(\boldsymbol{x}_0|\boldsymbol{x}_t = \frac{\tilde{s}_t}{s_{t'}}\boldsymbol{x})$, plug it in equation 129 at $t'$ we have

$$\max_{\tilde{q}_t} \mathbb{E}_{\boldsymbol{x}_0\sim p(\boldsymbol{x}_0), \epsilon\sim\mathcal{N}(\mathbf{0}, \boldsymbol{I})} \log \tilde{q}_t(\boldsymbol{x}_0|\boldsymbol{x}_t = \tilde{s}_t(\boldsymbol{x}_0 + \sigma_{t'}\epsilon)), \quad \sigma_{t'} = \tilde{\sigma}_t \tag{132}$$

which is equivalent to equation 130. $\square$

For example, suppose we are given optimal solutions $q_t$ under DDPM perturbation kernels, i.e., $p_t(\boldsymbol{x}_t|\boldsymbol{x}_0) = \mathcal{N}(\boldsymbol{x}_t|\sqrt{\bar{\alpha}_t}\boldsymbol{x}_0, \bar{\beta}_t\boldsymbol{I})$. We aim to convert these solutions to optimal solutions $\tilde{q}_t$ under perturbation kernels used in Section 2.2, i.e., $p_t(\boldsymbol{x}_t|\boldsymbol{x}_0) = \mathcal{N}(\boldsymbol{x}_t|\boldsymbol{x}_0, t^2\boldsymbol{I})$. We can realize it using

following two steps: (1) finding $t'$ such that $\sqrt{\frac{\bar{\beta}_{t'}}{\bar{\alpha}_{t'}}} = t$, and (2) scaling the input of $q_{t'}$ by the factor of $\sqrt{\bar{\alpha}_{t'}}$. Formally,

$$\tilde{q}_t(\boldsymbol{x}_0|\boldsymbol{x}_t = \boldsymbol{x}) = q_{t'}(\boldsymbol{x}_0|\boldsymbol{x}_{t'} = \sqrt{\bar{\alpha}_{t'}}\boldsymbol{x}), \quad \sqrt{\frac{\bar{\beta}_{t'}}{\bar{\alpha}_{t'}}} = t \tag{133}$$

# E    ADDITIONAL EXPERIMENTAL DETAILS AND RESULTS

## E.1    RESULTS ON COMPLETE VERSION OF PRIOR WORKS

Aside from using Gaussians to approximate the posteriors, prior works also propose several tricks to improve the performance. However, to eliminate the influence other than the choice of posterior covariance, results reported Table 3 based on the re-implementations that remove these tricks. For the sake of completeness, we also re-implement these tricks and to investigate if replace the last sampling steps on these complete re-implementations with our techniques can also improve the performance.

| Dataset | Method | Inpaint (Random) | | | Deblur (Gaussian) | | | Deblur (Motion) | | | Super resolution ($4\times$) | | |
|---|---|---|---|---|---|---|---|---|---|---|---|---|---|
| | | PSNR $\uparrow$ | LPIPS $\downarrow$ | FID $\downarrow$ | PSNR $\uparrow$ | LPIPS $\downarrow$ | FID $\downarrow$ | PSNR $\uparrow$ | LPIPS $\downarrow$ | FID $\downarrow$ | PSNR $\uparrow$ | LPIPS $\downarrow$ | FID $\downarrow$ |
| FFHQ | $\Pi$GDM | 25.48 | 0.2605 | 77.46 | 25.74 | 0.2421 | 71.19 | 25.16 | 0.2607 | 75.15 | 25.70 | 0.2442 | 72.41 |
| | Analytic (*Ours*) | **33.38** | **0.0852** | 28.63 | **27.32** | 0.1971 | 59.80 | 25.97 | 0.2336 | 69.82 | 26.79 | 0.2195 | 68.84 |
| | Convert (*Ours*) | 33.23 | 0.0822 | **27.50** | 27.29 | **0.1969** | **59.31** | **25.99** | **0.2324** | **66.18** | **26.80** | **0.2183** | **67.17** |
| ImageNet | $\Pi$GDM | 22.79 | 0.4293 | 141.80 | 22.82 | 0.4095 | 127.41 | 22.45 | 0.4267 | 132.38 | 22.88 | 0.4098 | 127.40 |
| | Analytic (*Ours*) | 28.91 | 0.1484 | 37.98 | 23.89 | 0.3583 | 103.67 | **23.05** | 0.3947 | **126.39** | 23.57 | 0.3867 | 116.87 |
| | Convert (*Ours*) | **29.01** | **0.1394** | **33.46** | **23.98** | **0.3568** | **100.50** | **23.05** | **0.3944** | 131.48 | **23.68** | **0.3846** | **116.27** |

Table 4: **Results for complete version of $\Pi$GDM on FFHQ and ImageNet dataset.** $\Pi$GDM introduces adaptive weight to adjust the guidance strength according to the timestep. We use bold and underline for the best and second best, respectively.

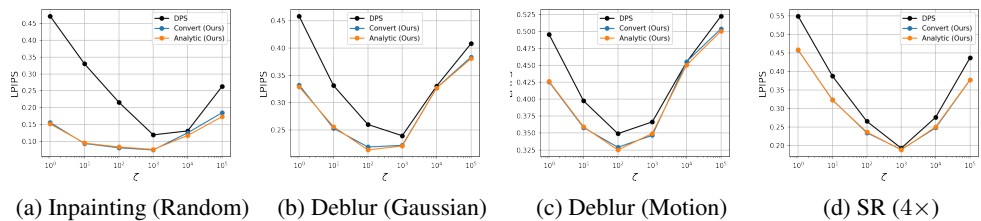

(a) Inpainting (Random)    (b) Deblur (Gaussian)    (c) Deblur (Motion)    (d) SR ($4\times$)

Figure 4: **Quantitative results (LPIPS) for complete version of DPS on FFHQ dataset.** We report the LPIPS performance under different $\zeta$.

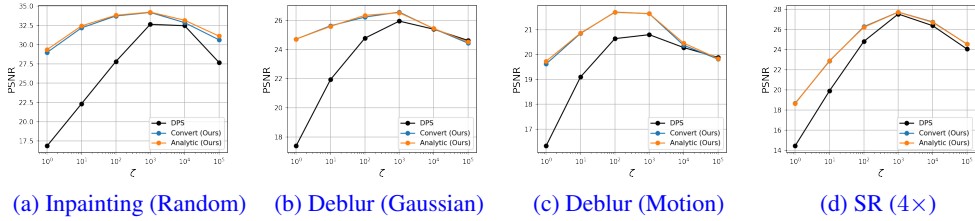

(a) Inpainting (Random)    (b) Deblur (Gaussian)    (c) Deblur (Motion)    (d) SR ($4\times$)

Figure 5: **Quantitative results (PSNR) for complete version of DPS on FFHQ dataset.** We report the PSNR performance under different $\zeta$.

**$\Pi$GDM** $\Pi$GDM introduces *adaptive weight* in front of the likelihood score $\nabla_{\boldsymbol{x}_t} \log p_t(\boldsymbol{y}|\boldsymbol{x}_t)$ (Section 3.3, Song et al. (2023)), which will adjust the guidance strength according to the timestep. Specifically, they replace the likelihood score with $r_t^2 \nabla_{\boldsymbol{x}_t} \log p_t(\boldsymbol{y}|\boldsymbol{x}_t)$, where $r_t^2 = \frac{\sigma_t^2}{1+\sigma_t^2}$.

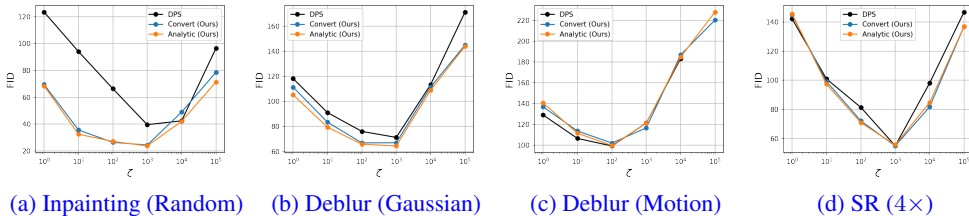

| (a) Inpainting (Random) | (b) Deblur (Gaussian) | (c) Deblur (Motion) | (d) SR ($4\times$) |

Figure 6: **Quantitative results (FID) for complete version of DPS on FFHQ dataset.** We report the FID performance under different $\zeta$.

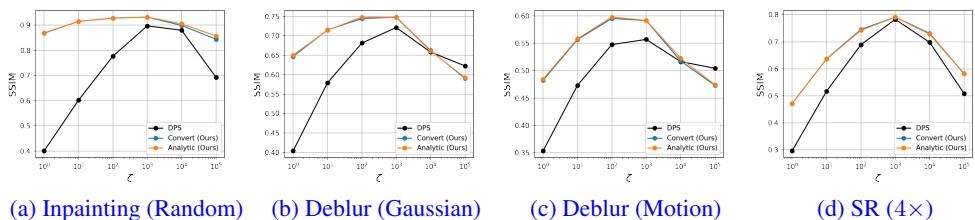

| (a) Inpainting (Random) | (b) Deblur (Gaussian) | (c) Deblur (Motion) | (d) SR ($4\times$) |

Figure 7: **Quantitative results (SSIM) for complete version of DPS on FFHQ dataset.** We report the SSIM performance under different $\zeta$.

**DPS** As mentioned in Section 3.1, DPS introduce a trick to determine the strength of the guidance. Since this trick involves a hyper-parameter $\zeta$, we report the performance under different $\zeta$.

Table 4 summarizes the results for complete version of $\Pi$GDM. $\Pi$GDM refer to our re-implementation of complete version of $\Pi$GDM, `Analytic` and `Convert` refer to complete version of $\Pi$GDM with the last few sampling steps are replaced with Type I guidance using optimal posterior covariance presented in Section 4.3 and Section 4.2 [8], respectively. Figure 4, 5, 6, and 7 summarizes the quantitative results for complete version of DPS. `DPS` refer to our re-implementation of complete version of DPS, and similarly, `Analytic` and `Convert` refer to complete version of DPS with the last few sampling steps are replaced by our techniques.

## E.2 ADDITIONAL QUANTITATIVE RESULTS

In this section, we report the SSIM performance for supplementary results of Type I guidance, and PSNR, SSIM, and FID performance for supplementary results of Type II guidance.

| Dataset | Method | Inpaint (Random) SSIM ↑ | Deblur (Gaussian) SSIM ↑ | Deblur (Motion) SSIM ↑ | Super resolution ($4\times$) SSIM ↑ |
|---------|--------|---------|---------|---------|---------|
| FFHQ | DPS | 0.8891 | 0.6284 | 0.4904 | 0.7719 |
| | $\Pi$GDM | 0.8784 | 0.7890 | 0.7543 | 0.7850 |
| | Analytic (*Ours*) | 0.9272 | **0.7926** | 0.7579 | **0.7878** |
| | Convert (*Ours*) | **0.9279** | 0.7905 | **0.7584** | **0.7878** |
| ImageNet | DPS | **0.8623** | 0.4603 | 0.3582 | 0.5860 |
| | $\Pi$GDM | 0.7658 | 0.5946 | 0.5534 | 0.5925 |
| | Analytic (*Ours*) | 0.8481 | **0.6009** | 0.5611 | **0.5958** |
| | Convert (*Ours*) | 0.8559 | 0.6007 | **0.5634** | 0.5869 |

Table 5: **Quantitative results (SSIM) on FFHQ and ImageNet dataset for Type I guidance.** We use bold and underline for the best and second best, respectively.

---

[8] Same to Section 5.2, we replace when $\sigma_t < 0.2$, and it should be noted that we *do not* use adaptive weight here.

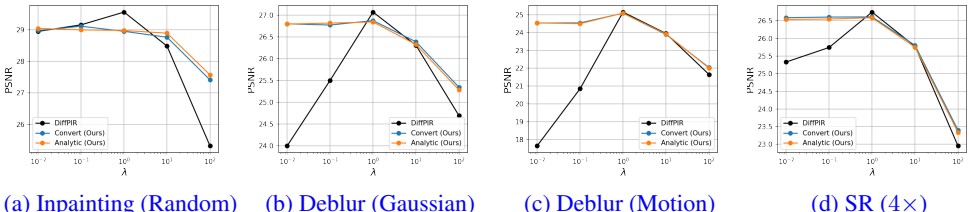

(a) Inpainting (Random)   (b) Deblur (Gaussian)   (c) Deblur (Motion)   (d) SR ($4\times$)

Figure 8: **Quantitative results (PSNR) on FFHQ dataset for Type II guidance.** We report the PSNR performance under different $\lambda$.

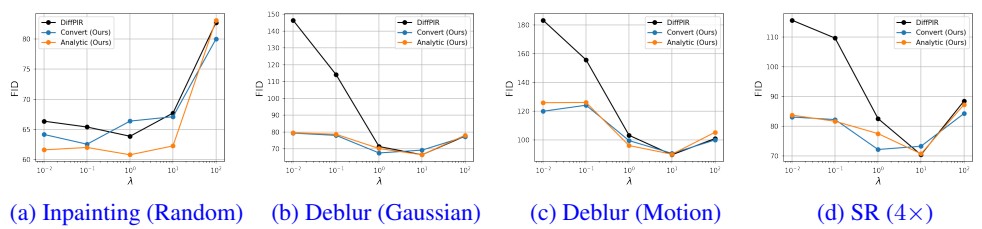

(a) Inpainting (Random)   (b) Deblur (Gaussian)   (c) Deblur (Motion)   (d) SR ($4\times$)

Figure 9: **Quantitative results (FID) on FFHQ dataset for Type II guidance.** We report the FID performance under different $\lambda$.

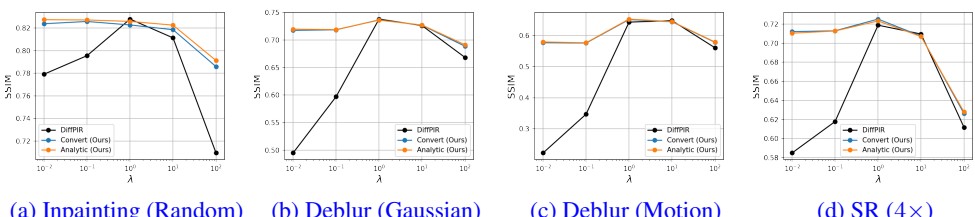

(a) Inpainting (Random)   (b) Deblur (Gaussian)   (c) Deblur (Motion)   (d) SR ($4\times$)

Figure 10: **Quantitative results (SSIM) on FFHQ dataset for Type II guidance.** We report the SSIM performance under different $\lambda$.

### E.3 TIME COMPLEXITY

The speed of inference of our method depends on whether `Analytic` or `Convert` is chosen.

In the case of `Analytic`, our method does not impose any additional computational cost for online inference compared to previous works. The computational expense arises from the offline pre-calculation of $\hat{r}_t^2$ using equation 24. Similar to Bao et al. (2022b), a small number of Monte Carlo samples is typically adequate for obtaining an accurate $\hat{r}_t$. With a pre-trained unconditional model and training dataset, we compute $\hat{r}_t$ offline and deploy it alongside the pre-trained unconditional model. As a result, the online inference cost of our method is the same as previous methods. In our paper, we initially calculate $\hat{r}_t$ using a randomly selected $0.5\%$ of the training set and reuse it for all our experiments. This process takes a couple of hours on a single 1080Ti GPU.

For `Convert`, the CG method can introduce additional computational cost. However, we've observed that CG converges rapidly ($\sim 5$ steps when initialize at $D_t(\boldsymbol{x}_t) \approx \mathbb{E}[\boldsymbol{x}_0|\boldsymbol{x}_t]$) and is only employed during the last few sampling steps. The additional computational cost of CG is negligible compared to the overall inference time.

### E.4 IMPLEMENTATION OF DDNM SOLUTIONS

Using DDNM solution (equation 11) for Type II guidance in noiseless inverse problems requires to compute the pseudo-inverse $\boldsymbol{A}^\dagger$ for a given linear operator $\boldsymbol{A}$. Below we present the implementations of the pseudo-inverses used in our experiments.

**Inpainting.** We construct the DDNM solution $\mathbb{E}_q[\boldsymbol{x}_0|\boldsymbol{x}_t, \boldsymbol{y}]$ for the inpainting case $\boldsymbol{A} = \boldsymbol{D_m}$ as follows

$$\mathbb{E}_q[\boldsymbol{x}_0|\boldsymbol{x}_t, \boldsymbol{y}] = \tilde{\boldsymbol{y}} + (1 - \boldsymbol{m}) \odot \mathbb{E}[\boldsymbol{x}_0|\boldsymbol{x}_t] \tag{134}$$

*Proof.* Since $\boldsymbol{D_m}\boldsymbol{D_m^T} = \boldsymbol{I}$ is non-singular, we can directly obtain $\boldsymbol{A}^\dagger$ by $\boldsymbol{A}^\dagger = \boldsymbol{D_m^T}(\boldsymbol{D_m}\boldsymbol{D_m^T})^{-1} = \boldsymbol{D_m^T}$. Plug in equation 11, we have

$$\mathbb{E}_q[\boldsymbol{x}_0|\boldsymbol{x}_t, \boldsymbol{y}] = \boldsymbol{D_m^T}\boldsymbol{y} + (\boldsymbol{I} - \boldsymbol{D_m^T}\boldsymbol{D_m})\mathbb{E}[\boldsymbol{x}_0|\boldsymbol{x}_t] \tag{135}$$
$$= \tilde{\boldsymbol{y}} + (\boldsymbol{I} - \operatorname{diag}(\boldsymbol{m}))\mathbb{E}[\boldsymbol{x}_0|\boldsymbol{x}_t] \tag{136}$$
$$= \tilde{\boldsymbol{y}} + \operatorname{diag}(1 - \boldsymbol{m})\mathbb{E}[\boldsymbol{x}_0|\boldsymbol{x}_t] \tag{137}$$
$$= \tilde{\boldsymbol{y}} + (1 - \boldsymbol{m}) \odot \mathbb{E}[\boldsymbol{x}_0|\boldsymbol{x}_t] \tag{138}$$

$\square$

**Debluring.** We construct the pseudo inverse $\boldsymbol{A}^\dagger$ for the linear operator in the debluring case $\boldsymbol{A} = \boldsymbol{F}^{-1}\operatorname{diag}(\hat{\boldsymbol{k}})\boldsymbol{F}$ as follows

$$\boldsymbol{A}^\dagger = \boldsymbol{F}^{-1}\operatorname{diag}(\hat{\boldsymbol{k}})^\dagger\boldsymbol{F} \tag{139}$$

where $\operatorname{diag}(\hat{\boldsymbol{k}})^\dagger$ is defined as $\operatorname{diag}(\hat{\boldsymbol{k}})^\dagger = \operatorname{diag}([l_1, l_2, ...]^T)$ with $l_i = 0$ if $\hat{\boldsymbol{k}}_i = 0$[9] otherwise $l_i = 1/\hat{\boldsymbol{k}}_i$.

*Proof.* It is easy to see that with the above construction, $\boldsymbol{A}^\dagger$ satisfies $\boldsymbol{A}\boldsymbol{A}^\dagger\boldsymbol{A} = \boldsymbol{A}$. $\square$

**Super resolution.** We directly use `torch.nn.functional.interpolate` in place of $\boldsymbol{A}^\dagger$ for the super resolution case.

### E.5 QUALITATIVE RESULTS

In this section, we present additional visual examples using FFHQ datasets to showcase the capabilities of our method. Figure 11, 12, 13, and 14 serve as complementary visual examples to Table 3. These illustrations consistently demonstrate that our optimal covariances generate superior images compared to previous heuristic covariances in Type I guidance. Furthermore, Figure 15 demonstrates how our methods can enhance the robustness of DiffPIR against the hyper-parameter $\lambda$, taking random inpainting as an example. We also display the notable performance improvement of ΠGDM with the adaptive weight, i.e., complete version of ΠGDM, in Figure 16, 17, 18, and 19. Lastly, Figure 20 illustrates that our methods can enhance the DPS robustness to the hyper-parameter $\zeta$, using random inpainting as an example.

---

[9]Numerically, $|\hat{\boldsymbol{k}}_i|$ is always larger than zero. We threshold $\hat{\boldsymbol{k}}_i$ to zero when $|\hat{\boldsymbol{k}}_i| < 3 \times 10^{-2}$ similar to Wang et al. (2023)

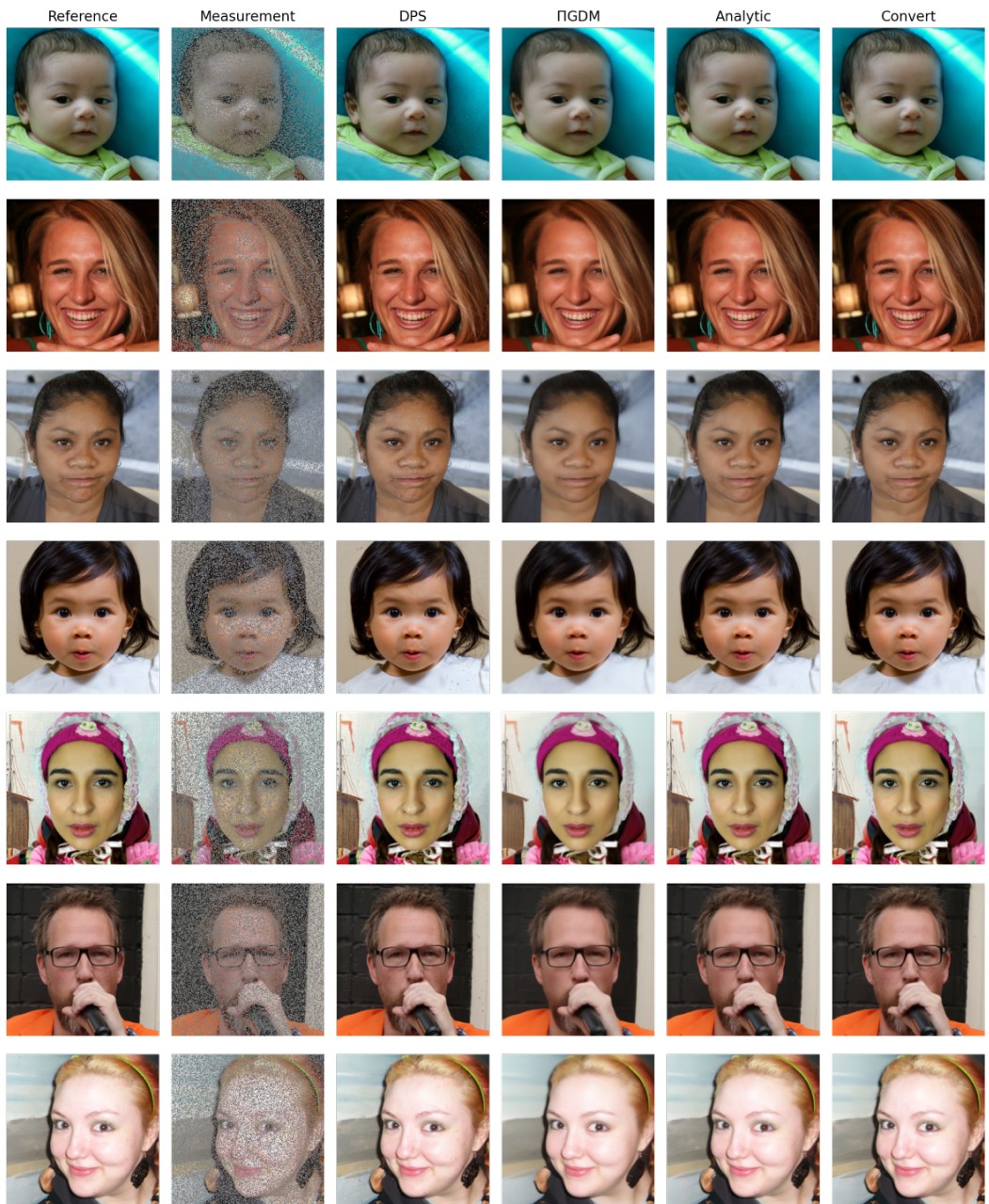

Figure 11: **Qualitative results for Table 3 on random inpainting.**

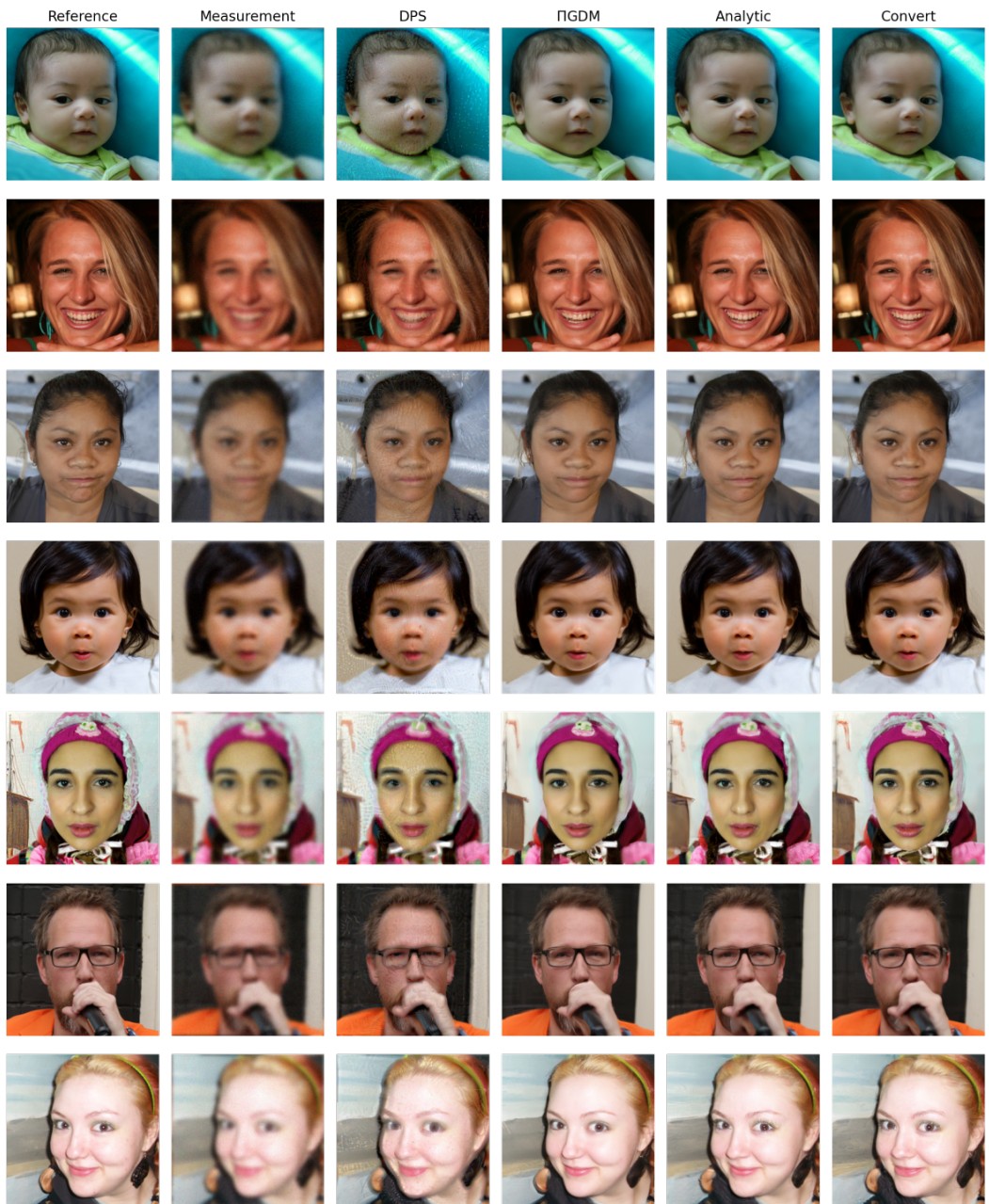

Figure 12: **Qualitative results for Table 3 on Gaussian debluring.**

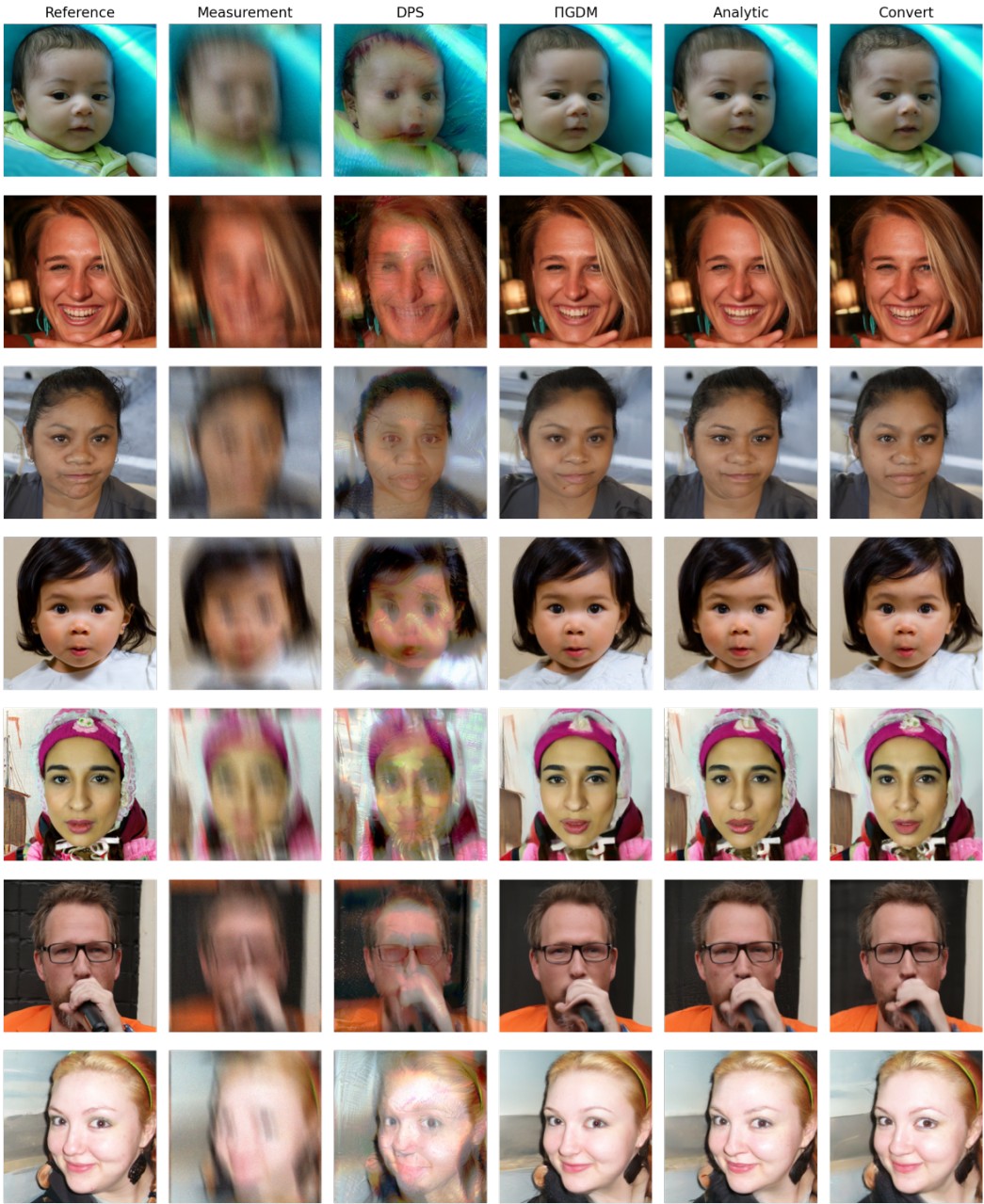

Figure 13: **Qualitative results for Table 3 on motion debluring.**

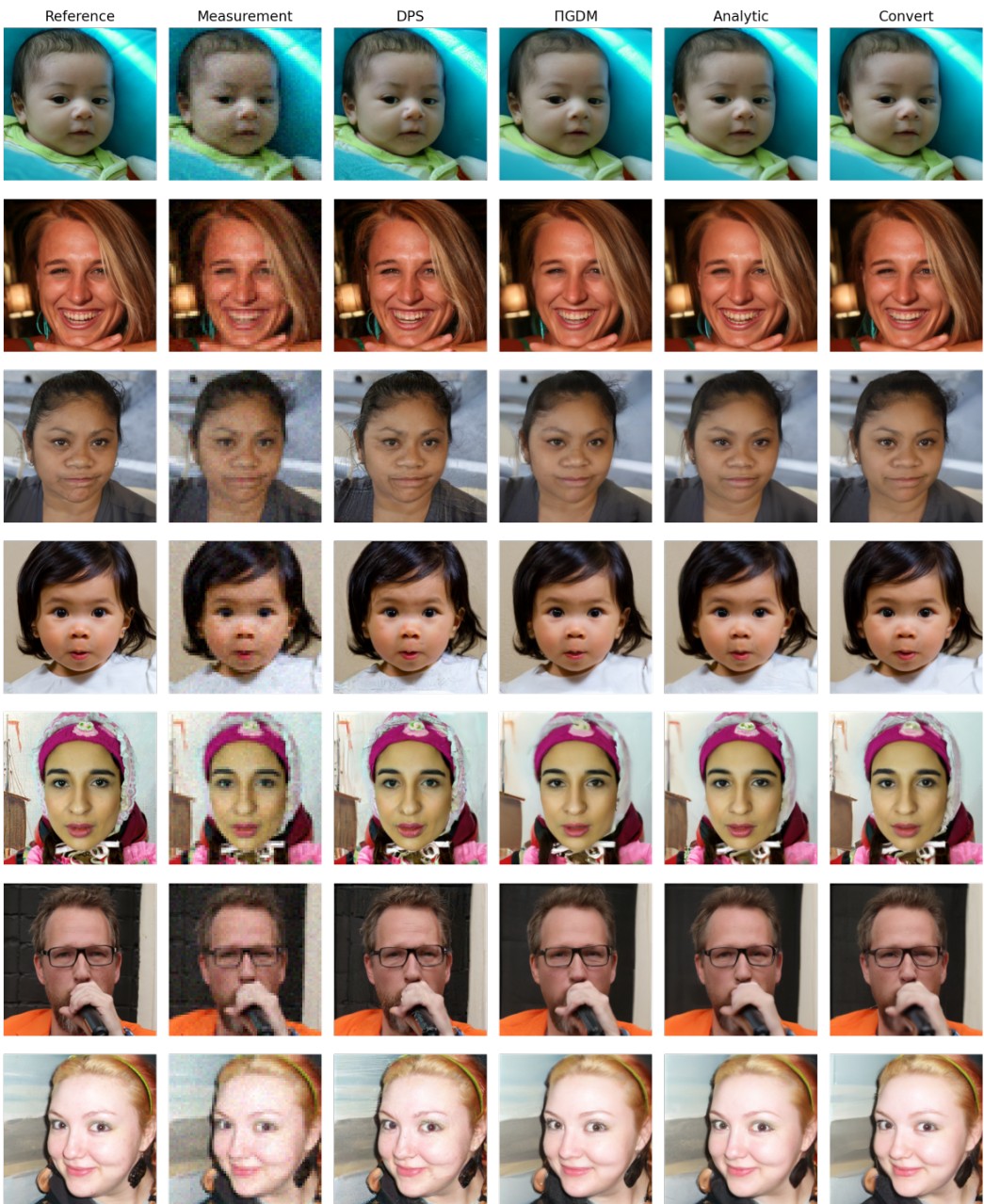

Figure 14: **Qualitative results for Table 3 on super resolution** ($4\times$).

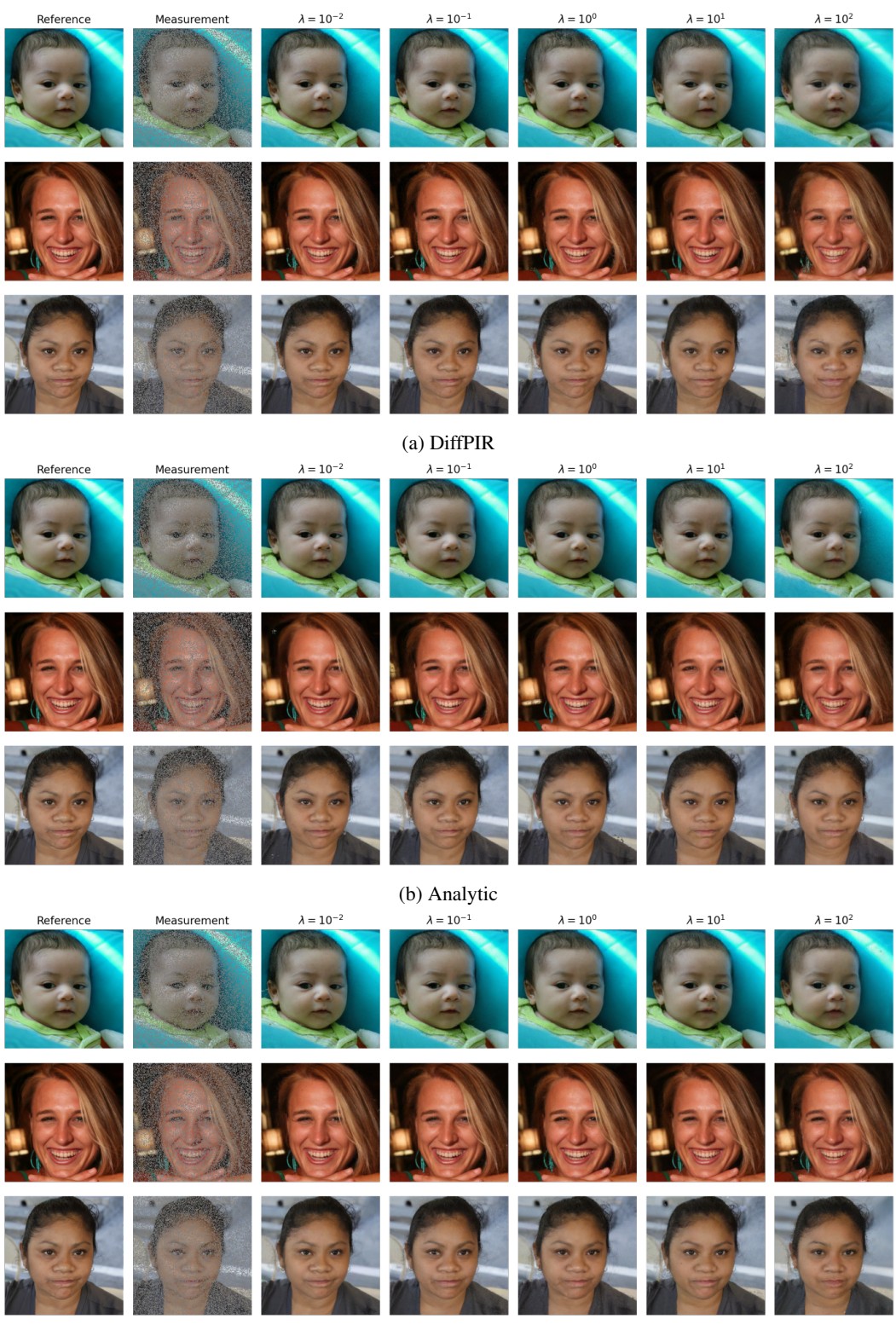

Figure 15: **Qualitative results for Type II guidance on random inpainting.** As can be seen, our methods significantly improve DiffPIR robustness to the hyper-parameter $\lambda$.

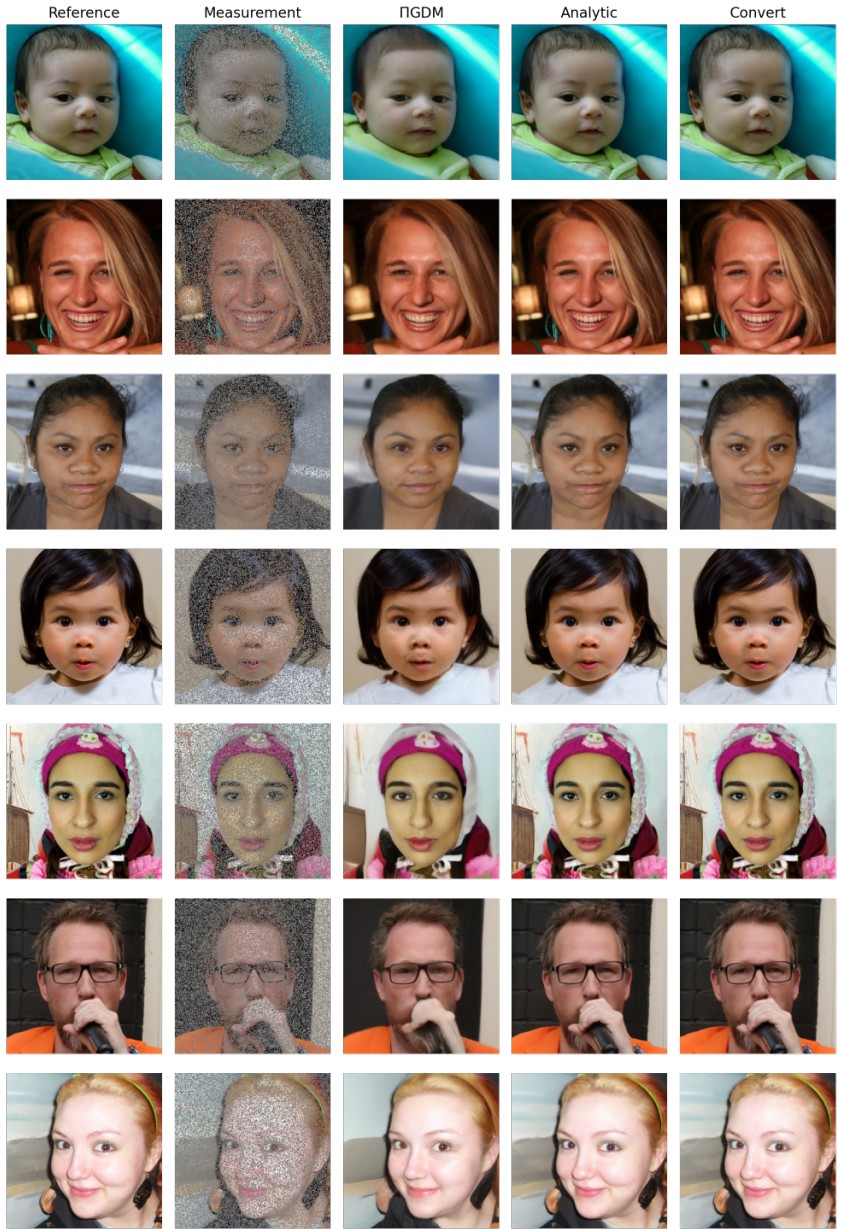

Figure 16: **Qualitative results for complete ΠGDM on random inpainting.**

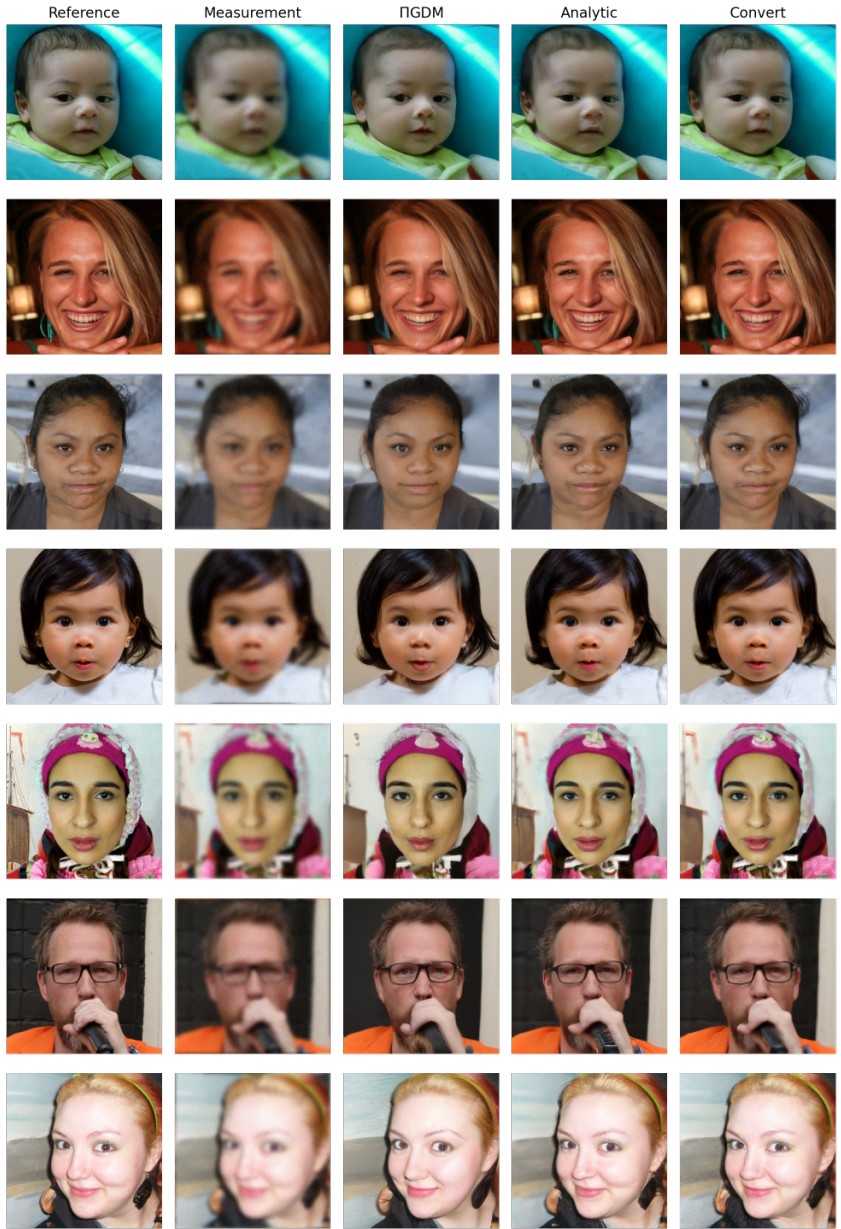

Figure 17: **Qualitative results for complete ΠGDM on Gaussian debluring.**

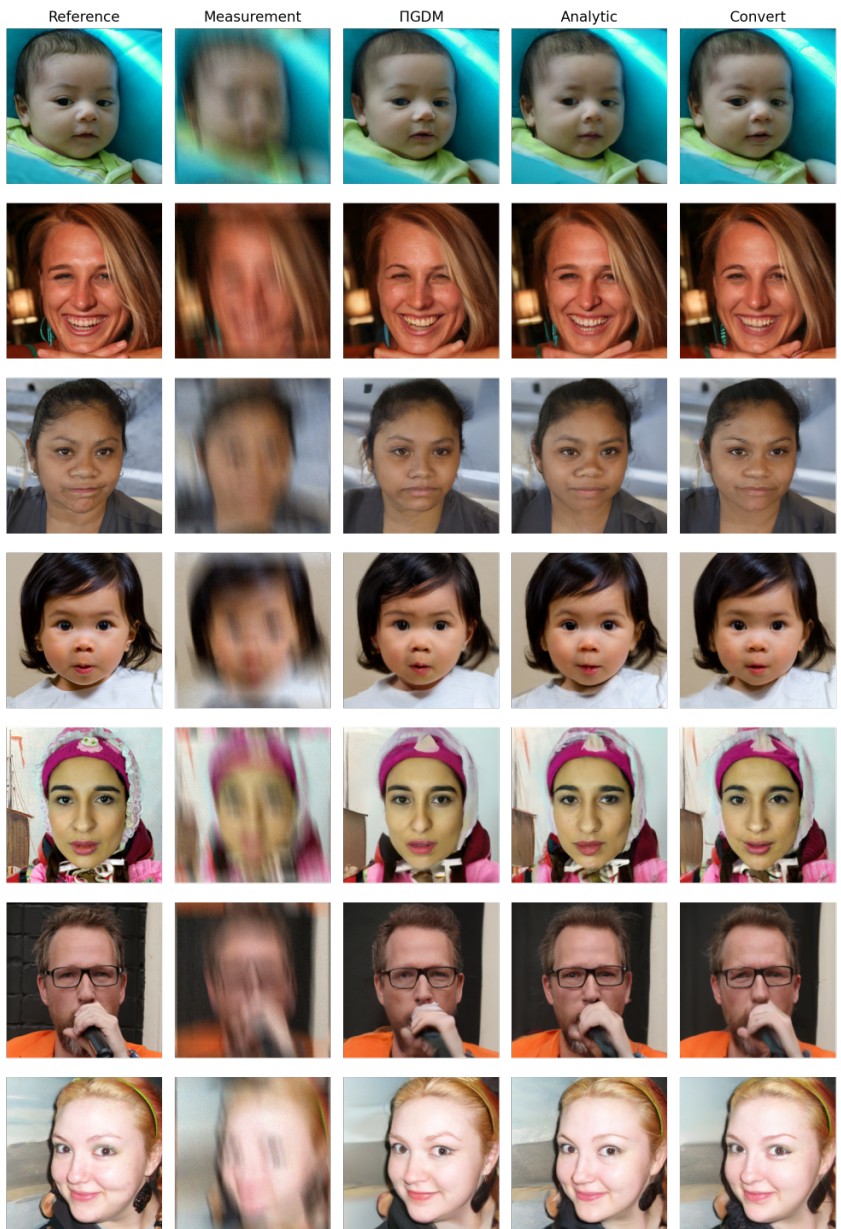

Figure 18: **Qualitative results for complete ΠGDM on motion debluring.**

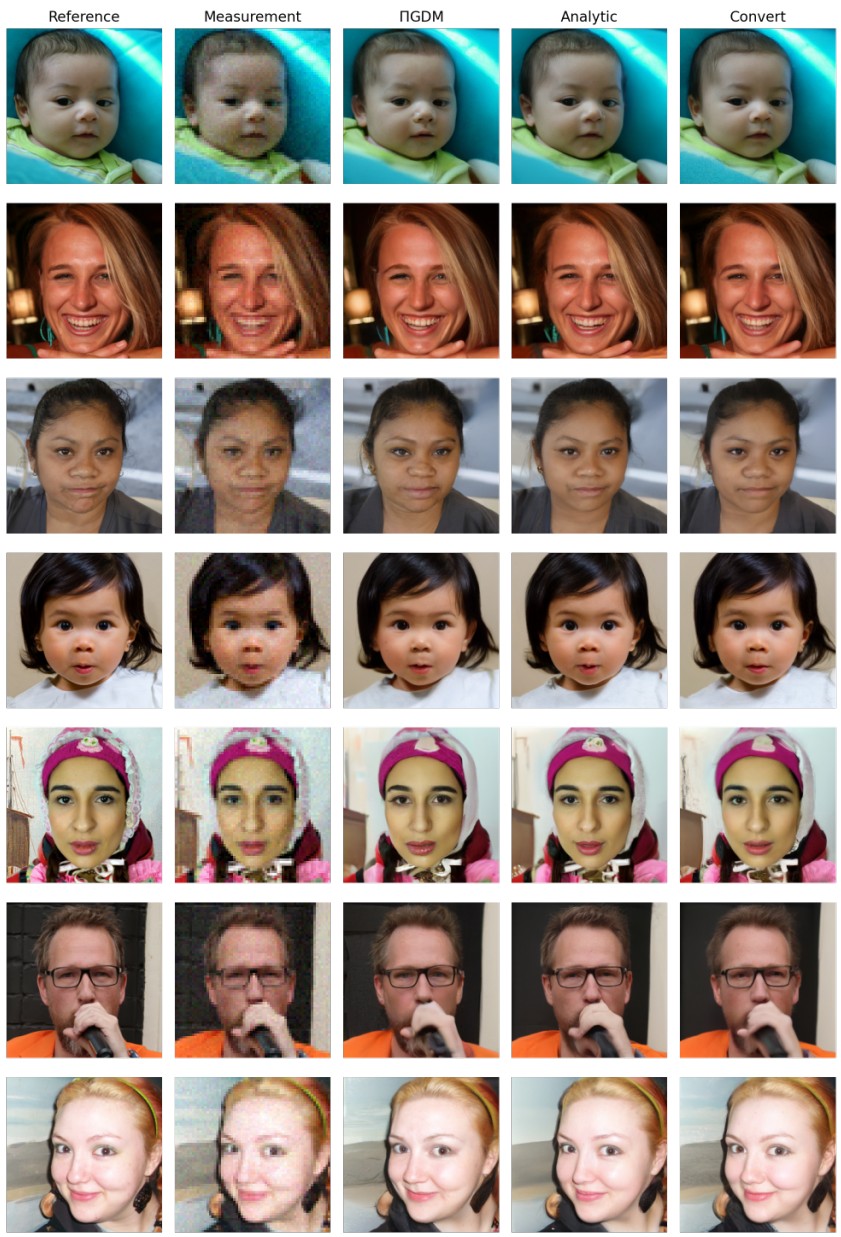

Figure 19: **Qualitative results for complete ΠGDM on super resolution** (4×).

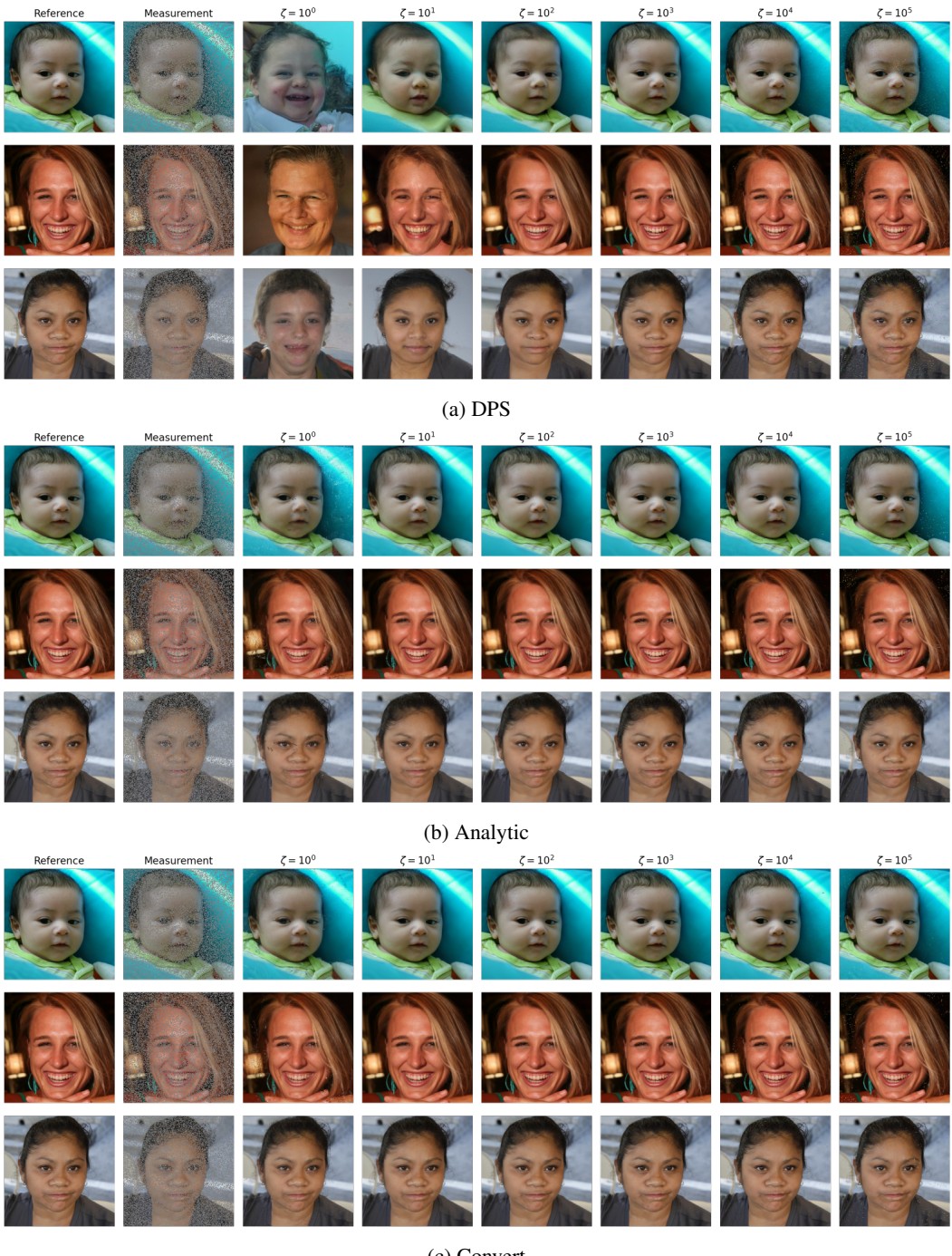

Figure 20: **Qualitative results for complete version of DPS on random inpainting.** As can be seen, our methods significantly improve DPS robustness to the hyper-parameter $\zeta$.

