# OpenReview forum: "Improving Diffusion Models for Inverse Problems Using Optimal Posterior Covariance"
_ICLR.cc/2024/Conference — Submitted to ICLR 2024_

### Official Review · Reviewer_FU6y · 2023-10-15

**Soundness:** 3 good
**Presentation:** 3 good
**Contribution:** 3 good
**Rating:** 8
**Confidence:** 5

**Summary:**

The paper is constructed into two parts. In the first part, a framework that unifies prior diffusion model-based inverse problem solvers (excluding the DDRM family) is proposed. It is shown that even the approaches that did not explicitly aimed for a Gaussian approximation of the posterior $p(\mathbf{x}_0|\mathbf{x}_t)$ implicitly makes this assumption, and the only difference is in the computation of the conditional posterior mean $\mathbb{E}[\mathbf{x}_0|\mathbf{x}_t,\mathbf{y}]$. In the second part, several methods on computing the optimal diagonal covariance of the variational Gaussian posterior $q(\mathbf{x}_0|\mathbf{x}_t)$ are proposed. Numerical experiments validate that using these optimal covariance to derive the step sizes of the gradient improves the performance of the heuristically-chosen step sizes.

**Strengths:**

1. Overall, the paper is well-written, clear, concise, with solid experiments.

2. This is the first work that shows that [3,4] can also be interpreted as approximating the posterior $p(\mathbf{x}_0|\mathbf{x}_t)$ with a Gaussian, similar to [1,2]. Such connection was non-trivial, and it is useful that one can understand [1-4] in a unified framework.

3. Theorem 1 is useful as many of the previous works leverage a pre-trained model that can estimate the reverse diagonal covariance, but simply discards it during inference. It is good that one can leverage such un-used information to acquire the optimal posterior covariance through variational inference. Moreover, the proof obtained from variational calculus is clean.

4. Previous approaches [1,2,4] have hyper-parameters that are hard to decipher and choose from. The proposed method partially alleviates this issue. (It is partially alleviated since it still resorts to the pre-configured hyperparameters of [2,4] when $\sigma_t \geq 0.2$.

5. On the practical side, it is the first time that I see a diffusion model-based inverse problem solver that is based on a 2nd order solver. All previous works that I am aware of utilize DDIM/DDPM sampling.



**References**

[1] Chung, Hyungjin, et al. "Diffusion posterior sampling for general noisy inverse problems." ICLR 2023
[2] Song, Jiaming, et al. "Pseudoinverse-guided diffusion models for inverse problems." ICLR 2023
[3] Wang, Yinhuai, Jiwen Yu, and Jian Zhang. "Zero-shot image restoration using denoising diffusion null-space model." ICLR 2023
[4] Zhu, Yuanzhi, et al. "Denoising Diffusion Models for Plug-and-Play Image Restoration." CVPRW 2023.

**Weaknesses:**

1. The construction of the theory mostly follows [2] (except that the proof is simpler), and hence the contribution could be limited.

2. Proposition 1 was also proposed in [1] for estimating $\mathbb{E}[\mathbf{x}_0|\mathbf{x}_t,\mathbf{y}]$ through DPS approximation. It should be worth citing and discussing.

3. For readers' convenience, it is worth highlighting that Type II circumvents the computation of $\nabla_{\mathbf{x}_t}$ that requires expensive backpropagation as a closed-form solution for deriving $\mathbb{E}[\mathbf{x}_0|\mathbf{x}_t,\mathbf{y}]$ exists, which is the key difference for the two types.

4. It is unclear what the authors mean by the first sentence of 5.1. paragraph **Posterior variances prediction**. By *ground-truth* square errors, do they mean that they computed the pixel-wise variance from multiple posterior mean that was obtained through the denoiser? Clarification is needed. If so, should this really be called *ground truth*?

5. There is a discrepancy between theory and practice. For one, the authors use a stochastic sampler for Type II, which would not be sampling from (4). How bad are the samples if one simply uses Type II with a deterministic sampler? Any reasons why a deterministic sampler would under-perform in this case?

6. How bad are the samples if one uses the predicted optimal posterior covariance for all the timesteps?

**References**

[1] Ravula, Sriram, et al. "Optimizing Sampling Patterns for Compressed Sensing MRI with Diffusion Generative Models." arXiv preprint arXiv:2306.03284 (2023).

[2] Bao, Fan, et al. "Analytic-dpm: an analytic estimate of the optimal reverse variance in diffusion probabilistic models." ICLR 2022.

**Questions:**

1. For estimating the optimal posterior variance with (24), how long does this computation take? Is it a bottleneck for inference?

2. The authors state that they only use the proposed variance when $\sigma_t < 0.2$. Out of 50 steps, how many steps does this correspond to?

3. DPS is known to excel with 1000 NFE, $\Pi$GDM with 100 NFE, etc. Is there any specific reason why the authors chose 50 NFE?

4. (12pg A.1. first sentence) here we proof --> here we prove

5. Is $\tilde\beta_t$ defined somewhere in the paper?

---

> ### Author Response · Authors · 2023-11-23
> **Our response to Reviewer FU6y (first part)**
>
> ## W1: Limited contribution
> We thank the reviewer for comments on our paper. We would like to emphasize that the main contribution of this paper is to provide a novel framework that can unify existing diffusion-based zero-shot methods for inverse problems. This framework can address all the cases, including training-from-scratch, pre-trained models with and without reverse covariances. As recognized by the reviewer, it is useful to convert reverse covariances for many existing models and this novel idea has been long neglected. It is non-trivial to reveal the concept of using the optimality theorem of DDPM [8,9] to transform reverse variance predictions into posterior variance predictions, despite that the construction of Theorem 1 mostly follows [4]. We have also claimed this point as our contribution in the paper.
>
> ## W2 & W3: Add Citation & Highlight difference between type I and II
> We thank the reviewer for the constructive suggestion. In our revised version, the citation and discussion have been included in the last paragraph of Section 3.1, and the difference has been highlighted in the first paragraph of Section 3.2.
>
> ## W4: Confusion of validation of posterior variances prediction
> The computation of the ground-truth square error involves the following steps:
> 1. We sample a clean image $x_0$ from the training set.
> 2. We inject Gaussian noise to $x_0$ to generate a noisy image $x_t=x_0 + \sigma_t \epsilon, \epsilon\sim \mathcal{N}(0, I)$.
> 3. We employ the pretrained diffusion model to denoise $x_t$, resulting in $\hat{x}_0 = D_t(x_t) \approx \mathbb{E}[x_0|x_t]$.
> 4. We calculate the pixel-wise square errors as $e = (x_0 - \hat{x}_0)^2$.
>
> We call $e$ the ground-truth square errors as it is the target (ground truth) that we want the posterior variances prediction $\hat{r}_t^2(x_t)$ to approximate. We compare $e$ with $\hat{r}_t^2(x_t)$ because posterior variances $r_t^{*2}(x_t)$ is the MMSE estimator of $e$ given $x_t$ (assuming $D_t(x_t) = \mathbb{E}[x_0|x_t]$), as suggested by equation 17, implying that a good posterior variances prediction $\hat{r}_t^2(x_t)$ should be a reliable predictor of $e$. Therefore, we may use the comparison between $e$ and $\hat{r}_t^2(x_t)$ as a sanity check to gauge the effectiveness of equation 22 in practice. As shown in Figure 2, $e$ is indeed accurately predicted by $\hat{r}_t^2(x_t)$ computed using equation 22 in regions with low noise levels. However, we encounter numerical issues using equation 22 in regions with high noise levels, and we attribute this to the fact that equation 22 approaches a 0/0 limit in these regions.
>
> ## W5: Sampling using SDE for Type II
> As noted in footnote 1, replacing the posterior mean with the conditional posterior mean (e.g., from equation 2 to equation 4 for ODE) to achieve conditional sampling is not limited to diffusion models formalized by ODEs. The extension to DDPM, DDIM, or SDE-formalized diffusion models can be similarly proved by leveraging conditional Tweedie’s formula (Lemma 2). Therefore, there is no discrepancy between theory and practice. We chose ODEs (from equations 2 to equation 4) due to the comparatively simpler mathematical form than SDEs and discrete diffusion models like DDPM or DDIM.
>
> However, due to discretization in practice, the characteristics of samples based on ODEs and SDEs differ. When discretizing SDEs, a large step size (small number of steps) often causes non-convergence [3]. Hence, ODEs are more favored for designing fast samplers than SDEs [2,3]. However, SDEs excel at correcting errors made at each step [2], which ODEs lack. This may explain why Type II performs better using a stochastic sampler than a deterministic one, as it can correct errors caused not only by discretization but also by the approximation from the conditional posterior mean $\mathbb{E}_q[x_0|x_t,y]$ to the true one $\mathbb{E}[x_0|x_t,y]$.
>
> To see example samples of Type II using a deterministic sampler, please refer to "comparison_between_deterministic_and_stochastic_sampler_for_type_II.png" in the Supplementary Materials, which is generated by running DiffPIR with $\lambda=10$ using deterministic and stochastic_samplers. To see more results of Type II using a deterministic sampler, please run
> ```
> bash quick_start/rebuttal/eval_guidance_II_ode.sh
> ```
> using our source code provided in the Supplementary Material.

---

> ### Author Response · Authors · 2023-11-23
> **Our response to Reviewer FU6y (second part)**
>
> ## W6: How bad are the samples when using optimal variances for all steps
> The experiment results for Type I guidance using Analytic-type posterior variances for all sampling steps (referred to as Analytic (all steps)) are shown in the following tables and have been included in Appendix E of our revised version. We do not conduct experiments for Convert-type posterior variances since we cannot obtain accurate posterior variances prediction using Equation 22 at high noise level region (see 5.1 Validation of theoretical results, posterior variances prediction).
>
> **Inpainting:**
> | Method | PSNR | SSIM | LPIPS | FID |
> | :---: | :---: | :---: | :---: | :---: |
> | Analytic (all steps) | 33.97 | 0.9242 | 0.0832 | 25.77
> | Analytic | 33.76 | 0.9272 | 0.0845 | 28.83
>
> **Gaussian debluring:**
> | Method | PSNR | SSIM | LPIPS | FID |
> | :---: | :---: | :---: | :---: | :---: |
> | Analytic (all steps) | 26.11 | 0.6889 | 0.2995 | 99.59
> | Analytic | 27.71 | 0.7926 | 0.1850 | 53.09
>
> **Motion debluring:**
> | Method | PSNR | SSIM | LPIPS | FID |
> | :---: | :---: | :---: | :---: | :---: |
> | Analytic (all steps) | 21.77 | 0.5795 | 0.3917 | 149.90
> | Analytic | 26.73 | 0.7579 | 0.2183 | 64.77
>
> **Super resolution:**
> | Method | PSNR | SSIM | LPIPS | FID |
> | :---: | :---: | :---: | :---: | :---: |
> | Analytic (all steps) | 27.31 | 0.7763 | 0.2016 | 56.90
> | Analytic | 27.58 | 0.7878 | 0.1968 | 59.83
>
> As can be seen, while using optimal variances in all steps shows good performance in tasks such as inpainting and super-resolution, there is a significant performance decrease for the deblurring tasks. Example samples are given in the file “debluring_with_analytic_variances_for_all_steps.png” of Supplementary Material. Our solution, which considers sampling with optimal variance only in regions with low noise levels, demonstrates robust performance across all tasks, including the new colorization tasks brought up by Reviewer rzGL.
>
> ## Q1: Computational cost of equation 24
> Similar to [4], a small number of Monte Carlo samples is typically adequate for obtaining an accurate $\hat{r}_t$. With a pre-trained unconditional model and training dataset, we compute $\hat{r}_t$ offline and deploy it alongside the pre-trained unconditional model. As a result, the online inference cost of our method is the same as previous methods. In our paper, we initially calculate $\hat{r}_t$ using a randomly selected $0.5 \%$ of the training set and reuse it for all our experiments. This process takes a couple of hours on a single 1080Ti GPU.
>
> ## Q2: Number of sampling steps with optimal posterior variances
> As we sample more densely in regions with low noise levels suggested by [2], it corresponds to 12 out of 50 steps. We have added this information in footnote 7 of our revised version.

---

> ### Author Response · Authors · 2023-11-23
> **Our response to Reviewer FU6y (third (and last) part)**
>
> ## Q3: Why we choose 50 sampling steps?
> We implement our techniques based on an open-source diffusion codebase k-diffusion (https://github.com/crowsonkb/k-diffusion), where using 50 sampling steps is a default choice.
> To further demonstrate the effectiveness of our method, we conducted additional experiments with **100** and **1000** sampling steps using **Euler's one order** sampler (deterministic) following the conventional choice of DPS and $\Pi$ GDM, shown in the following tables.
>
> **Inpainting:**
>
> | Method (100 steps) | PSNR | SSIM | LPIPS | FID |
> | :---: | :---: | :---: | :---: | :---: |
> | DPS | 33.51 | 0.9111 | 0.1000 | 30.03
> | $\Pi$ GDM | 31.09 | 0.8807 | 0.1474 | 53.66
> | Analytic (Ours) | 33.79 | 0.9281 | 0.0921 | 32.85
> | Convert (Ours) | **33.72** | **0.9294** | **0.0869** | **29.78**
>
>
> | Method (1000 steps) | PSNR | SSIM | LPIPS | FID |
> | :---: | :---: | :---: | :---: | :---: |
> | DPS | 32.96 | 0.9166 | 0.0909 | 31.20
> | $\Pi$ GDM | 31.16 | 0.8818 | 0.1419 | 50.57
> | Analytic (Ours) | **33.70** | 0.9268 | 0.0897 | 31.92 | 59.19
> | Convert (Ours) | 33.60 | **0.9281** | **0.0839** | **29.34**
>
>
> **Gaussian debluring:**
>
>
> | Method (100 steps) | PSNR | SSIM | LPIPS | FID |
> | :---: | :---: | :---: | :---: | :---: |
> | DPS | 26.79 | 0.7379 | 0.2553 | 83.36
> | $\Pi$ GDM | 27.90 | 0.8013 | 0.1898 | 60.41
> | Analytic (Ours) | **27.99** | **0.8053** | 0.1829 | 59.19
> | Convert (Ours) | 27.96 | 0.8040 | **0.1818** | **58.25**
>
>
>
> | Method (1000 steps) | PSNR | SSIM | LPIPS | FID |
> | :---: | :---: | :---: | :---: | :---: |
> | DPS | 27.15 | 0.7685 | 0.2011 | 61.53
> | $\Pi$ GDM | 27.70 | 0.7973 | 0.1904 | 60.75
> | Analytic (Ours) | 27.80 | 0.8012 | **0.1827** | 59.57
> | Convert (Ours) | **27.85** | **0.8024** | 0.1837| **58.37**
>
>
>
> **Motion debluring:**
>
>
> | Method (100 steps) | PSNR | SSIM | LPIPS | FID |
> | :---: | :---: | :---: | :---: | :---: |
> | DPS | 22.08 | 0.6086 | 0.3597 | 122.45
> | $\Pi$ GDM | 26.92 | 0.7681 | 0.2141 | 64.69
> | Analytic (Ours) | 26.96 | 0.7704 | 0.2124 | 64.18
> | Convert (Ours) | **26.98** | **0.7705** | **0.2103** | **64.12**
>
>
>
> | Method (1000 steps) | PSNR | SSIM | LPIPS | FID |
> | :---: | :---: | :---: | :---: | :---: |
> | DPS | 25.94 | 0.7331 | 0.2305 | 74.09
> | $\Pi$ GDM | 26.85 | 0.7637 | 0.2136 | 64.61
> | Analytic (Ours) | 26.85 | 0.7637 | 0.2110 | 62.35
> | Convert (Ours) | **26.90** | **0.7668** | **0.2103** | **61.78**
>
>
>
> **Super resolution:**
>
> | Method (100 steps) | PSNR | SSIM | LPIPS | FID |
> | :---: | :---: | :---: | :---: | :---: |
> | DPS | 27.56 | 0.7903 | **0.1883** | **57.79**
> | $\Pi$ GDM | 27.70 | 0.7970 | 0.1986 | 63.50
> | Analytic (Ours) | **27.78** | 0.7995 | 0.1952 | 65.31
> | Convert (Ours) | **27.78** | **0.8001** | 0.1963 | 65.61
>
>
>
> | Method (1000 steps) | PSNR | SSIM | LPIPS | FID |
> | :---: | :---: | :---: | :---: | :---: |
> | DPS | 26.59 | 0.7665 | 0.2068 | 70.49
> | $\Pi$ GDM | 27.60 | 0.7940 | 0.1985 | 63.15
> | Analytic (Ours) | 27.65 | **0.7979** | **0.1954** | 64.08
> | Convert (Ours) | **27.67** |0.7968  | 0.1957 | **62.37**
>
>
>
> ## Q4 & Q5: Grammar issues
> We thank the reviewer for the suggestions. We have fixed these issues in our revised version.
>
> ## Reference
> [1] Song, Y., et al. (2020). "Score-based generative modeling through stochastic differential equations." arXiv preprint arXiv:2011.13456.
>
> [2] Karras, T., et al. (2022). "Elucidating the design space of diffusion-based generative models." arXiv preprint arXiv:2206.00364.
>
> [3] Lu, Cheng, et al. "Dpm-solver: A fast ode solver for diffusion probabilistic model sampling in around 10 steps." Advances in Neural Information Processing Systems 35 (2022): 5775-5787.
>
> [4] Bao, F., et al. (2022). "Analytic-dpm: an analytic estimate of the optimal reverse variance in diffusion probabilistic models." arXiv preprint arXiv:2201.06503.
>
> [5] Bao, Fan, et al. "Estimating the optimal covariance with imperfect mean in diffusion probabilistic models." arXiv preprint arXiv:2206.07309 (2022).

---

> > ### Comment · Reviewer_FU6y · 2023-11-23
> >
> > I appreciate the authors for the thorough review. As pointed out in the initial review, I think this paper is a good contribution to the field. Therefore, I raised my score.

---

### Official Review · Reviewer_rzGL · 2023-10-28

**Soundness:** 3 good
**Presentation:** 3 good
**Contribution:** 2 fair
**Rating:** 5
**Confidence:** 4

**Summary:**

This paper provides a unified view of the existing linear inverse samplers with diffusion models. Motivated by this, they proposed an improved approach by calculating the optimal covariance in the Gaussian approximation. Experimental results show that the proposed approach outperforms previous methods in most cases.

**Strengths:**

1. The unified view of various existing methods to solve linear inverse problems with diffusion models is interesting and useful.
2. The proposed improved approach (both convert and analytic) by optimal covariance estimation performs well.

**Weaknesses:**

1. There is a lack of analysis or experimental results on the inference speed of the proposed method.
2. As mentioned by the authors, the proposed approach performs badly if it is implemented purely with the optimal variance, which is wired and unreasonable, even contradicting the starting point of this paper.  From a practical implementation side, how to decide where we should switch to the optimal variance value for a specific problem?
3.  How does the proposed approach perform on linear tasks like colorization and denoising, compared with other methods like DPS and other variants?

**Questions:**

Please see the above

---

> ### Author Response · Authors · 2023-11-23
> **Our response to Reviewer rzGL (first part)**
>
> ## Q1: Analysis on inference speed
> The speed of inference of our method depends on whether Analytic or Convert is chosen.
>
> **Analytic:** Our method does not introduce any additional computational cost for online inference compared to previous methods, since the only computational cost arises from the **offline pre-calculation** of $\hat{r}_t^2$ using equation (24). For the offline pre-calculation, similar to [3], a small number of Monte Carlo samples is adequate for obtaining an accurate $\hat{r}_t$ from a pre-trained unconditional model and training dataset. In the paper, we pre-compute $\hat{r}_t$ using a randomly selected $0.5\%$ of the training set and reuse it for all the experiments. This process takes a couple of hours on a single 1080Ti GPU. When pre-computed, $\hat{r}_t$ is deployed along with the pre-trained unconditional model. As a result, our method is the same as previous methods in computational cost for online inference.
>
> **Convert:** The conjugate gradient (CG) method could introduce additional computational cost. However, since CG converges rapidly and is only employed during the last few sampling steps, its additional computational cost is negligible compared to the overall inference time. We have included the discussion in Appendix E.3 in our revised version.
>
> The results of inference speed on 1080Ti GPU are shown in the following tables.
>
> **Inference time (sec) for Type I using 50 steps Heun's 2nd sampler**
> | DPS | $\Pi$ GDM | Analytic | Convert |
> | :---: | :---: | :---: | :---: |
> | 15.07 | 15.07 | 15.07 | 15.56 |
>
>
> **Inference time (sec) for Type II using 50 steps Heun's 2nd sampler**
> | DiffPIR | Analytic | Convert |
> | :---: | :---: | :---: |
> | 6.28 | 6.28 | 6.66 |
>
>
> ## Q2: Concern to only using optimal variances at the last few sampling steps
> The Gaussian distribution effectively approximates the posterior only in the infinitesimal limit of the noise level [1,2]. As the noise level increases, the posterior distribution can evolve into a non-Gaussian multimodal distribution [2]. We also observed this phenomenon in our initial experiments (refer to 4.2 Quantitative results, Paragraph 2). To address this, we heuristically employed our techniques only in the last few sampling steps, where the noise level is sufficiently low for the Gaussian approximation to be effective. To decide when to switch to the optimal variance, we empirically observe that switching when $\sigma_t < 0.2$ is a very robust choice. We use this heuristic to switch to the optimal variance across all inverse problems and sampler setups and observe consistent performance improvements over baselines.

---

> ### Author Response · Authors · 2023-11-23
> **Our response to Reviewer rzGL (second (and last) part)**
>
> ## Q3: Performance on colorization and denoising
> We have performed additional experiments on colorization and demonstrate that our method has significant performance improvement over baselines like DPS, $\Pi$ GDM, and DiffPIR. Denoising is a special case of noisy inpainting when the elements of the mask are all one and we have already demonstrated the superiority of our method in inpainting. We believe that current experimental results on image inpainting, deblurring, super-resolution, and colorization are sufficient to validate the effectiveness of the proposed method. Below, we elaborate on the evaluations for colorization.
>
> The colorization experiments are performed on the FFHQ dataset. We provide quantitative results in terms of PSNR, SSIM, LPIPS, and FID to compare the proposed method with Type I guidance (i.e., DPS and $\Pi$ GDM) and Type II guidance (e.g., DiffPIR). Here, PSNR refers to how close the reconstructed image is to the ground truth in the sense of Euclidean distance, and SSIM, LPIPS, and FID are subjective metrics that reflect the perceptual quality. As shown in the following tables, our method yields evident performance gains, especially in terms of subjective metrics, over baselines on the colorization task. The proposed method significantly outperforms Type I guidance (i.e., DPS and $\Pi$ GDM) in all the metrics. Regarding Type II guidance like DiffPIR, we yield evident gains in LPIPS, FID, and SSIM and comparable PSNR. Note that, considering that multiple solutions can be consistent with the measurement, PSNR could not be a desirable metric for the task of colorization. To further validate our method, we provide the qualitative results by visualizing the colorized images in "colorization_type_I.png" and "colorization_type_II_lambda=1.png" in the Supplementary Material. Figure "colorization_type_I.png" shows that our methods significantly outperform baselines in Type I. Figure "colorization_type_II_lambda=1.png" demonstrates that although DiffPIR is slightly better than ours in PSNR, its image contains more artifacts than ours.
>
>
> **Quantitative results (PSNR, SSIM, LPIPS, FID) on Colorization for Type I guidance:**
> | Method | PSNR | SSIM | LPIPS | FID |
> | :---: | :---: | :---: | :---: | :---: |
> | DPS       | 20.40 | 0.7804 | 0.2537 | 85.46 |
> | $\Pi GDM$ | 21.11 | 0.8406 | 0.2440 | 82.38 |
> | Analytic (Ours) | 21.24 | 0.8645 | 0.2100 | 76.87 |
> | Convert (Ours)  | **21.47** | **0.8662** | **0.2089** | **76.12** |
>
> **Quantitative results (SSIM) on Colorization for Type II guidance under different $\lambda$:**
> | Method \ $\lambda$ | $10^{-2}$ | $10^{-1}$ | $10^{0}$ | $10^{1}$ | $10^{2}$ |
> | :---: | :---: | :---: | :---: | :---: | :---: |
> | DiffPIR | 0.7333 | 0.7397 | 0.7848 | 0.7791 | 0.6824
> | Analytic (Ours) | **0.7967** | 0.7945 | **0.7979** | 0.7896 | 0.7546
> | Convert (Ours) | 0.7927 | **0.7947** | 0.7973 | **0.7924** | **0.7604**
>
> **Quantitative results (LPIPS) on Colorization for Type II guidance under different $\lambda$:**
> | Method \ $\lambda$ | $10^{-2}$ | $10^{-1}$ | $10^{0}$ | $10^{1}$ | $10^{2}$ |
> | :---: | :---: | :---: | :---: | :---: | :---: |
> | DiffPIR | 0.3634 | 0.3529 | 0.3040 | 0.2765 | 0.3526
> | Analytic (Ours) | 0.2671 | **0.2634** | 0.2617 | 0.2738 | 0.3150
> | Convert (Ours) | **0.2661** | 0.2655 | **0.2594** | **0.2677** | **0.2992**
>
> **Quantitative results (FID) on Colorization for Type II guidance under different $\lambda$:**
> | Method \ $\lambda$ | $10^{-2}$ | $10^{-1}$ | $10^{0}$ | $10^{1}$ | $10^{2}$ |
> | :---: | :---: | :---: | :---: | :---: | :---: |
> | DiffPIR         | 98.42 | 97.90 | 95.36 | **85.00** | 99.10
> | Analytic (Ours) | **86.23** | **85.64** | 85.56 | 85.77 | 96.96
> | Convert (Ours)  | 87.04 | 88.31 | **83.48** | 86.26 | **92.40**
>
> **Quantitative results (PSNR) on Colorization for Type II guidance under different $\lambda$:**
> | Method \ $\lambda$ | $10^{-2}$ | $10^{-1}$ | $10^{0}$ | $10^{1}$ | $10^{2}$ |
> | :---: | :---: | :---: | :---: | :---: | :---: |
> | DiffPIR | **21.26** | **21.07** | **21.16** | 20.57 | 19.77
> | Analytic (Ours) | 20.72 | 20.97 | 21.07 | **20.93** | 20.27
> | Convert (Ours) | 20.56 | 20.91 | 20.83 | 20.79 | **20.77**
>
> ## Reference
> [1] Sohl-Dickstein, Jascha, et al. "Deep unsupervised learning using nonequilibrium thermodynamics." International conference on machine learning. PMLR, 2015.
>
> [2] Xiao, Zhisheng, Karsten Kreis, and Arash Vahdat. "Tackling the generative learning trilemma with denoising diffusion gans." arXiv preprint arXiv:2112.07804 (2021).
>
> [3] Bao, F., et al. (2022). "Analytic-dpm: an analytic estimate of the optimal reverse variance in diffusion probabilistic models." arXiv preprint arXiv:2201.06503.

---

### Official Review · Reviewer_8Q2y · 2023-11-01

**Soundness:** 2 fair
**Presentation:** 3 good
**Contribution:** 2 fair
**Rating:** 5
**Confidence:** 4

**Summary:**

The paper establish a unified framework for zero-shot inverse problem solvers based on pre-trained diffusion model.
Especially, they unified two different approaches to approximate $\mathbb{E}[x_0|x_t, y]$ (one by likelihood estimation and the other by proximal optimization) into isotropic Gaussian approximation, which is a novel interpretation.
Based on the unified framework, the paper proposes to optimize the covariance for the approximated posterior via maximum likelihood estimation, which demonstrates effectiveness on various inverse problems.

**Strengths:**

- The proposed interpretation is novel. It seems to be a good try to connect different approaches.
- The paper is clearly written so easy to follow. Specifically, I like how the paper provides three possible scenarios for estimating the posterior distribution from the case without pre-trained model to the case with pre-trained model that only predicts the posterior mean.
- Underlying theories are well-aligned with prior works.

**Weaknesses:**

- The approximation of posterior by the Gaussian distribution may limit performance and interpretability of the diffusion process. Especially, it inevitably leads to interpretation of diffusion model as sampling through trajectory between two Gaussian distributions.
- Leveraging diagonal covariance is easy to lose significant information in images so that the performance of the proposed method would be limited.
- The performance of the proposed method seems limited (i.e. comparable to baselines for multiple tasks)
- Section 4.3 is the most useful case where previous methods are depending on. However, the proposed method is only about optimizing $r^2_t$ which was hand-crafted in the previous method, which limits the novelty of the proposed method.

**Questions:**

**Soundness**
- I would like to ask whether approximating the conditional posterior distribution using the standard normal distribution is practical. Specifically, I believe this is equivalent to assuming that the posterior distribution follows a Gaussian distribution, which inevitably implies that the diffusion model is sampling through the trajectory between two normal distributions. This seems inadequate for describing the image generation. Can authors provide their opinions on this matter? This is a crucial point for the my review because the proposed method highly relies on this aspect (section 4).
- I agree with the statement that "letting all the elements of covariance be learnable is computationally demanding" and believe that restricting it to diagonal posterior covariance could be one solution. However, this raises questions about the significance of that covariance. It may lead to the neglect of crucial information, casting doubt on whether the proposed method can achieve "optimal" variances. Additionally, the diagonal covariance assumption implies that each pixel is independent, which may not hold true for images. Note that this assumption is also for the posterior distribution of clean image $x_0$ given noisy image $x_t$. These issues raise concerns about the soundness of the proposed method.

**Experiments**
- When we see the Figure 3, we can observe that the performance of the proposed method is nearly identical to that of DiffPIR when we use $\lambda=10$. This might suggest that the effectiveness of the covariance optimization is marginal, regardless of how the covariance  is computed (i.e. Analytic or Convert). Could authors provide further explanation on this point? Specifically, I wonder that whether this lack of improvement in performance can be attributed to either non-realistic approximation of Gaussian posterior or an excessively simplified covariance structure. If not, have any experimental result been obtained to support the idea that lack of performance improvement is not due to the Gaussian approximation or diagonal covariance?
- Image quality metrics such as PSNR and FID is missing in the Figure 3. Did they show the same robustness and the performance tendency compared to the baseline?
-  I noticed that the paper also presents the results for the 'complete version' of prior works in the appendix E. I understand that the authors moved these results to the appendix for the sake of comparison within the proposed unified framework. However, the 'complete version' of $\Pi$GDM and DPS involves controlling the weight of likelihood guidance, which is orthogonal to the posterior approximation and independent to the proposed unified framework. Hence, the 'complete version' should have been reported in the main paper, as they demonstrate the effectiveness of (adaptive) likelihood guidance in solving the inverse problem without the need to optimize the covariance of approximated posterior distribution. From this perspective, when we see the Figure 4 in the appendix E, the performance of DPS and the proposed method are comparable with proper guidance. This raises the question about the effectiveness of the proposed method once again.

**minors**
- In the last sentence of the section 5.1, why the equation 22 becomes 0/0 when $v^{*2}(x_t)-\tilde{\beta}_t \approx 0$? According to the equation 22, it should be 0 rather than 0/0.  If I overlooked anything, it would be appreciated pointing it out.

---

> ### Author Response · Authors · 2023-11-23
> **Our response to Reviewer 8Q2y (first part)**
>
> ## W1 & Q1: Justification of Gaussian approximation
> As discussed in Section 3 in the paper, recent zero-shot methods, including DPS [4], $\Pi GDM$ [6], DiffPIR [5] and DDNM [7], all make Gaussian approximation with isotropic covariance $q_t(x_0|x_t)$ for the true unconditional posterior $p_t(x_0|x_t)$. They explicitly or implicitly leverage HANDCRAFTED design of isotropic posterior covariances and have achieved empirical success. Song et al. (2023) justify that the score of Gaussian approximation is close to that of the true posterior under the assumption that the denoiser behaves like a pseudo linear filter (see [4], Appendix A.6).
>
> Our approximation $q_t(x_0|x_t)$ is more accurate than previous methods by optimizing posterior variance (rather than using handcrafted design) without incurring additional computational costs and has demonstrated significant empirical improvements in solving various inverse problems. It should be noted that **we do not use standard normal distribution to approximate the conditional posterior distribution**. Actually, when $p(y|x_0)$ (by equation 1) and $q_t(x_0|x_t)$ are both Gaussians, the approximated conditional posterior $q_t(x_0|x_t,y)\propto p(y|x_0)q_t(x_0|x_t)$ is also a Gaussian (but not standard).
>
> Using complex distributions to model the posterior accurately may improve performance, but it will significantly increase computational costs. Gaussian approximation for the posterior is a reasonable choice for balancing computational cost and the accuracy of conditional posterior mean approximation. It is known that the Gaussian distribution is an effective approximation of the posterior in the infinitesimal limit of the noise level [1,2]. As the noise level increases, the posterior distribution can evolve into a non-Gaussian multimodal distribution [2]. We have also observed this phenomenon in our initial experiments (refer to 4.2 Quantitative results, Paragraph 2). To circumvent this problem, in this paper, we employ our techniques only in the last few sampling steps, where the noise level is sufficiently low for the Gaussian approximation to be effective.
>
> ## W2: Justification of diagonal covariance constraint
> In fact, there are currently no tractable methods to implement a full covariance matrix-based approach for high-dimensional signals like images, despite that using a more expressive posterior covariance could potentially enhance performance. We identify two main challenges: (1) the computational cost of predicting the full posterior covariance and (2) the computational cost of implementing guidance based on the full posterior covariance. The concurrent work [3] (also submitted to ICLR 2024) proposed a method to optimize the posterior covariance by computing the Hessian matrix of $\log p_t(x_t)$ (i.e., the derivative of the score function) to obtain a full covariance prediction from a pre-trained unconditional diffusion model. However, this method is only evaluated on low-dimensional synthesis data but still uses diagonal covariance for high-dimensional signals like images (see Appendix E.1 in [3]). Taking RGB images with a resolution of 256 $\times$ 256 as an example, it is cost-ineffective as it requires computing the Hessian matrix and performing $d$ (dimension of signals) times backpropagation (in our case, 256 $\times$ 256 $\times$ 3 times backpropagation). This process is significantly slower than our approach since we do not introduce additional computational costs to compute the posterior covariance. Since our paper focuses on real-world inverse problems, choosing a diagonal covariance is reasonable for balancing computational costs and performance.
>
> In addition, we focus on zero-shot solutions to inverse problems using available checkpoints for unconditional diffusion models since it is costly to train them from scratch. As mentioned in the response to W1 & Q1, existing works [4,5,6,7] make the Gaussian approximation with diagonal covariance for the true unconditional posterior. Our methods are built upon these milestone works, of which the improvement comes from carefully designed training-free methods for obtaining optimal diagonal covariances.

---

> ### Author Response · Authors · 2023-11-23
> **Our response to Reviewer 8Q2y (second part)**
>
> ## W3 & Q2 & Q3 & Q4 & Q5: Lack of performance improvement and missing metrics
> We sincerely apologize for any confusion caused by incorrect experimental results. These incorrectnesses were the consequence of the numerical issue from the implementation of our closed-form solutions for debluring. We have presented the corrected results for debluring tasks in the revised version of our paper. To provide a more thorough evaluation, we have included additional image quality metrics, such as PSNR, FID, and SSIM, in Appendix E of the revised version. To facilitate the reproduction of our results, we have provided our source code and quick start bash scripts in (Supplementary Materials, src). We are grateful for your understanding and patience.
>
> In Figures 3 and 4 in our revised version, we clearly find that our proposed method achieves better LPIPS performance than baselines under almost all hyperparameters. We have also provided the additional results on PSNR, SSIM and FID in Appendix E, which show the same robustness and the performance tendency compared to baselines.
>
> We can also find that DiffPIR is inferior to our method under most values of $\lambda$ due to its handcrafted design. In equation (10) in the revised version, $\sigma_t$ is related to $\lambda$ and affects the performance with varying $\lambda$. Our method optimizes $\sigma_t$ for better performance. In the case of $\lambda=10$, DiffPIR coincidently finds a similar value to our estimation and yields a similar performance (see "diffpir_var_vs_analytic_var.png" in the Supplementary Material for details).
>
> ## W4: Limited novelty
> Our paper proposes a unified framework for diffusion-based zero-shot methods. As claimed in the paper, our novelty is three-fold.
>
> First, we provide an insight of interpreting existing methods from the view of approximating conditional posterior mean for the reverse process and reveal that they are equivalent to making isotropic Gaussian approximation with different handcrafted isotropic posterior covariances. This finding enables us to transform the handcrafted design in existing methods (classified as Type I and Type II guidance) to a general solution to optimization via maximum likelihood estimation. It is especially non-trivial for Type II guidance in which posterior variances could not be optimized due to the implicit assumption of a Gaussian distribution.
>
> Second, we develop three approaches to address all the cases for realizing diffusion-based zero-shot methods, including training-from-scratch, and pre-trained models with and without reverse covariances, respectively. Remarkably, both pre-trained models with and without reverse covariances are useful to existing methods as elaborated below.
>
> **Pretrained models with reverse covariance.** In Section 4.2, we are the first to reveal the concept of utilizing the optimality theorem of DDPM [8,9] to transform reverse variance predictions into posterior variance predictions. This result allows one to leverage previous works’ pre-trained models that estimate reverse covariance to predict posterior variance for inverse problems and enhance performance without incurring additional costs. Actually, unconditional reverse covariance is ignored in many of the existing methods for inverse problems, despite it can be calculated.
>
> **Pretrained models without reverse covariance.** In Section 4.3, optimizing $r_t^2$ is not trivial for pre-trained models without reverse covariances. In fact, existing methods [4,5,6,7] all focus on finding optimal variance but they resort to handcrafted design. Contrary to them, we theoretically develop the optimal solution to the problem and empirically achieve MMSE estimation via Monte Carlo samples.
>
> Third, as highlighted in the revised version, we offer an efficient way of calculating guidance in noisy inverse problems. $\Pi GDM$ does not provide an efficient method for implementing guidance in noisy inverse problems, as it requires the computation of high-dimensional matrix inversion for the vector-Jacobian product (equation 91). As a result, $\Pi GDM$ did not evaluate the performance on the noisy deblurring and noisy super-resolution (they conduct the noisy inpainting experiments as the guidance calculation is trivial). In contrast, we propose closed-form solutions to Type I guidance in Appendix B.1 (for inpainting, deblurring, super-resolution, and colorization) that can be efficiently evaluated. In the cases that the closed-form solutions to guidance are complex to derive, we further present numerical solutions in Appendix C based on the conjugate gradient (CG) method for Type I and II guidance, thereby enhancing the practicality of our approach.

---

> ### Author Response · Authors · 2023-11-23
> **Our response to Reviewer 8Q2y (third (and last) part)**
>
> ## Q5: Moving results for "complete version of prior works" to the main paper
> We thank the reviewer for the constructive suggestion. We put these results into Appendix due to page limitation. We will try to highlight these results in our final version. For the performance concerns, please refer to our reply for "Lack of performance improvement".
>
> ## Q6: Why equation 22 is a 0/0 limit?
> When noise levels are high (or when $t$ is large), the posterior variances cannot be zero (LHS of equation 22) due to the considerable uncertainty of the clean image $x_0$ given $x_t$. Therefore, if the numerator $\hat{v}_t^2(x_t) - \tilde{\beta}_t$ in equation 22 approaches zero, it necessitates that the denominator $(\frac{\sqrt{\bar{\alpha}\_{t-1}}\beta_t}{1-\bar{\alpha}_t})^2$ also approaches zero. This can be understood by directly examining $(\frac{\sqrt{\bar{\alpha}\_{t-1}}\beta_t}{1-\bar{\alpha}_t})^2$. In the case of DDPM, where $x_t=\sqrt{\bar{\alpha}_t}x_0+\sqrt{1-\bar{\alpha}_t}\epsilon$, and for large $t$, $\sqrt{\bar{\alpha}_t}\rightarrow 0$, it follows that $(\frac{\sqrt{\bar{\alpha}\_{t-1}}\beta_t}{1-\bar{\alpha}_t})^2 \rightarrow 0$ for large $t$.
>
> ## Reference
> [1] Sohl-Dickstein, Jascha, et al. "Deep unsupervised learning using nonequilibrium thermodynamics." International conference on machine learning. PMLR, 2015.
>
> [2] Xiao, Zhisheng, Karsten Kreis, and Arash Vahdat. "Tackling the generative learning trilemma with denoising diffusion gans." arXiv preprint arXiv:2112.07804 (2021).
>
> [3] Boys, Benjamin, et al. "Tweedie Moment Projected Diffusions For Inverse Problems." arXiv preprint arXiv:2310.06721 (2023).
>
> [4] Song, Jiaming, et al. "Pseudoinverse-guided diffusion models for inverse problems." International Conference on Learning Representations. 2022.
>
> [5] Zhu, Y., et al. (2023). "Denoising Diffusion Models for Plug-and-Play Image Restoration." arXiv preprint arXiv:2305.08995.
>
> [6] Chung, Hyungjin, et al. "Diffusion posterior sampling for general noisy inverse problems." arXiv preprint arXiv:2209.14687 (2022).
>
> [7] Wang, Yinhuai, Jiwen Yu, and Jian Zhang. "Zero-shot image restoration using denoising diffusion null-space model." arXiv preprint arXiv:2212.00490 (2022).
>
> [8] Bao, F., et al. (2022). "Analytic-dpm: an analytic estimate of the optimal reverse variance in diffusion probabilistic models." arXiv preprint arXiv:2201.06503.
>
> [9] Bao, Fan, et al. "Estimating the optimal covariance with imperfect mean in diffusion probabilistic models." arXiv preprint arXiv:2206.07309 (2022).

---

> > ### Comment · Reviewer_8Q2y · 2023-11-23
> > **Thanks for the response**
> >
> > I appreciate authors for their response.
> >
> >
> > **Gaussian approximation**
> >
> > I agree with author's response and my concern on it is resolved.
> >
> >
> > **Diagonal constraint**
> >
> > As I mentioned in the review comment, I agree that there is computational challenge in computing the covariance.
> > However, it is hard to agree that the diagonal covariance is the best way for balancing computational costs and performance.
> > For example, one can approximate the full covariance via low-rank decomposition or block matrix, which are more reasonable than the diagonal covariance. Note that the diagonal covariance implies independent pixels in a single image.
> > Have authors tried with other approximation than the diagonal covariance?
> >
> > **Experimental results**
> >
> >
> > I really appreciate authors for their effort in finding numerical issues in the implementation, which is hard to point out in short duration. However, in the revised version, the performance of the proposed method does not outperform Guidance Type II methods, especially DiffPIR.
> >
> > > In Figures 3 and 4 in our revised version, we clearly find that our proposed method achieves better LPIPS performance than baselines under almost all hyperparameters
> >
> > I agree that the proposed method is robust. However, for deblurring tasks, DiffPIR shows comparable (or better, cannot precisely say as numerical metrics are not reported) to the proposed method. Also, in Figure 8 and 10 in the revised paper, DiffPIR achieves better PSNR and SSIM than the proposed method.
> >
> > > Our method optimizes $\sigma_t$ for better performance. In the case of $\lambda=10$, DiffPIR coincidently finds a similar value to our estimation and yields a similar performance
> >
> > If this is true, why should we optimize the covariance, not just use DiffPIR with $\lambda=10$?
> >
> > Overall, I appreciate the response from the authors, but my major concerns remain. Therefore, I will keep my score at 5.

---

> ### Author Response · Authors · 2023-11-23
> **Our response to Reviewer 8Q2y**
>
> Thank you for your thoughtful response! Regarding the diagonal covariance, we have to emphasize that our paper aims to propose a unified framework for inverse problems and also considers to accommodate to existing diffusion-based zero-shot methods. To this end, it is hard to directly leverage widely adopted checkpoint of unconditional diffusion models used by existing methods without using diagonal covariance. The use of the diagonal constraint has another significant advantage: it enables our method to be seamlessly integrated with existing methods. This plug-and-play nature of our approach makes it the resonable step after recent works.

---

### Meta-Review · Area_Chair_SW4B · 2023-12-15

**Metareview:**

The paper establishes a unified framework for zero-shot inverse problem solvers based on a pre-trained diffusion model. Based on the unified framework, the paper proposes to optimize the covariance for the approximated posterior via maximum likelihood estimation, which demonstrates effectiveness on various inverse problems.

The reviewers are concerned about the soundness of the proposed diagonal covariance, and the marginal improvements compared with existing methods. Please consider incorporating the reviewers' comments for future submissions.

**Justification For Why Not Higher Score:**

Reviewers concerned about the soundness of the approach and the marginal improvements

**Justification For Why Not Lower Score:**

n/a

---

### Decision · Program_Chairs · 2024-01-16

Reject